# TGFβ links EBV to multisystem inflammatory syndrome in children

Carl Christoph Goetzke[1,2,3,4,5✉], Mona Massoud[1], Stefan Frischbutter[6,7], Gabriela Maria Guerra[1], Marta Ferreira-Gomes[1], Frederik Heinrich[1], Anne Sae Lim von Stuckrad[2,4], Sebastian Wisniewski[2], Jan Robin Licha[1], Marina Bondareva[1], Lisa Ehlers[1,2,3], Samira Khaldi-Plassart[8,9], Etienne Javouhey[10], Sylvie Pons[11], Sophie Trouillet-Assant[11,12], Yasemin Ozsurekci[13], Yu Zhang[14], Maria Cecilia Poli[15,16], Valentina Discepolo[17,18], Andrea Lo Vecchio[17], Bengü Sahin[2], Murielle Verboom[19], Michael Hallensleben[19], Anja Isabelle Heuhsen[1], Camila Astudillo[16], Yazmin Espinosa[16], Maria Cecilia Vial Cox[15], Kerry Dobbs[14], Ottavia M. Delmonte[14], Gina A. Montealegre Sanchez[14], Mary Magliocco[14], Karyl Barron[14], Jeffrey Danielson[14], Lev Petrov[20], Nadine Unterwalder[21], Birgit Sawitzki[20], Mareen Matz[3], Katrin Lehmann[1], Alexander Gratopp[2], Horst von Bernuth[2,3,22,23], Lisa-Marie Burkhardt[24], Niklas Wiese[24], Lena Peter[23], Michael Schmueck-Henneresse[23], Leila Amini[23,24], Marcus Maurer[6,7,37], Jobst Fridolin Roehmel[2,3,25], Benjamin E. Gewurz[26,27], Lael M. Yonker[28,29,30], Mario Witkowski[1,31,32], Andrey Kruglov[1,33], Marcus Alexander Mall[2,3,5,25], Helen C. Su[14], Seza Ozen[34], Andreas Radbruch[1], Alexandre Belot[8,35], Pawel Durek[1], Tilmann Kallinich[1,2,3,4,5,36✉] & Mir-Farzin Mashreghi[1,5,36✉]

In a subset of children and adolescents, SARS-CoV-2 infection induces a severe acute hyperinflammatory shock[1] termed multisystem inflammatory syndrome in children (MIS-C) at four to eight weeks after infection. MIS-C is characterized by a specific T cell expansion[2] and systemic hyperinflammation[3]. The pathogenesis of MIS-C remains largely unknown. Here we show that acute MIS-C is characterized by impaired reactivation of virus-reactive memory T cells, which depends on increased serum levels of the cytokine TGFβ resembling those that occur during severe COVID-19 (refs. 4,5). This functional impairment in T cell reactivity is accompanied by the presence of TGFβ-response signatures in T cells, B cells and monocytes along with reduced antigen-presentation capabilities of monocytes, and can be reversed by blocking TGFβ. Furthermore, T cell receptor repertoires of patients with MIS-C exhibit expansion of T cells expressing TCRVβ21.3, resembling Epstein–Barr virus (EBV)-reactive T cell clones capable of eliminating EBV-infected B cells. Additionally, serum TGFβ in patients with MIS-C can trigger EBV reactivation, which is reversible with TGFβ blockade. Clinically, the TGFβ-induced defect in T cell reactivity correlates with a higher EBV seroprevalence in patients with MIS-C compared with age-matched controls, along with the occurrence of EBV reactivation. Our findings establish a connection between SARS-CoV-2 infection and COVID-19 sequelae in children, in which impaired T cell cytotoxicity triggered by TGFβ overproduction leads to EBV reactivation and subsequent hyperinflammation.

At the beginning of the COVID-19 pandemic, children were only mildly affected by SARS-CoV-2, and were found to have immune advantages that protected them from the severe outcomes seen in older adults[6]. However, in April 2020, paediatric intensive care physicians in the UK observed a cluster of patients with hyperinflammatory shock linked to previous SARS-CoV-2 infection[1]. Simultaneously, paediatricians in Italy noted an increase in Kawasaki-like disease[7]. These patients exhibited symptoms of toxic shock and Kawasaki shock syndromes, typically starting four to eight weeks after SARS-CoV-2 infection[8]. Untreated, this hyperinflammation involving multiple organs (MIS-C) led to organ failure. Despite extensive studies, the pathogenesis of MIS-C remains

poorly characterized. Studies suggest that it involves impaired viral clearance and intestinal barrier dysfunction[9]. Although autoantibody formation[10,11] has been observed, the specificity of these antibodies varies across studies. For example, in one cohort, interleukin-1 receptor antagonist (IL-1Ra)-neutralizing antibodies were found in 13 out of 21 patients, alongside low levels of IL-1Ra[10], potentially explaining the widespread inflammation in these individuals. In another cohort, cross-reactive antibodies and T cells were found, but the cross-reactive T cells did not originate from the MIS-C-defining TCRVβ21.3[+] T cell subset[11], which is robustly expanded in MIS-C[2,12]. Some researchers hypothesize a superantigen-like immune reaction, as MIS-C is linked to

a unique T cell expansion, and/or activation associated with TCRVβ21.3[+] CD4 and CD8 T cells[2,12]. Structural models predict a superantigen-like region within the spike protein, which could bind to TCRVβ21.3 (ref. 12). However, pre-COVID-19 MIS-C data suggest that SARS-CoV-2 spike protein is not required for TCRVβ21.3[+] T cell expansion[13]. The pathophysiological role of TCRVβ21.3[+] T cells thus remains unclear. Additionally, autosomal recessive deficiencies in the oligoadenylate synthase (OAS)−RNAse L pathway in around 1% of children with MIS-C highlight the role of monocyte activation[14]. Despite advances, the pathobiology of MIS-C remains incompletely understood. Here we conducted a multi-centre study involving 6 centres across 4 continents, including 145 patients with MIS-C and 221 paediatric controls, to clarify the pathogenesis of MIS-C.

## TGFβ is uniformly upregulated in MIS-C

We screened cytokines and chemokines in patients with acute phase MIS-C as well as during follow-up visits, comparing the results with children with SARS-CoV-2 infection at 6 weeks post infection (wpi) as an at-risk control. We also sampled healthy children and children in the acute phase of SARS-CoV-2 infection (with mild or moderate, or severe symptoms) (Fig. 1a and Extended Data Fig. 1a–g). Cohort (Extended Data Table 1) characteristics are comparable to those in prior publications[3]. We observed upregulation of type 1, type 2, type 3 and IL-1 cytokines, as well as various chemokines and growth factors in the MIS-C group (Extended Data Fig. 1a–f). Given TGFβ1 is upregulated in severe COVID-19, we measured serum levels of TGFβ1 in the paediatric cohorts and compared them to previously published data on TGFβ1 in healthy young adults (below 30 years of age), patients with upper respiratory tract infection and SARS-CoV-2-infected adults with varying disease severity[4] (Fig. 1b). TGFβ1 serum levels in patients with MIS-C (median: 398 pg ml[−1]) resembled those in adults who were severely affected by COVID-19 (median: 415 pg ml[−1], $P \geq 0.9999$) and were approximately 3-fold higher than those in non-infected children (median: 132.2 pg ml[−1]), 2.6-fold higher than during acute SARS-CoV-2 infection (median: asymptomatic/mild, 150.1 pg ml[−1]; moderate/severe, 150.7 pg ml[−1]) and 7-fold higher than in paediatric controls at 6 wpi without MIS-C (6 wpi no MIS-C median: 63 pg ml[−1], $P = 0.0018$) or in healthy young adults (median: 64 pg ml[−1], $P = 0.0003$). A few paediatric patients in the at risk (6 wpi no MIS-C) group had high TGFβ1 levels that were linked to underlying rheumatic disease and anti-inflammatory therapy (tocilizumab or colchicine). TGFβ1 levels decreased after treatment of patients with MIS-C with immunoglobulin or methylprednisolone (Fig. 1b and Extended Data Fig. 1g,h), resolving hyperinflammation. These TGFβ1 levels showed a high negative correlation with therapy time (Spearman $r = −0.73$, $P = 0.0003$) (Extended Data Fig. 1g,h). These findings align with data from patients with severe COVID-19 that link dexamethasone treatment with reduced serum TGFβ1 (ref. 5). TGFβ−SMAD pathway variants are linked to genetic disease susceptibility[15] to Kawasaki disease, which closely resembles MIS-C, highlighting the role of TGFβ in hyperinflammatory syndromes.

In MIS-C, TGFβ upregulation is linked to previous SARS-CoV-2 infection or gut viral persistence[9], as the SARS-CoV-2 nucleoprotein interacts with SMAD3 and activates the TGFβ pathway[16], similar to the SARS-CoV nucleoprotein[17]. SMAD3 activation triggers a positive feedforward loop that upregulates TGFβ[18]. In addition, SARS-CoV-2 spike protein can interact with integrins to activate latent TGFβ[19]. Increased TGFβ may result from an excessive TGFβ response during acute SARS-CoV-2 infection in these children or from viral persistence[9]. *TGFB1* (encoding TGFβ1) has been used in a five-gene whole-blood RNA-expression signature to distinguish MIS-C from other inflammatory conditions[20], suggesting a key role for TGFβ in MIS-C.

As TGFβ is dependent on enzymatic cleavage for its activity, high serum levels may not correlate with biological function. We therefore tested whether sera from patients with MIS-C can induce phosphorylation of SMAD2 and SMAD3 (SMAD2/3). SMAD2/3 phosphorylation was significantly increased after incubating healthy donor T cells with sera from patients with MIS-C. The effect was significantly dampened in the presence of a neutralizing antibodies to TGFβ1, 2 and 3 (anti-TGFβ) (Fig. 1c and Extended Data Fig. 1i,j) or by overexpression of dominant-negative TGFBR2 in HEK293T cells (Extended Data Fig. 1k−m).

## MIS-C immune cells show TGFβ imprinting

We collected peripheral blood mononuclear cells (PBMCs) from 11 patients with acute phase MIS-C (Extended Data Table 1) and 4 children from the 6 wpi no MIS-C group. We performed single-cell RNA sequencing (scRNA-seq) of sorted, activated T cells (defined by expression of HLA-DR and CD38, as these markers have been previously identified to represent T cells activated via the T cell receptor (TCR)[21]), memory B cells, plasmablasts and monocytes (Extended Data Fig. 2). Uniform manifold approximation and projection (UMAP) separated by condition revealed distinct monocyte clusters that differed between controls and patient subgroups (Fig. 1d,e and Extended Data Fig. 2j,l). Monocyte clusters 3 (methylprednisolone-primed *CD163*[hi]) and 18 (methylprednisolone-primed *CD163*[hi] intermediate monocytes) were defined by high expression of *CD163* (Supplementary Data 1), resembling monocytes found in patients with severe COVID-19 (ref. 22). Others, however, have shown that in monocytes, *CD163* is upregulated by glucocorticoids[23]. To determine whether this observation is linked to hyperinflammation or treatment, we segregated the UMAP by individual patients (Extended Data Fig. 2j) and compared cell cluster frequencies on the basis of methylprednisolone doses (Fig. 1e and Extended Data Fig. 2l). The *CD163*[hi] monocyte clusters 3 (methylprednisolone-primed *CD163*[hi]) and 18 (methylprednisolone-primed *CD163*[hi] intermediate monocytes) were detected only in samples from patients with MIS-C who were treated with methylprednisolone, whereas monocytes from untreated patients clustered in cluster 7 (inflammation-primed monocytes). To confirm these glucocorticoid effects, we collected paired blood samples from four treatment-naive paediatric patients with newly diagnosed rheumatic diseases (two with systemic lupus erythematosus, one with juvenile idiopathic arthritis and one with morphea) before and after methylprednisolone pulse therapy (Extended Data Fig. 3). We did not observe major differences in the activated T cell or memory B cell compartments, and observed the appearance of clusters containing *CD163*-expressing monocytes only after methylprednisolone treatment (cluster 3 (methylprednisolone monocytes) and cluster 7 (methylprednisolone dendritic cells); Extended Data Fig. 3c−i and Supplementary Data 2). Clusters 3 (methylprednisolone-primed *CD163*[hi]) and 18 (methylprednisolone-primed *CD163*[hi] intermediate monocytes) reflect effects of methylprednisolone, whereas cluster 7 represents inflammation-primed monocytes in MIS-C. Effects of glucocorticoid treatment on activated T cells and memory B cells were limited (Extended Data Fig. 3).

The most prominent changes in MIS-C occurred within the activated T cell compartment (Fig. 1f), with *Ki-67*[+] proliferating T cells (median 1.4% versus 6.2%; $P = 0.0176$) and activated cytotoxic T cells (median 7.8% versus 13.4%; $P = 0.0557$) being particularly enriched in patients with MIS-C. The proliferation of CD38[+] and HLA-DR[+] T cells indicates recent activation of these T cells via the TCR[21,24]. Additionally, we observed decreased switched memory B cells, indicative of an impaired T cell−B cell interaction[25]. In two patients with MIS-C, we detected an increase in predominantly non-switched memory B cells expressing high levels of *CD83*, *CD69* and *TGFB1* (Fig. 1g). CD83 on B cells has been shown to be upregulated by activated T cells independently of antigen[26]. Similar B cells were previously found to be increased in patients with multiple sclerosis[27].

We assessed TGFβ1 activity in MIS-C using single-cell gene set enrichment analysis (GSEA). For this purpose, we analysed T cells, B cells and monocytes separately. Activated T cells, B cells and monocytes from

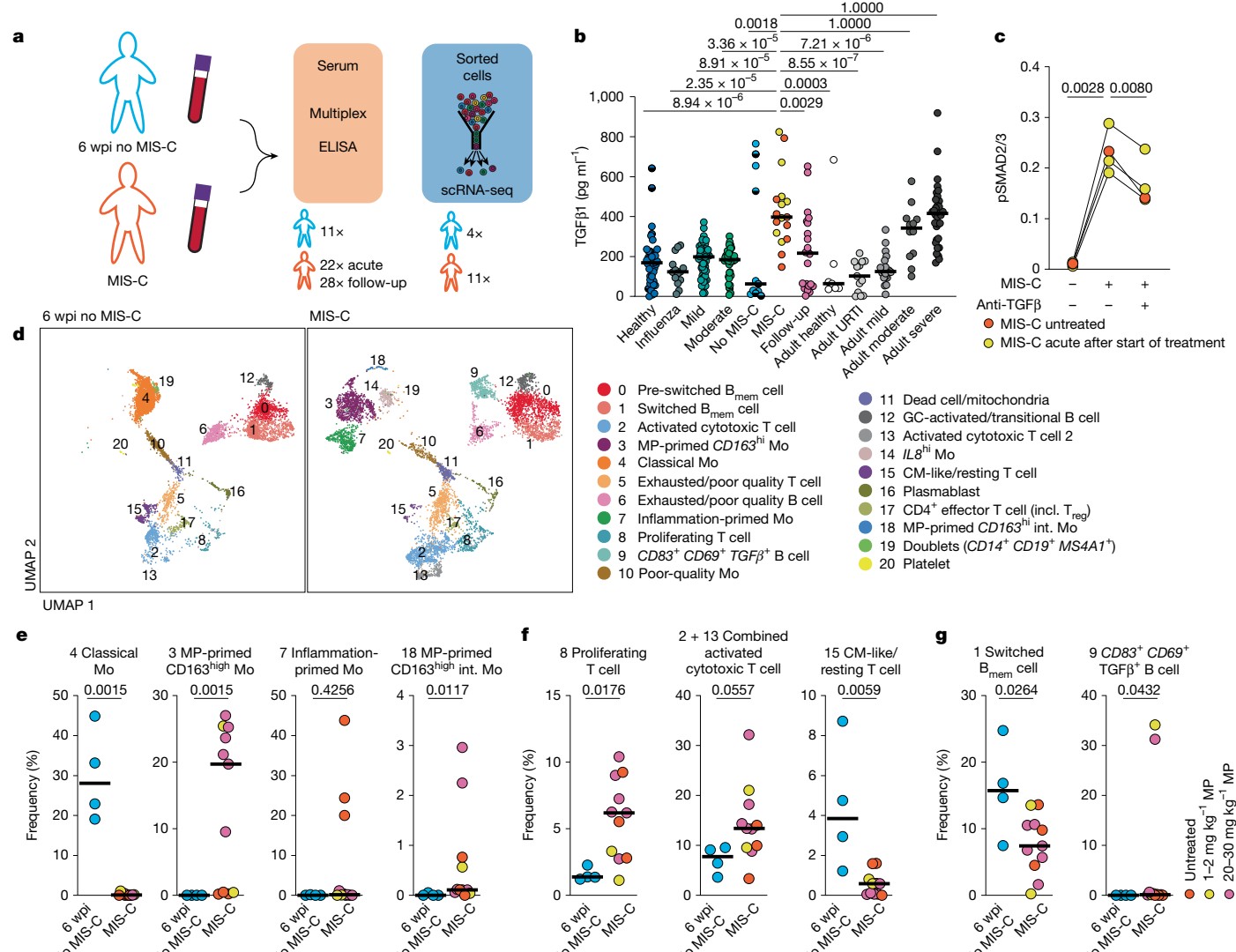

**Fig. 1 | Cytokine profile and effect on immune cell compartments in MIS-C.**
**a**, Schematic of the experimental setup. **b**, Serum TGFβ1 levels in patients with
MIS-C during the first 7 days of hospitalization ($n = 22$) and during follow-up
($n = 28$ time points, 23 patients) versus paediatric controls: 6 weeks after SARS-
CoV-2 infection (no MIS-C, $n = 11$, 5 with underlying rheumatic condition), acute
influenza (influenza, $n = 14$), asymptomatic or mild (mild, $n = 57$, 13 without
symptoms) or moderate or severe (moderate, $n = 42$, 2 with severe symptoms)
SARS-CoV-2 infection, and non-infected children (healthy, $n = 40$, 5 with
underlying rheumatic condition). Additional comparison was made with
previously published TGFβ1 levels[4] in healthy adults under 30 years of age
(adult healthy, $n = 7$), adults with upper respiratory tract infection (URTI)
(adult URTI, $n = 13$) and adults with COVID-19 in the first two weeks (adult mild,
$n = 19$; adult moderate, $n = 12$; adult severe, $n = 34$). **c**, T cells from healthy donors
($n = 2$) were incubated with sera from patients with MIS-C (MIS-C, $n = 4$; 3 after
start of treatment (yellow)) with or without neutralizing anti-TGFβ and SMAD2/3
phosphorylation (pSMAD2/3) was quantified using a capillary-based western
blot assay (Extended Data Fig. 1i–o). **d–g**, scRNA-seq of PBMCs enriched by

FACS for monocytes, HLA-DR$^{hi}$CD38$^+$ T cells and CD27$^+$ B cells from $n = 4$
paediatric controls (6 wpi no MIS-C) and $n = 11$ patients with MIS-C (sorting
strategy in Extended Data Fig. 2b). **d**, UMAP of 6,589 cells from 6 wpi no MIS-C
controls and 34,794 cells from the MIS-C group, separated by cohort and
rarefied. Cluster colours are unsupervised. UMAPs for individual patients are
presented in Extended Data Fig. 2j. B$_{mem}$, memory B cell; CM, central memory;
GC, germinal centre; incl., including; int., intermediate; Mo, monocyte; MP,
methylprednisolone. **e–g**, Frequencies of cells per cluster for clusters 3, 4, 7
and 18 (**e**), 8, 2 and 13, and 15 (**f**) and 1 and 9 (**g**). Methylprednisolone treatment
is colour-coded as indicated in **g**. Lines represent the median. Other cluster
frequencies are shown in Extended Data Fig. 2k–n. Kruskal–Wallis test with
Dunn's multiple comparison with correction for multiple comparisons,
comparing each group to patients with MIS-C (**b**); two-tailed repeated-measures
ANOVA with Geisser–Greenhouse correction with Holm–Šídák's multiple
comparison test with correction for multiple comparisons (**c**); two-tailed
Mann–Whitney $U$-tests (**e–g**).

patients with MIS-C showed significant upregulation of TGFβ-induced
gene sets using the Hallmark gene set 'TGF-β signalling' (Extended
Data Fig. 4a–c and Supplementary Data 3). For T cells, we used an
additional previously described gene set, which was generated from
cytokine-activated natural killer (NK) cells in the presence or absence
of TGFβ1 (ref. 4). This showed strong enrichment of TGFβ-induced
genes in the MIS-C group (Fig. 2a, T cells: median normalized enrich-
ment score (NES): 1.2 versus 1.4; $P = 1.304 \times 10^{-16}$). This underlines the

biological importance of the increased TGFβ1 serum levels that we
observed. Monocyte analysis revealed significant downregulation of
antigen-presentation genes in MIS-C (monocytes: 81% of cells with
significant GSEA have negative NES in MIS-C versus 18% in no MIS-C;
$P < 1 \times 10^{-15}$; Fig. 2b). TGFβ can induce monocyte deactivation in patients
with sepsis[28].

We validated these results using pseudobulk GSEA of all activated
T cells or monocytes from patients with MIS-C compared with children

without MIS-C (Extended Data Fig. 4t,u). We also included data from three individuals who were hospitalized owing to acute influenza infection (Extended Data Figs. 5 and 6a–u) as an additional inflammatory control condition. TGFβ imprinting in activated immune cells from MIS-C remained significantly enriched compared with non-inflammatory and inflammatory influenza infection controls (T cells Hallmark TGFβ signalling positive enrichment: 6 wpi no MIS-C, 66.7%; influenza, 80.3%; MIS-C, 96.1%, $P = 2.7 \times 10^{-4}$; MIS-C versus 6 wpi no MIS-C; for enrichment, $P = 2.0 \times 10^{-4}$; and for height of NES, $P = 4.7 \times 10^{-4}$ for MIS-C versus influenza) (Extended Data Fig. 6a,c–e). Moreover, the gene set for monocytic antigen processing and presentation remained selectively downregulated in activated immune cells isolated from patients with MIS-C (negative enrichment: 6 wpi no MIS-C, 25.0%; influenza, 37%; MIS-C, 88.6%; $P = 4.3 \times 10^{-5}$ MIS-C versus 6 wpi no MIS-C; $P = 1.8 \times 10^{-8}$ MIS-C versus influenza) (Extended Data Fig. 6b). To a lower extent, antigen processing and presentation genes were also downregulated in B cells from patients with MIS-C (B cells: 69% of cells with significant GSEA have negative NES in MIS-C versus 55% in no-MIS-C; $P = 3.93 \times 10^{-8}$) (Extended Data Fig. 4d).

Upregulation of NF-κB signalling and type I and II interferon signalling have previously been shown in MIS-C monocytes[3]. Similarly, we found upregulation of inflammatory pathways in T cells, B cells and monocytes from patients with MIS-C (Extended Data Fig. 4e–g). We observed a marked upregulation of chemokines and their receptors (Extended Data Fig. 4h,i) in MIS-C T cells and B cells. Although a downregulation of chemokines and their receptors was detected in MIS-C monocytes, most prominently in the methylprednisolone-primed *CD163*[hi] monocytes (29% negative enrichment in MIS-C versus 14% in controls; $P = 0.0002$) (Extended Data Fig. 4j). Simultaneously, we observed a highly significant induction of the Hallmark 'TNF signalling via NF-κB' (Extended Data Fig. 4k–m) and 'Interferon-α response' (Extended Data Fig. 4n–p) gene sets and to a lower deree, 'Interferon-γ response' (Extended Data Fig. 4q–s). These findings remained robust even when the false discovery rate (FDR) threshold of the normalized enrichment scores was increased, as suggested for bulk sequencing data analysis (Extended Data Fig. 6v–ap). However, only TGFβ was selectively upregulated in all MIS-C cell subsets when compared with influenza-induced hyperinflammation (Extended Data Fig. 6a–u). Other proinflammatory gene sets peaked in children with influenza, highlighting the role of TGFβ in MIS-C.

## TGFβ impairs T cell cytotoxicity

We investigated the effect of this hyperinflammation on memory T cell function. To quantify memory T cell reactivity to viral epitopes in acute MIS-C and after recovery, we used a T cell activation assay based on the rapid induction of CD69, CD154 and CD137 upon antigen-specific stimulation[29] (Fig. 2c,d, Extended Data Fig. 7a,b and Supplementary Data 4). CD4[+] and CD8[+] T cells from acute MIS-C co-cultured with antigen-presenting cells and viral peptides exhibited impaired function, with reduced CD69 expression compared with follow-up samples (Fig. 2d). Specific reactivation (characterized by expression of CD69 and CD154 or CD137) was impaired during acute MIS-C, ranging from 0.3- to 15-fold (median 2.4-fold, $P = 0.0175$) for CD4[+] T cells and 0.2- to 9.9-fold (median 9.9-fold; $P = 0.0391$ (CD137[+]CD69[+])) for CD8[+] T cells (Fig. 2d). The effect was even more pronounced in absolute antigen-specific T cell counts in the blood of patients with MIS-C (Extended Data Fig. 7b). T cell reactivity was not impaired during acute symptomatic SARS-CoV-2 infection compared with the same paediatric patients after recovery (median 1.4-fold, $P = 0.9609$ for specific reactivation of CD4[+] T cells; and median 0.8-fold, $P = 0.9101$ for specific reactivation of CD8[+] T cells) (Extended Data Fig. 7c,d). Given the strong TGFβ instruction of T cells in MIS-C (Fig. 2a), we tested whether TGFβ could impair memory T cell reactivation in healthy donor T cells; we found that TGFβ significantly impaired T cell reactivity (Extended Data

Fig. 7e). We then tested whether anti-TGFβ could reverse the impairment of memory T cell reactivation. We reactivated PBMCs from healthy donors using media with sera from patients with MIS-C, with or without anti-TGFβ (Fig. 2e). Anti-TGFβ increased overall reactivation of CD4[+] and CD8[+] memory T cells, with increased frequencies of CD69[+] memory T cells (CD4[+]: median 1.9-fold increase, $P = 0.0023$; CD8[+]: median 2.4-fold increase, $P = 0.0001$) (Fig. 2e). Specific reactivation was enhanced, with increased frequencies of CD154[+]CD69[+]CD4[+] or CD8[+] memory T cells (CD4[+]: median 1.3-fold increase, $P = 0.0144$; CD8[+]: median 8.9-fold increase, $P = 0.0077$) (Fig. 2e). Similar results were obtained with sera from patients with severe COVID-19 (Fig. 2f), suggesting that TGFβ-induced suppression of antigen-specific memory T cell reactivation is a hallmark of both MIS-C and severe COVID-19 (refs. 4,30).

## TCRVβ21.3[+] T cells react to EBV peptides

MIS-C is associated with expansion of CD4[+] and CD8[+] T cells with TCRVβ21.3 (refs. 2,3,12,14) (Fig. 3a,b). However, as previously described[2,12,14], TCRVβ21.3[+] T cell expansion was not universal (76% of CD4[+] and 62% of CD8[+] T cells) when assessed in total MIS-C PBMCs by flow cytometry (Fig. 3c). Surprisingly, this expansion was present in all patients after 1–3 days of treatment with methylprednisolone and intravenous immunoglobulins (IVIG) (Fig. 3c) ($P = 0.0499$ for CD4[+] T cells and $P = 0.0372$ for CD8[+] T cells). TCRVβ21.3[+] T cell expansion peaked at day three of treatment. Similarly, in the few patients measured during the early acute phase, the number of CD8[+] TCRVβ21.3[+] T cells increased after the first few days of treatment and clinical improvement (Extended Data Fig. 7f). Initially, MIS-C T cell expansion was thought to be driven by spike protein acting as a superantigen[12]. However, we observed increased T cell expansion after treatment (Fig. 3c). Additionally, limited reaction of TCRVβ21.3[+] T cells to SARS-CoV-2 peptides[31], and SARS-CoV-2 spike protein did not exhibit superantigen-like activity[32]. This suggests a specific function of these T cells. To identify *TRBV11-2* (encoding TCRVβ21.3)-expressing T cells, we analysed our scRNA-seq dataset. Most *TRBV11-2*-positive T cells mapped to proliferating T cell cluster 8 and activated cytotoxic T cell clusters 2 and 13 (Fig. 3a), which contained both CD8a[+] and CD4[+] T cells (Fig. 3b). Independent of treatment, we observed *TRBV11-2*[+] T cell expansion in all patients with MIS-C (Fig. 3d). This highlights CD38[+]HLA-DR[hi] T cells as the key disease-driving population and suggests that this type of high-resolution analysis could serve as a diagnostic test for MIS-C.

We validated these findings in independent cohorts from Europe (France and Italy) and South America (Chile) using flow cytometry (Fig. 3e). In all patients with MIS-C, TCRVβ21.3[+] cell frequencies in activated CD38[+]HLA-DR[hi] T cells were higher than in all T cells (median difference +44.5 percentage points (Europe); +16.2 percentage points South America; combined +30.9 percentage points; $P = 3.81 \times 10^{-6}$ (Europe) and $P = 0.0002$ (South America)) (Fig. 3e). Five patients had normal-range TCRVβ21.3[+] T cell frequencies in all T cells, but in four of these patients' TCRVβ21.3[+] T cell frequencies were above-normal frequencies in activated (CD38[+], HLA-DR[hi]) T cells.

We tested whether all or a subset of *TRBV11-2*[+] T cells were expanded in MIS-C. We analysed TRAV genes (which encode the TCRα chain variable region) associated with *TRBV11-2* and identified a subset of full TCRs that were unique to patients with MIS-C and absent in paediatric controls (Fig. 3f and Extended Data Fig. 7h). This suggests antigen-driven expansion of a specific *TRBV11-2*[+] T cell subset. We analysed HLA haplotypes of our Berlin cohort and a previously published American cohort[12], and found that no single haplotype or subset of haplotypes associated with MIS-C in the combined data. We identified a predominance of *HLA-A*02* (28%, $P \geq 0.9999$), *HLA-B*35* (24%, $P = 0.1056$) and *HLA-C*04* (27%, $P = 0.7633$) that was also prominent in controls (Extended Data Fig. 7i–k). We observed a significant predominance of HLA-class-II haplotypes in MIS-C: *HLA-DRB1*01* (0% in controls versus 19.4% in MIS-C,

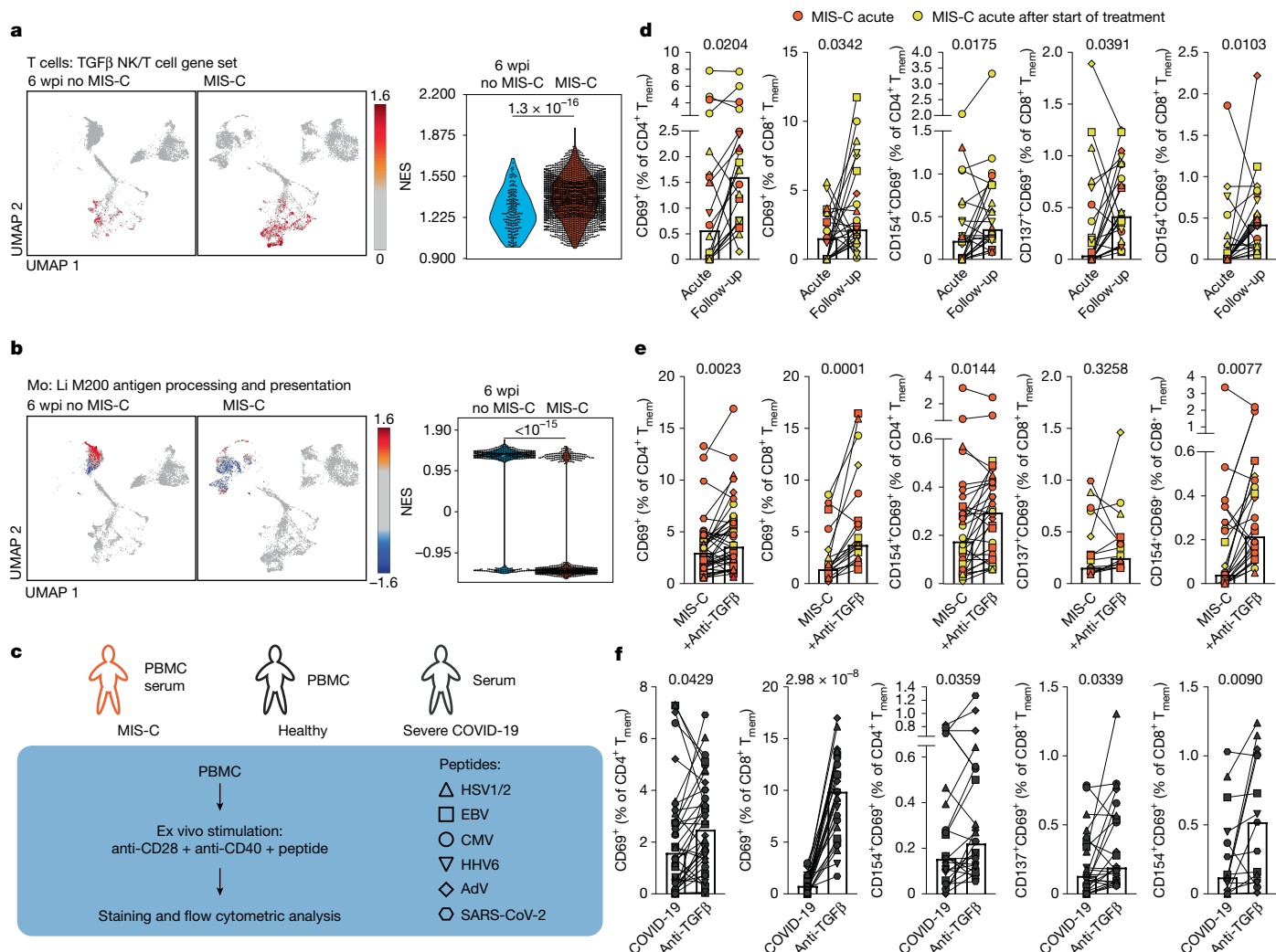

**Fig. 2 | Impaired T cell reactivity during acute phase of MIS-C is induced by TGFβ. a,b**, GSEA using a previously defined TGFβ gene set[4] applied to T cells (**a**) and the gene set 'Li M200 antigen processing and presentation'[59] applied to monocyte clusters (**b**) depicted as both UMAP (left) and a dot plot (right). **c**, Schematic overview of T cell reactivation assays. **d**, Frequencies of overall activated (CD69+) and antigen-specific reactivated (CD137+CD69+ and CD154+CD69+) CD4+ or CD8+ memory T cells (T$_{mem}$, CD45RO+) from patients with MIS-C during the acute phase and at follow-up after symptoms resolved ($n$ = 8 patients and $n$ = 5 different viral peptides). **e,f**, Frequencies of overall activated and antigen-specific reactivated cells of CD4+ and CD8+ memory T cells (T$_{mem}$; CD45RO+) from healthy donors ($n$ = 6) treated with serum from patients with MIS-C (**e**; $n$ = 7) or patients with severe COVID-19 (**f**; $n$ = 5) with or without anti-TGFβ. Samples that were obtained more than 24 h after the start of treatment are colour-coded in yellow. Unpaired (**a,b**) or paired (**d–f**) two-tailed Mann–Whitney $U$-tests.

$P$ = 0.0313) and *HLA-DQB1\*05* (0% in controls versus 23% in MIS-C, $P$ = 0.0027) (Extended Data Fig. 7i–k and Supplementary Data 5 and 6). Both haplotypes are present in only a fraction of patients with MIS-C. In the autoimmune disease ankylosing spondylitis, HLA haplotypes (*HLA-B\*27*) combined with specific TCR repertoire (*TRBV9*) are linked to inflammation and specific peptides[33]. A link between *HLA-B\*27* and CD8 TCR clonotypes that recognize EBV and CMV has been established[34]. However, antigen presentation in ankylosing spondylitis[35] and misfolding or mistrafficking of HLA-B\*27 protein may also induce autoimmunity and inflammation[36].

Herpesvirus reactivation occurs in up to 82% of patients with severe COVID-19 (ref. 37). EBV, cytomegalovirus (CMV) and human herpesvirus 6 (HHV-6) reactivation is common in post-COVID-19 sequelae[38]. One hypothesized mechanism of long COVID is immune dysregulation and viral reactivation; other explanations include microbiota changes, autoimmune priming, microvascular alterations and dysfunctional neurological signalling[38]. Immune profiling in long COVID shows TGFβ upregulation and distinct EBV epitopes[39].

A case series of unexplained hepatitis in children during the SARS-CoV-2 Omicron outbreak was linked to a specific adenovirus (AdV) strain[40]. Affected children showed high prevalence of previous SARS-CoV-2 infection[40]. We thus explored whether simultaneous viral reactivation in MIS-C might explain the systemic hyperinflammation. We compared the expanded TCR repertoire of MIS-C and paediatric controls (6 wpi no MIS-C and patients hospitalized owing to influenza infection) using a virus-specific TCR atlas focusing on EBV and CMV. We isolated TCRs that recognize EBV, CMV, AdV or SARS-CoV-2 TCRs from healthy donors using the antigen-reactive T cell enrichment (ARTE) assay[29] (Fig. 4a–d and Extended Data Fig. 8a,b), generated TCR libraries and combined the data with our previously published SARS-CoV-2 and measles-specific TCR repertoire dataset[30]. *TRBV11-2*+ T cells were abundant in cytotoxic T cells (*ICOS*low*PRF1*hi*GZMB*hi*LAMP1*hi; Fig. 4c,d). Frequencies of specific TCRα chains with *TRBV11-2* were clustered in an unbiased manner (Fig. 4a). This resulted in clustering of all MIS-C TCR repertoires with EBV-specific TCR repertoires from four out of five donors, whereas controls clustered separately (Fig. 4e), suggesting

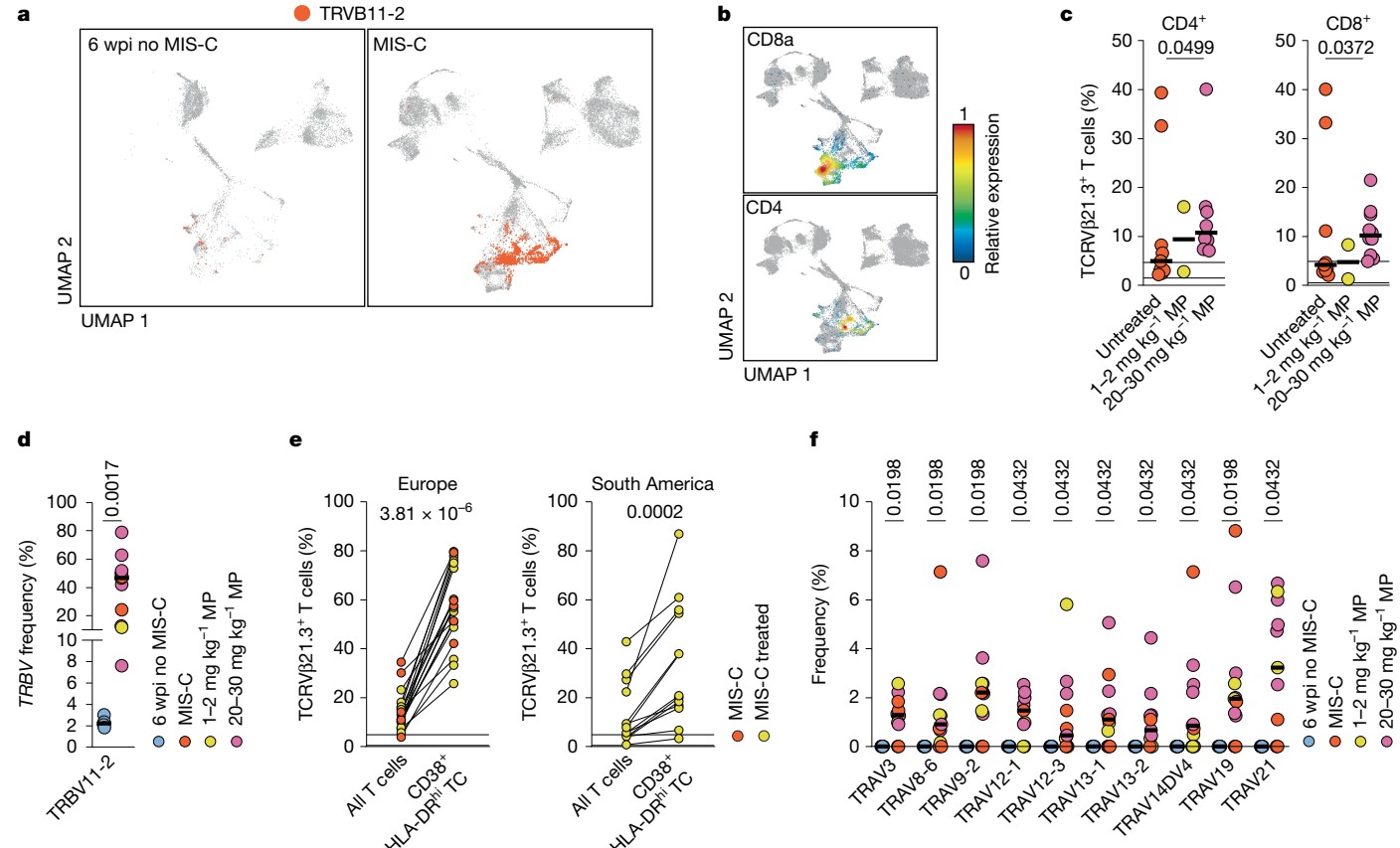

**Fig. 3 | Expansion of a specific subset of *TRBV11-2*⁺ T cells occurs during MIS-C. a**, *TRBV11-2*⁺ T cells superimposed on UMAP of enriched activated cells of paediatric controls at 6 wpi and patients with MIS-C (Fig. 1d). **b**, Expression of CD8a and CD4 genes (CD4 expression only on T cells is depicted for better overview) superimposed on the UMAP from Fig. 1d. **c**, Flow cytometric analysis of TCRVβ21.3⁺ T cells among CD4⁺ or CD8⁺ T cells from patients with MIS-C before start of treatment (untreated, *n* = 9), after 1–3 days of 1–2 mg kg⁻¹ methylprednisolone (*n* = 2) or after 1–3 days of 20–30 mg kg⁻¹ methylprednisolone (*n* = 10). Contingencies of TCRVβ21.3⁺ T cell expansion above the laboratory determined upper cut-off were determined by two-tailed Fisher's exact test. **d**, Frequency of *TRBV11-2*⁺ T cells among total T cells for

paediatric controls at 6 wpi and patients with MIS-C during the acute phase (**d**; Berlin cohort) and validation of single-cell TCR sequencing data (Berlin cohort) using protein quantification by flow cytometry in two independent cohorts from Europe (France and Italy; *n* = 19) and South America (Chile; *n* = 13). Gating strategy in Extended Data Fig. 7l,m. Frequencies of TCRVβ21.3⁺ cells gated on all CD3⁺ cells were compared to frequencies of TCRVβ21.3⁺ cells gated on CD3⁺CD38⁺HLA-DR⁺ cells. **f**, Distribution of indicated TRAV genes associated with *TRBV11-2* (*n* = 4 controls; *n* = 11 patients with MIS-C). Treatment with methylprednisolone is colour-coded as in **d**; MIS-C indicates sampling before start of treatment. Two-sided Mann–Whitney *U*-tests (**d**,**f**); two-sided Wilcoxon matched-pairs signed-rank tests (**e**).

that the expanded MIS-C *TRBV11-2*⁺ T cells could be EBV-specific. It was previously shown that an EBV-derived peptide (EBNA2 residues 276–295 (EBNA2₂₇₆₋₂₉₅)) containing a highly promiscuous epitope (TVFYNIPPMPL (EBNA2₂₇₉₋₂₈₉)) can be linked to TCRVβ21.3 repertoire bias[41,42]. EBNA2 is a major EBV-encoded oncogene that is expressed during EBV lytic reactivation and in newly infected, transformed B cells[43]. We therefore stimulated T cells with autologous antigen-presenting cells, using the promiscuous EBNA2₂₇₉₋₂₈₉ epitope as well as a longer peptide containing this region using the ARTE assay to determine whether TCRVβ21.3⁺ T cells were enriched among peptide-specific memory T cells. We observed consistent enrichment of TCRVβ21.3⁺ T cells among EBNA2-peptide-specific CD154⁺CD69⁺ memory T cells, compared with the entire memory T cell pool (Fig. 4f) in all donors tested (median fold change of enrichment: for CD4⁺ memory T cells with EBNA2₂₇₅₋₂₉₄, 2.6-fold, *P* = 1.77 × 10⁻⁵; for CD4⁺ memory T cells with EBNA2₂₇₉₋₂₈₉, 2.3-fold, *P* = 5.96 × 10⁻⁵; for CD8⁺ memory T cells with EBNA2₂₇₅₋₂₉₄, 3.1-fold, *P* = 1.20 × 10⁻⁵; and for CD8⁺ memory T cells with EBNA2₂₇₉₋₂₈₉, 2.7-fold, *P* = 1.17 × 10⁻⁵). This enrichment in CD4⁺ and CD8⁺ memory T cells highlights the robust association between EBV-derived peptides and TCRVβ21.3⁺ T cells. These findings support an EBV-directed immunological response linked to a specific EBNA2 peptide in MIS-C.

## MIS-C specific T cells target EBV⁺ cells

To further test this hypothesis, we analysed whether *TRBV11-2*⁺ T cells control EBV reactivation or the expansion of latently infected B cells. We isolated and enriched TCRVβ21.3⁺ CD8 and CD4 T cells from healthy donors (Extended Data Fig. 8d), as some CD4⁺ T cells clustered in the cytotoxic T cell cluster (Fig. 3a,b). As a control, we used the T cells depleted of TCRVβ21.3. After polyclonal expansion, T cells were co-cultured with autologous EBV-transformed B cells (lymphoblastic cell line (LCL)), which express EBNA2. TCRVβ21.3⁺CD8⁺ T cells showed an increased killing capacity (16.5 median percentage points increase, *P* = 0.0156) of autologous EBV-infected B cells (Fig. 5a,b and Extended Data Fig. 8e). A 2.5-fold increase in CD107a⁺ T cells (*P* = 0.0313; Fig. 5b and Extended Data Fig. 8f) indicated specific killing of infected B cells via degranulation[44]. TCRVβ21.3⁺CD4⁺ T cells were cultured in a ratio of 30 T cells to 1 LCL for 24 h. As with CD8⁺ T cells, TCRVβ21.3⁺CD4⁺ T cells had a median enhanced killing capability of 14.5 percentage points for LCL (*P* = 0.0273) with increased CD107a expression on T cells (Fig. 5c). These results suggest an expansion of EBV-specific cytotoxic T cells with impaired cytotoxicity and killing capacity in patients with MIS-C.

Despite being fully armed, these T cells seem to be incapable of efficiently attaching to and killing EBV-infected cells, similar to

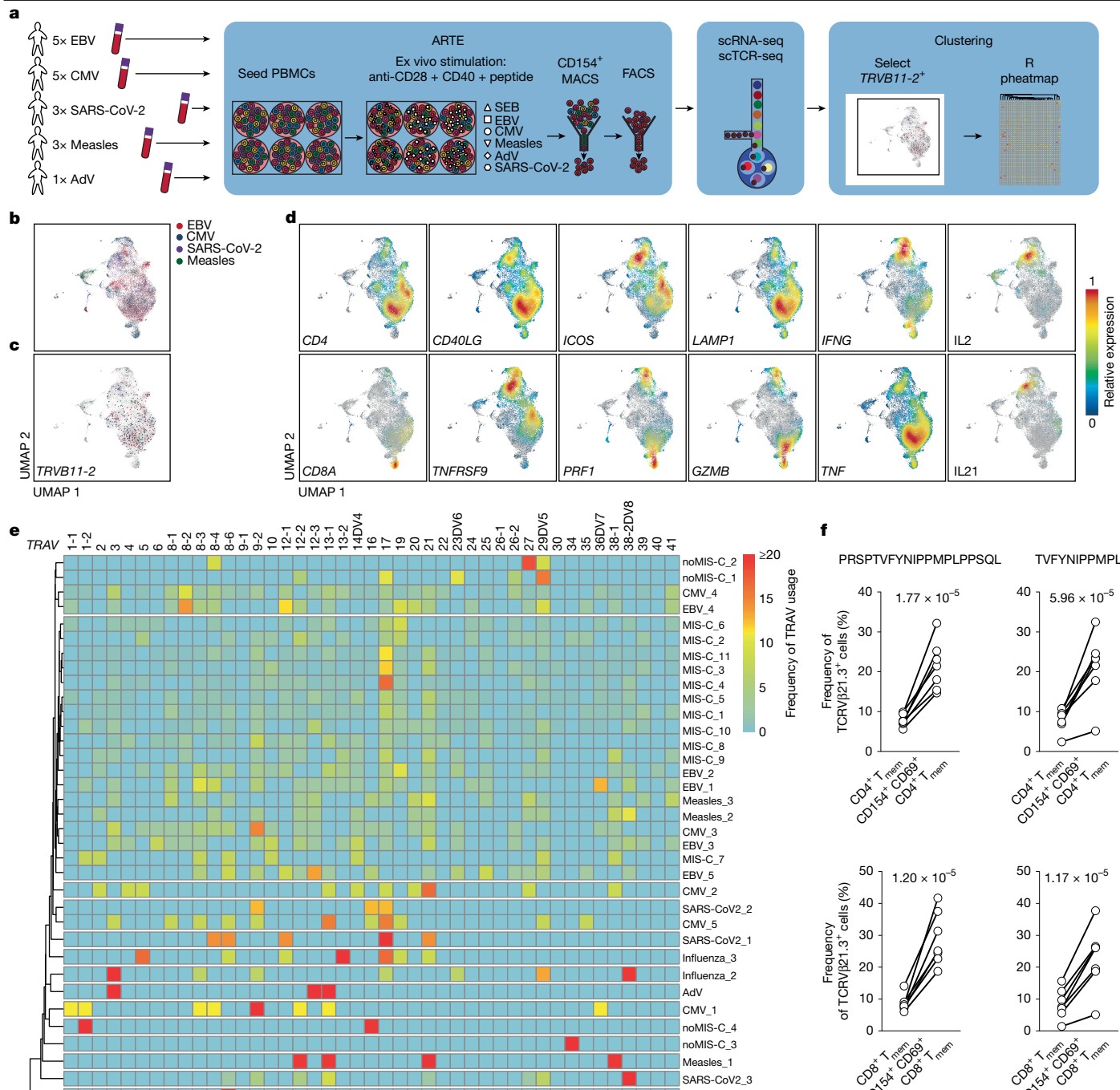

**Fig. 4 | TCR repertoires of expanded T cells in MIS-C show overlap with EBV-specific TCR repertoires. a**, Schematic showing generation of virus-specific TCR libraries and comparison of virus-specific TCRs with MIS-C-specific TCRs. scTCR-seq, single-cell TCR sequencing. **b**, UMAP of 22,344 virus-specific T cells from donors restimulated with EBV (*n* = 5), CMV (*n* = 5), SARS-CoV-2 (*n* = 3) or measles (*n* = 3) peptides, representing 18,010 sequenced TCRβ chains and 15,496 full TCRs. AdV-specific T cells were TCR-sequenced. Virus-specificities are colour-coded. **c**, *TRVB11-2*+ T cells superimposed on the UMAP in **b**. **d**, Gene expression superimposed on the UMAP of antigen-specific T cells, showing that most *TRVB11-2*+ T cells have a CD4 or CD8 cytotoxic phenotype (low: *ICOS*; high: *PRF1, GZMB, LAMP1*). **e**, TCR repertoires of EBV (*n* = 5), CMV (*n* = 5),

SARS-CoV-2 (*n* = 3), measles (*n* = 3) and AdV (*n* = 1) virus-specific T cells from healthy donors, analysed by ARTE[29]. Heat map showing distribution of TRAV gene expression associated with *TRBV11-2*-positive T cells in virus-specific and MIS-C T cells (*n* = 11) T cells, compared with 6 wpi no MIS-C (*n* = 4) and paediatric influenza (*n* = 3) T cells. Unsupervised clustering was performed with the R package pheatmap. **f**, TCRVβ21.3 expression on memory T ($T_{mem}$) cells after stimulation with EBNA2$_{275-294}$ (left) or EBNA2$_{279-289}$ (right) peptides, analysed by ARTE. Frequencies of TCRVβ21.3+ in all CD4+ (top) and CD8+ (bottom) memory T cells and those with antigen-specific reactivation (CD154+CD69+) from *n* = 7 donors. Flow cytometry gating is shown in Extended Data Fig. 8c. Two-sided paired *t*-test.

TGFβ-instructed NK cells that do not attach to SARS-CoV-2-infected cells[4]. TGFβ strongly impairs T cell lysis of EBV-infected B cells[45]. Furthermore, TGFβ overproduction is an immune evasion mechanism in EBV-associated malignancies. Making EBV-specific T cells resistant

to TGFβ enhanced anti-tumour immunity in humanized mouse models and patients with EBV-associated lymphomas[45,46]. As viral infections can trigger EBV reactivation, we tested whether TGFβ causes prolonged reactivation. In line with data showing that TGFβ triggers

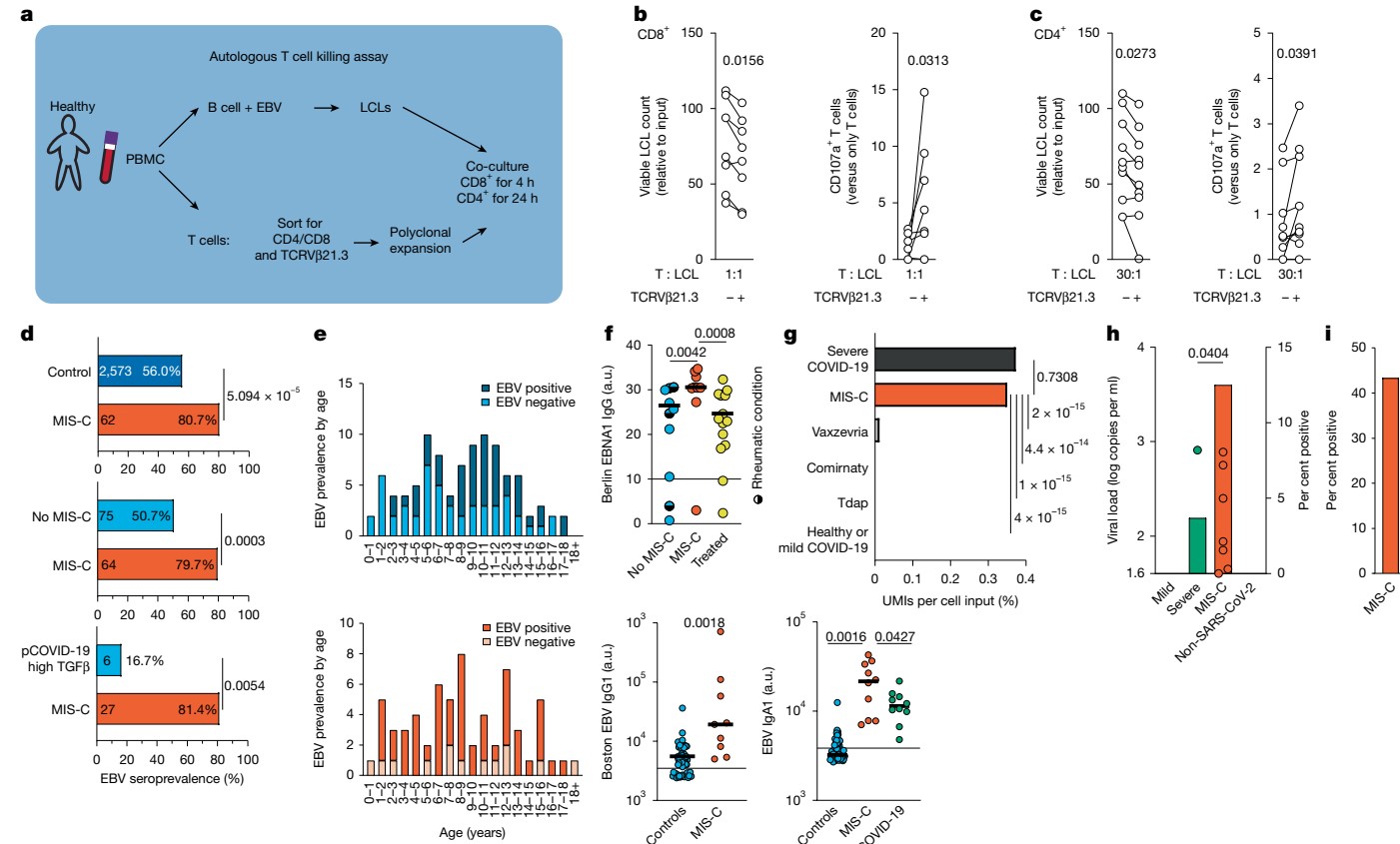

**Fig. 5 | EBV reactivation is prominent in MIS-C. a**, Setup of autologous T cell killing assay. **b,c**, CD4⁺ and CD8⁺ T cells (*n* = 4 donors, 2 time points each) with enriched or depleted TCRVβ21.3⁺ cells were expanded. EBV-transfected LCLs were generated. T cells and LCLs were co-cultured. **b**, Viable LCL counts and CD107a expression measured by flow cytometry for CD8⁺ T cells grown in a 1:1 ratio with LCLs for 4 h (**b**) and CD4⁺ T cells grown in a 30:1 ratio with LCLs for 24 h (**c**). Normalized to LCL incubated without T cells. **d**, EBV seroprevalence from patients with MIS-C and age-matched controls (no MIS-C), paediatric patients with COVID-19 with high TGFβ (pCOVID-19 high TGFβ) and age-matched healthy controls (control). Group sizes are indicated on bars. **e**, EBV seroprevalence by age in patients with MIS-C and healthy controls. **f**, EBV antibody titres in MIS-C versus local paediatric control groups (Berlin: no MIS-C, *n* = 10; MIS-C, *n* = 9; MIS-C treated, *n* = 15; Boston[49]: pre-pandemic healthy controls, *n* = 57; MIS-C, *n* = 9; age-matched COVID-19, *n* = 9). Titres depicted with assay-specific cut-offs. **g**, Unmapped reads from activated B cell and plasmablast datasets (healthy or mild COVID-19 (ref. 30), *n* = 6; Tdap, *n* = 7; Comirnaty, *n* = 15; Vaxzevria[57], *n* = 8; MIS-C, *n* = 32; severe COVID-19 (ref. 30), *n* = 12; same cells sorted) tested against the EBV genome. EBV-specific UMI counts were compared (healthy or mild COVID-19, *n* = 12,716 cells, 0 EBV UMIs; Tdap, *n* = 32,677 cells, 0 EBV UMIs; Comirnaty, *n* = 11,600 cells, 0 EBV UMIs; Vaxzevria, *n* = 16,445 cells, 2 EBV UMIs; MIS-C, *n* = 16,849 cells, 59 EBV UMIs; severe COVID-19, *n* = 96,703 cells, 361 EBV UMIs). **h**, Viral load in 100 µl of cell-free plasma from children with mild or asymptomatic COVID-19 (mild, *n* = 14), severe COVID-19 (severe, *n* = 27), MIS-C (*n* = 56) or other severe viral infections (non-SARS-CoV-2, *n* = 9), measured by quantitative PCR. **i**, EBV DNA detection in whole blood from patients with MIS-C. Two-tailed Wilcoxon signed-rank test (**b,c**); one-tailed Fisher's exact test (**d**); two-tailed Mann–Whitney *U*-test or ANOVA followed by Welch's *t*-tests (**f**); two-tailed Fisher's exact test (**g**); Kruskal–Wallis test followed by Dunn's multiple comparison test, comparing all groups to MIS-C (**h**).

EBV reactivation[47], TGFβ1 induced the lytic cycle (quantified via expression of the EBV immediate early lytic cycle transcription factor *BZFL1*, which triggers lytic reactivation[48]) in LCLs (Extended Data Fig. 8g). In LCLs treated with MIS-C or in sera from patients with severe COVID-19, neutralizing TGFβ significantly reduced lytic cycle induction. TGFβ thus suppresses T cell cytotoxicity and facilitates EBV reactivation.

## MIS-C shows high EBV seropositivity

This led us to assess prevalence of antibodies against EBV and other common latent viruses in our combined MIS-C cohort. As all patients were treated with IVIG consisting of highly purified human IgG, only samples collected before initiation of IVIG treatment were included to complement routinely tested serology (Fig. 5d,e, Extended Data Fig. 8h–k and Supplementary Data 7–12). We tested EBV prevalence in children with MIS-C from three continents and four countries (Berlin, Germany; Lyon, France; Ankara, Turkey and Boston, USA[49]). Patients with MIS-C exhibited significantly higher EBV seroprevalence than age-matched healthy children[50] (80.7% versus 56.0%; *P* = 5.094 × 10⁻⁵).

In this multi-centre cohort, 79.7% of patients with MIS-C were seropositive compared with 50.7% (*P* = 0.0003) of age-matched children at risk for MIS-C after SARS-CoV-2 infection. Children with TGFβ1 serum levels above 250 pg ml⁻¹ during acute SARS-CoV-2 infection had lower seroprevalence than age-matched patients with MIS-C (16.7% versus 81.4% in MIS-C; *P* = 0.0054) (Fig. 5d and Supplementary Data 7). A strong correlation exists between multiple sclerosis and EBV seroprevalence[51]. EBV seroprevalence in paediatric multiple sclerosis[52] was similar to that in our MIS-C cohorts (paediatric multiple sclerosis: up to 90% seropositive, median age 14.1 years; MIS-C: 79.9% MIS-C, mean age 8.7 years).

EBV antibody titres were higher in patients with MIS-C compared with controls and treated patients (MIS-C median 30.6 arbitrary units (a.u.), controls median 26.4 a.u., treated median 24.6; *P* = 0.0042 (MIS-C versus controls) and 0.0008 (MIS-C versus treated)) (Fig. 5f). This was validated in the Boston cohort (MIS-C median a.u. 19,300 versus pre-pandemic controls median a.u. 5,593, *P* = 0.0018) (Fig. 5f), indicating an activated anti-EBV-immune response. For anti-AdV IgG, anti-pertussis IgG1 and anti-RSV IgG1, we detected no difference between patients with MIS-C and controls (Extended Data Fig. 8l).

Furthermore, children with MIS-C showed highest anti-EBV-IgA1 antibody levels (Fig. 5f), supporting a role for TGFβ in regulating the EBV immune response in MIS-C, as TGFβ distinctly targets class switching to IgA[30]. Virus-specific IgM detection revealed high latent virus reactivation at the serological level (Supplementary Data 8–10). EBV and CMV seroprevalences, but not HHV-6 or HSV-1, were markedly increased compared with age-matched controls (Fig. 5d,e and Extended Data Fig. 8h–k). No significant age differences were observed within age-matched subgroups (Extended Data Fig. 8m). More than 86% (59 out of 68) of MIS-C children were seropositive for EBV, CMV or both, potentially influencing clinical manifestations.

We next tested EBV reactivation, which is known to increase mortality in sepsis[53], for its role in MIS-C severity. The most sensitive test is EBV DNA detection in saliva[54]. However, we did not collect saliva samples from children during acute MIS-C. Thus, we tested for EBV reactivation and latent EBV-infected B cell expansion using single-cell transcriptomes of plasmablasts and B cells, as others have done for SARS-CoV-2 (ref. 55) or HHV-6 (ref. 56), and analysed cell-free plasma and whole blood by PCR. These methods are highly specific, but the sensitivity depends on the amount of input material. We sequenced 16,849 B cells and plasmablasts from patients with active MIS-C through a multi-centre effort. As controls, we analysed B cells and plasmablasts from previously published datasets[30,57] that include healthy donors, cases with mild COVID-19 (12,716 cells), a protective immune response on day 7 after vaccination with a toxoid vaccine (Tdap vaccine, 32,677 cells) from which one patient had mild COVID-19, an mRNA vaccine (Comirnaty, 11,600 cells), a viral vector vaccine (Vaxzevria, 16,445), and 96,703 cells from adults with severe COVID-19 (Extended Data Fig. 9). We observed multiple EBV mRNAs in MIS-C samples (Supplementary Data 13 and 14) equating to one unique molecular identifier (UMI) per 286 cells, but none in healthy donors, donors with mild COVID-19 or post-vaccination donors on day 7 after vaccination with toxoid vaccine or mRNA vaccine ($P = 4 \times 10^{-15}$, $P < 1 \times 10^{-15}$ or $P = 4.4 \times 10^{-14}$ respectively). In one patient who had received a viral vector vaccine, which is known to induce an overstimulating plasmablast reaction[57], we could detect 2 EBV UMI (overall 1 UMI per 8,223 cells input, $P = 2 \times 10^{-15}$). In adults with severe COVID-19, EBV mRNA levels were similar to those in MIS-C (1 UMI per 268 cells input, $P = 0.7308$) (Fig. 5g). We could detect EBV DNA in cell-free-plasma from patients with MIS-C, but not in individuals with other paediatric viral infections or in children with mild COVID-19, and in only one child with severe COVID-19 (Fig. 5h). We also checked whether the MIS-C cohorts were screened for EBV load by PCR during hospitalization. One patient had EBV and HHV-6 DNA in a myocardial biopsy. In the Berlin and Lyon cohorts, 26% of patients with MIS-C were routinely tested for EBV reactivation or infection by PCR in whole blood. Out of 23 patients who were tested, EBV viraemia was detected by PCR in peripheral blood in 10 patients (Fig. 5i and Supplementary Data 8 and 9). In a subset of six patients from our plasmablast and B cell dataset, we could match cells to patients. In six patients that we analysed, we identified EBV transcripts that could be clearly matched to 4 patients ($P = 0.0072$ versus healthy, $P = 0.0047$ versus Tdap vaccine, $P = 0.0002$ versus Comirnaty and $P = 0.0182$ versus Vaxzevria). Additionally, in MIS-C, we frequently found EBV mRNA that could not be allocated to individual patients. For severe COVID-19 in adults, EBV mRNA was found in 10 out of the 12 analysed patients ($P = 0.7887$ versus MIS-C) (Extended Data Fig. 8n).

Together, these results show that high EBV seroprevalence and increased TGFβ1 serum levels in MIS-C and severe adult COVID-19 increase susceptibility to EBV reactivation, potentially affecting disease severity[53].

## Conclusion

Here we establish a link between increased TGFβ levels in MIS-C, impaired T cell cytotoxicity and reactivation of EBV. The timing of TGFβ induction before hospital admission remains indeterminable within the scope of this study. For ethical and practical reasons, this question may remain unanswered. Even if TGFβ activation occurs in later stages, prior studies have linked EBV reactivation to increased mortality in critically ill patients[53]. This underscores the importance of considering TGFβ as a potential treatment target, particularly if its activity is delayed. As MIS-C is rare, age-matched EBV prevalence was calculated with small sample sizes, but the significant increase is unlikely to be due to sampling bias.

Our data suggest that EBV and TGFβ1 have a fundamental role in MIS-C pathogenesis. Increased TGFβ1 levels impair immune cell interactions and T cell cytotoxicity, and limit T cell surveillance of EBV-infected B cells. These mechanisms may contribute to MIS-C hyperinflammation. Targeted TGFβ blockade may help to manage MIS-C and other post-COVID-19 sequelae. Beyond post-COVID-19 sequelae, therapies such as autologous EBV- or SARS-CoV-2-specific T cells engineered to resist TGFβ or immunosuppression[58] or antivirals that limit EBV replication could help to alleviate virus-induced hyperinflammation.

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

[1]German Rheumatology Research Center, a Leibniz-Institute (DRFZ), Berlin, Germany. [2]Department of Pediatric Respiratory Medicine, Immunology and Critical Care Medicine, Charité – Universitätsmedizin Berlin, Corporate Member of Freie Universität Berlin and Humboldt-Universität zu Berlin, Berlin, Germany. [3]Berlin Institute of Health at Charité-Universitätsmedizin Berlin, Berlin, Germany. [4]Center for Chronically Sick Children, Charité – Universitätsmedizin Berlin, Corporate Member of Freie Universität Berlin and Humboldt-Universität zu Berlin, Berlin, Germany. [5]German Center for Child and Adolescent Health (DZKJ), Berlin, Germany. [6]Institute of Allergology, Charité – Universitätsmedizin Berlin, Corporate Member of Freie Universität Berlin and Humboldt-Universität zu Berlin, Berlin, Germany. [7]Fraunhofer Institute for Translational Medicine and Pharmacology (ITMP), Immunology and Allergology, Berlin, Germany. [8]National Reference Center for Rheumatic, Autoimmune and Systemic Diseases in Children (RAISE), Pediatric Nephrology, Rheumatology, Dermatology Unit, Hôpital Femme Mère Enfant, Hospices Civils de Lyon, Lyon, France. [9]Clinical Investigation Center (CIC 1407), Hospices Civils de Lyon, Bron, France. [10]Pediatric Intensive Care Unit, Hôpital Femme Mère Enfant, Hospices Civils de Lyon, Lyon, France. [11]Joint Research Unit, Hospices Civils de Lyon-bioMérieux, Hospices Civils de Lyon, Lyon Sud Hospital, Pierre-Bénite, France. [12]CIRI—Centre International de Recherche en Infectiologie, Université Claude Bernard Lyon 1, Inserm, U1111, CNRS, UMR5308, ENS Lyon, Université Jean Monnet de Saint-Etienne, Lyon, France. [13]Department of Pediatric Infectious Diseases, Hacettepe University, Ankara, Turkey. [14]Laboratory of Clinical Immunology and Microbiology, Division of Intramural Research, National Institute of Allergy and Infectious Diseases, National Institutes of Health, Bethesda, MD, USA. [15]Faculty of Medicine, Clínica Alemana-Universidad del Desarrollo, Santiago, Chile. [16]Immunology and Rheumatology Unit, Hospital de Niños Dr. Roberto del Río, Santiago, Chile. [17]Department of Translational Medical Sciences, University of Naples Federico II, Naples, Italy. [18]European Laboratory for the Investigation of Food Induced Diseases (ELFID), University of Naples Federico II, Naples, Italy. [19]Hannover Medical School, Institute of Transfusion Medicine and Transplant Engineering, Hannover, Germany. [20]Translational Immunology, Berlin Institute of Health (BIH) and Charité University Medicine, Berlin, Germany. [21]Department of Microbiology and Hygiene, Labor Berlin, Charité-Vivantes, Berlin, Germany. [22]Department of Immunology, Labor Berlin, Charité-Vivantes, Berlin, Germany. [23]Berlin Institute of Health (BIH)-Center for Regenerative Therapies (B-CRT), Charité – Universitätsmedizin Berlin, Corporate Member of Freie Universität Berlin and Humboldt-Universität zu Berlin, Berlin, Germany. [24]Berlin Center for Advanced Therapies, Charité – Universitätsmedizin Berlin, Corporate Member of Freie Universität Berlin and Humboldt-Universität zu Berlin, Berlin, Germany. [25]German Center for Lung Research (DZL), Berlin, Germany. [26]Division of Infectious Disease, Brigham and Women's Hospital and Program in Virology, Harvard Medical School, Boston, MA, USA. [27]Center for Integrated Solutions for Infectious Diseases, Broad Institute of Harvard and MIT, Cambridge, MA, USA. [28]Mucosal Immunology and Biology Research Center, Massachusetts General Hospital, Boston, MA, USA. [29]Department of Pediatrics, Division of Pulmonology, Massachusetts General Hospital, Boston, MA, USA. [30]Harvard Medical School, Boston, MA, USA. [31]Institute of Microbiology, Infectious Diseases and Immunology, Charité – Universitätsmedizin Berlin, Corporate Member of Freie Universität Berlin and Humboldt-Universität zu Berlin, Berlin, Germany. [32]The Picower Institute for Learning and Memory, Department of Brain and Cognitive Sciences, Massachusetts Institute of Technology, Cambridge, MA, USA. [33]A. N. Belozersky Institute of Physico-Chemical Biology, M. V. Lomonosov Moscow State University, Moscow, Russia. [34]Department of Pediatric Rheumatology, Hacettepe University, Ankara, Turkey. [35]Centre International de Recherche en Infectiologie, University of Lyon, Institut National de la Santé et de la Recherche Médicale, U1111, Université Claude Bernard, Lyon 1, Le Centre National de la Recherche Scientifique, Lyon, France. [36]These authors jointly supervised this work: Tilmann Kallinich, Mir-Farzin Mashreghi. [37]Deceased: Marcus Maurer. ✉e-mail: carl-christoph.goetzke@charite.de; tilmann.kallinich@charite.de; mashreghi@drfz.de

## Methods

### Experimental model and study participant details

The sex, age and ethnicity of participants and their clinical manifestation details during acute MIS-C are reported in Extended Data Table 1 and in Supplementary Data 9–11 and 15, and for controls in Supplementary Data 8. Information on the gender of the patients with MIS-C and controls was not collected. The patients and controls were recruited in Berlin (Germany), Lyon (France), Ankara (Turkey), Santiago (Chile) and Napoli (Italy). This study was approved by the local institutional review boards (IRBs) of the Charité (Pa-COVID-19 and EA2/178/22) (Berlin, Germany), the Comité de Protection des Personnes Sud Méditerranée I (Marseille, France) (ID-RCB: 2020-A01102-37) for the French patients and by the ethical committee of Hospices civils de Lyon (Lyon University Hospitals, France) no. 23_5231 for French controls, by the Hacettepe University Ethical Committee (2021/09-45, Ankara, Turkey) for patients with MIS-C from Turkey, by Mass General Brigham IRB 2020P000955 for the patients and controls from Boston, and by the Ethics Committee Federico II in Naples as a collaborative study protocol with the NIH in 2020 (158/2020); and by the ethical committee of Clínica Alemana Universidad del Desarrollo (IRB ID 202098) for the samples from Chile. Written and informed consent was provided by legal representatives of the patients who participated in this study.

### Human participants

Thirty-nine patients were recruited from the Charité Universitätsmedizin Berlin Department of Pediatric Respiratory Medicine, Immunology and Critical Care Medicine, 49 patients were recruited in Lyon and 20 patients were recruited in Ankara, Turkey, 2 patients in Naples, Italy and 14 patients in Santiago, Chile. MIS-C was diagnosed in all patients by a consultant paediatric rheumatologist according to the definitions from World Health Organization[60], Centers for Disease Control and Prevention[61] and Royal College of Paediatrics and Child Health[62]. Blood samples were taken during the acute phase of MIS-C and during follow-up visits in the outpatient clinic. Paediatric controls included in this study had a positive SARS-CoV-2 PCR result six weeks before inclusion. Additionally, samples from 36 healthy children, 57 children with asymptomatic or mild SARS-CoV-2 infection and 39 children with a moderate and 2 with a severe infection with SARS-CoV-2 during acute and follow-up and 11 children 6 wpi with SARS-CoV-2 were used as controls. Disease severity was defined according to the WHO clinical management guideline[60].

### Isolation of PBMCs and serum

Peripheral blood was drawn into EDTA collection tubes and SST tubes as previously described[4]. PBMC were isolated from peripheral blood by Ficoll-Paque PLUS (Cytiva) density gradient centrifugation at room temperature. Cells were either used directly for analysis or stored at −80 °C in heat-inactivated fetal bovine serum (FCS; Corning, 35-079-CV) with 10% v/v dimethylsulfoxide before analysis. Serum samples were stored at −80 °C before analysis.

### Cytokine measurements

A bead-based multiplex cytokine array (Cytokine/Chemokine/Growth Factor 45-Plex Human ProcartaPlex Panel1, ThermoFisher Scientific) was used according to the manufacturer's protocol. For TGFβ1 measurements, TGFβ1 was transformed to the bioactive form using 1 N HCl followed by neutralization with 1.2 N NaOH. A Human TGFβ1 Simplex ProcartaPlex kit (ThermoFisher Scientific) was used as previously described[4]. To normalize data between different runs, 4 to 16 samples were measured repeatedly per run. The average fold change difference was used to normalize data. The assay-specific lower limits of quantification are indicated in the graphs and listed in Supplementary Data 16.

### Simple size-based western blot assay

T cells were isolated from PBMC using the human Pan T Cell Isolation Kit (Miltenyi Biotec) and $1–2 \times 10^6$ cells were seeded per well in TexMACS Medium (Miltenyi Biotec). pLenti-CMV-Blast-DNTGFBR2-HA was a gift from G.-P. Dotto (Addgene plasmid #130888; http://n2t.net/addgene:130888; RRID:Addgene_130888)[63]. Plasmid (1.6 µg) was overexpressed in $2 \times 10^5$ HEK293T cells (not mycoplamsa tested; originally purchased from ATCC, not reauthenticated) using Lipofectamine (Invitrogen) (DNA:lipofectamine ratio 1:3). As a control a non-expressing vector with a blasticidin selection marker was used. Six hours after transfection, cells were incubated with 10 µg ml⁻¹ blasticitidin (ThermoFisher) for 18–48 h for positive selection of transfected cells. T cells and transfected HEK293T cells were serum-starved overnight prior to incubation with 10% v/v serum from patients with MIS-C for 30 min. For T cells as a control, patients' sera were pre-incubated with antibodies directed against TGFβ1, TGFβ2 and TGFβ3 (50 µg ml⁻¹, R&D Systems, MAB1835-SP) for 10 min. Cells were collected after 30 min and lysed using a buffer containing 0.5% v/v NP40 (Sigma), 150 mM NaCl, 50 mM TrisHCl, cOmplete protease inhibitors (Roche) and phosSTOP phosphatase inhibitors (Roche). Protein concentration was quantified by BCA (ThermoScientific) and 3 µg were loaded on a SimpleWestern 12–230 kDa separation plate (Bio-Techne) according to manufacturer's protocol. For detection of SMAD2 or SMAD3 and phosphorylated SMAD2 or SMAD3 following antibodies were used: SMAD2/3 (D7G7) XP rabbit monoclonal antibody 8685 and phospho-SMAD2 (Ser465/467)/SMAD3 (Ser423/425) (D27F4) rabbit monoclonal antibody 8828 (both 1:100; both Cell Signaling). As a loading control rabbit anti-β-Tubulin (NB600-936; 1:100; Novus Biologicals) was used. Anti-Rabbit Secondary HRP Antibody (042-206; Bio-Techne; provided as working stock dilution from manufacturer) was used as a secondary antibody. The samples were acquired on a Wes instrument (Protein Simple; Bio-Techne) and the data were analysed using the Compass for SW software (Bio-Techne; Protein Simple version 3.1.7). Intensity of the signal was used to quantify relative abundance of phosphorylated SMAD2/3 and normalized to total SMAD2/3 signal.

### Cell isolation from PBMC for single-cell sequencing

Frozen PBMC from 11 patients with MIS-C and 4 samples from patients 6 weeks after SARS-CoV-2 who did not develop MIS-C were thawed in RPMI 1640 (Gibco 61870-044) with 20% v/v FCS (FCS; Corning, 35-079-CV). Cells were washed once in PBS (Th. Geyer) containing 1% w/v BSA (PAN Biotech P06-1391500), 2 mM EDTA (Invitrogen) and 2 µg ml⁻¹ Actinomycin D (Sigma-Aldrich). Cells were incubated with human FcR-blocking reagent (1:50; Miltenyi Biotec) and stained using 1 of 4 TotalSeq anti-human Hashtags (C0251 C0252, C0253 or C0254) (1:250, clone LNH-94 + 2M2, BioLegend), CD3 FITC (1:100, clone UCHT1, in house), CD14 PerCP-Cy5.5 (1:50, clone QA18A22, BioLgend), CD38 APC (1:25, clone HIT2, Biolegend), HLA-DR APC-Vio770 (1:100, clone AC122, Miltenyi Biotec), CD19 V500 (1:200, clone HIB19, BD Biosciences) and CD27 PE (1:100, clone M-T271, BioLegend) at 4 °C for 30 min. Directly prior to sorting on a BD AriaII (BD Biosciences), DAPI (Sigma-Aldrich, 0.1 µg ml⁻¹) was added.

Activated T cells were identified as DAPI⁻CD14⁻CD19⁻CD3⁺CD38⁺HLA-DR^hi, memory B cells were identified as DAPI⁻CD14⁻CD3⁻CD19⁺CD27⁺ (and CD38^hi for plasmablasts), and monocytes were identified as DAPI⁻CD14⁺ (Extended Data Fig. 2b).

For identifying methylprednisolone effects on activated cells and for activated cells from influenza infected patients, cells were sorted on a MA900 Multi-Application Cell Sorter (Sony Biotechnology). Instead of plasmablasts, lineage⁻HLA-DR^hi cells were sorted (identified as DAPI⁻CD14⁻CD19⁻CD3⁻HLA-DR^hi) (Extended Data Figs. 3b and 5a). For sorting of B cells and plasmablasts and analysing TCRVβ21.3 frequencies on T cells, frozen cells were prepared for sorting as mentioned above and stained with CD3 FITC (1:100, clone UCHT1, in

house), CD27 PE (1:100, clone M-T271, BioLegend), HLA-DR PerCP-Cy5.5 (1:50, clone L243, BioLegend), CD21 PE-Vio770 (1:75, clone HB5, Miltenyi Biotec), CD14 VioBlue (1:200, clone TÜK4, Miltenyi Biotec) or CD14 AlexaFluor700 (1:500, clone TM1, in house), CD19 BV510 (1:25, clone HIB19, BioLegend), CD38 APC (1:25, clone HIT2, Biolegend), TCRVβ21.3 APC-Vio770 (1:100, clone REA894, Miltenyi Biotec). Directly prior to sorting DAPI (Sigma-Aldrich, 0.1 µg ml$^{-1}$) was added. Cells were sorted on a MA900 Multi-Application Cell Sorter (Sony Biotechnology) or a Cytek Aurora (Cytek Biosciences). B cells were identified as DAPI$^-$CD14$^-$CD3$^-$ CD19$^+$CD21$^+$ and plasmablasts as DAPI$^-$CD14$^-$CD3$^-$CD19$^+$CD27$^+$CD38$^{hi}$ (Extended Data Fig. 9a). T cells were identified as DAPI$^-$CD14$^-$CD19$^-$CD3$^+$, and activated T cells were identified as DAPI$^-$CD14$^-$CD19$^-$CD3$^+$CD38$^+$ HLA-DR$^{hi}$ (Extended Data Fig. 7l,m), Sorted cells were counted using a MACSQuant Analyzer 16 (Miltenyi Biotec) and processed for scRNA-seq using 10X Genomics technology.

## Single-cell RNA library preparation and sequencing
Single-cell RNA library construction and sequencing was done as previously described[4,30]. In brief, Chromium Next GEM Single Cell 5′ reagent kits v2 (dual index) with feature barcode technology for cell surface protein (CITE) mapping (10X Genomics) were used according to the manufacturer's protocol. Final CITE-seq libraries were generated after index PCR with dual Index Kit TN Set A (10X Genomics) and final GEX, TCR and B cell receptor (BCR) libraries were generated after fragmentation, adapter ligation and final index PCR with a dual Index Kit TT Set A (10X Genomics). Libraries were quantified using a Qubit HS DNA assay kit (Life Technologies) and fragment sizes were determined using a HS NGS Fragment (1–6,000 bp) kit (Agilent).

Sequencing was performed on a NextSeq2000 sequencer (Illumina) using a P3 reagent cartridge (100 cycles) (Illumina) with the following recommended sequencing conditions: read 1: 26 nt, index 1: 10 nt, read 2: 90 nt, index 2: 10 nt.

## Single-cell transcriptome analysis
Raw sequence reads were processed using cellranger (version 5.0.0), including the default detection of intact cells based on the Empty-Drops method[64]. Demultiplexing, mapping, detection of intact cells, as well as quantification of gene expression was performed using cellranger's count pipeline in default parameter settings with refdata-cellranger-hg19-1.2.0 as reference, Hashtag 1–4 as feature reference and expected number of 3,000 cells per sample. Of note, the used reference does not contain immunoglobulin genes and TCR genes as defined by the respective biotype, except for *TRBV11-2* which was removed in addition. This led to 8,531, 4,356 and 7,644 intact cells for MIS-C pools and 8,060 for controls. Next, cellranger's aggr was used to merge the libraries without size normalization and further analysed in R (version 4.1.2) using the Seurat package (version 4.0.5)[65]. In particular, the transcriptome profiles were read using Read10x and CreateSeurat-Object and log-normalized using NormalizeData. A UMAP was computed using ScaleData, RunPCA to compute 50 principal components and RunUMAP using 1:50 dimensions. Transcriptionally similar clusters were identified by shared nearest neighbour modularity optimization using FindNeighbors with pca as reduction and 1:50 dimensions as well as FindClusters with resolutions ranging from 0.1 to 1.0 in 0.1 increments using FindCluster. Clustering with a resolution of 0.6 (Extended Data Fig. 2c) was judged to best reflect the transcriptional community structure, by visually inspecting the percentage of mitochondrial genes, UMI counts, number of identified genes, housekeeping genes, TCRs and BCRs, as well as expression of typical marker genes projected on the UMAP (Extended Data Fig. 2d–i). Clusters were annotated based on the expression of core marker genes (Extended Data Fig. 2f–i), top differentially expressed genes (Supplementary Data 1, 2 and 17), cell cycle scores, derived from the scaled expression of genes associated with different cell cycle stages using the Loupe Browser (version 5 and 6, 10X Genomics). To further refine cell identification, the identification of

isotypes for B cells by BCR sequencing was used to distinguish between pre-switched and switched memory B cells (Extended Data Fig. 2i). TCR sequencing was additionally used to identify T cells. Clusters comprising low quality cells as well as clusters comprising contaminations were not considered in further analyses. Samples were demultiplexed using hashtag reads after log-normalization and manual distinction of positively tagged cells using histograms.

## Single-cell immune profiling
TCR annotations were based on single-cell immune profiling. In particular, raw sequence reads were processed by cellranger (version 5.0.0) with vdj in default parameter settings for demultiplexing and assembly of the TCR sequences using refdata-cellranger-vdj-GRCh38-alts-ensembl-2.0.0 as reference. Annotations of the TCR were assigned to the corresponding cells in the single-cell transcriptome analysis by identical cellular barcodes. In case of multiple contigs, the most abundant, productive and fully sequenced contig for the α and β or γ and δ chain was considered, respectively. Noteworthy, cellranger mislabelled *TRBV11-2* with *TRBV11-1* annotations as judged by multiple alignment of the sequences of the respective contigs to the references of *TRBV11-1* and *TRBV11-2* as well as mapping of transcriptome to the *TRBV11-2* gene locus.

## Gene set enrichment analysis
GSEA was performed as previously described[57] for each individual cell based on the difference to the mean of log-normalized expression values of monocytes, B cells including plasmablasts, or T cells manually selected using cloupe (version 6.3.0) in the analysed set as pre-ranked list and 1,000 randomizations[66,67]. The GSEA was performed separately for T cells, B cells (with plasmablasts) and monocytes. Permutations are conducted by gene set to calculate NES, FDRs, values and core genes for each cell. The number of significant cells for each set and core genes are represented in Supplementary Data 3. Significant up- or downregulation was defined by a FDR ≤ 0.50 or with an FDR ≤ 0.25 (the recommended FDR for bulk sequencing)[66] as an additional control (Extended Data Fig. 6) and by a normalized *P* value < 0.05. For visualization, NES for significant cells were plotted. The GSEA was performed for indicated cells using Hallmark gene sets, our previously published TGFβ NK-cell gene set[4] and previously defined immune transcription modules[59]. Hallmark gene sets were obtained from the MSigDB Collections[68]. The GSEA results were visualized by projection of the NES scores on UMAPs as well as on violin plots of the NES score of significant enriched cells (positive NES score).

## Ex vivo stimulation of isolated mononuclear cells
Antigen-specific restimulation experiments were performed in RPMI 1640 medium (Gibco), supplemented with 5% v/v human AB serum (Sigma-Aldrich) or 10% v/v patients' sera, 100 U ml$^{-1}$ Penicillin/ Streptomycin (Gibco) and 2 mM L-glutamine (HyClone). For antigen-reactivity measurements, a total of $5 \times 10^5$ PBMCs were stimulated for 16 h with the following antigens. For SARS-CoV-2: PepTivator SARS-CoV-2 Prot_N, PepTivator SARS-CoV-2 Prot_S, PepTivator SARS-CoV-2 Prot_M, Omicron-specific pepTivator SARS-CoV-2 Prot_S B.1.1.529 (all 60 nM per peptide; Miltenyi Biotech). For EBV: PepTivator EBV Consensus, premium grade (60 nmol per peptide; Miltenyi Biotech). For HSV-1: PepTivator HHV1 Envelope Glycoprotein D, research grade (60 nmol per peptide; Miltenyi biotech). For HSV2: PepMix HSV2 (gD) (15 nM per peptide; JPT peptide technology). For HHV-6 (U54) Peptide Pool (15 nM per peptide; peptides&elephants). For AdV: PepTivator AdV Select (60 nM per peptide; Miltenyi Biotech). Other peptides used were *Staphylococcus* enterotoxin B (SEB; Sigma-Aldrich) 1 µg ml$^{-1}$ or peptide pools of cytomegalovirus (CMV) pp65 (60 nM per peptide; Miltenyi Biotech). Stimulation was performed in the presence of 1 µg ml$^{-1}$ CD40 and 1 µg ml$^{-1}$ CD28 functional grade pure antibody. When used, patient sera were pre-incubated with antibodies directed against TGFβ1, TGFβ2 and TGFβ3 (50 µg ml$^{-1}$, R&D Systems, MAB1835-SP) for

10 min. For EBNA2-peptide-specific T cells the EBNA2$_{275-295}$ peptide PRSPTVFYNIPPMPLPPSQL (10 µg ml$^{-1}$) or the EBNA2$_{279-290}$ peptide TVFYNIPPMPL (5 µg ml$^{-1}$) were custom-made (peptides&elephants, purity ≥80%) and at least $2 \times 10^7$ PBMCs were stimulated.

Cells were incubated with FcR-blocking reagent (1:50, Miltenyi Biotec) and stained for flow cytometric analysis with CD14 VioBlue (1:200, clone REA599, Milenyti Biotec), CD19 VioBlue (1:100, clone REA675, Miltenyi Biotec), CD45RO BV510 (1:50, clone UCHL1, BD Biosciences), CD3 PE Cy5 (1:200, clone UCHT1, BioLegend), CD8a BV650 (1:100, clone RPA-T8, BioLegend), CD4 PerCPeFluor710 (1:200, clone SK3, eBioscience), CD69 APC-Cy7 (1:100, clone FN50, BioLegend), CD154 PE (1:100, clone REA238, Miltenyi Biotec) CD137 PE Cy7 (1:100, clone 4B4-1, BioLegend) as mentioned above. Directly prior to analysing on a MACSquant Analyzer 16 (Miltenyi Biotec) using MACSQuantify software, DAPI (Sigma-Aldrich, 0.1 µg ml$^{-1}$) was added. CD69$^+$CD4$^+$ memory T cells were defined as DAPI$^-$CD14$^-$CD19$^-$CD3$^+$CD45RO$^+$CD8$^-$CD4$^+$CD69$^+$. CD69$^+$CD8$^+$ memory T cells were defined as DAPI$^-$CD14$^-$CD19$^-$CD3$^+$CD45RO$^+$CD4$^-$CD8$^+$ and CD69$^+$. CD154$^+$CD69$^+$CD4$^+$ memory T cells were defined as DAPI$^-$CD14$^-$CD19$^-$CD3$^+$CD45RO$^+$CD8$^-$CD4$^+$CD154$^+$ and CD69$^+$CD154$^+$CD69$^+$CD8$^+$ memory T cells were defined as DAPI$^-$CD14$^-$CD19$^-$CD3$^+$CD45RO$^+$CD4$^-$CD8$^+$CD154$^+$CD69$^+$. CD137$^+$CD69$^+$CD8$^+$ memory T cells were defined as DAPI$^-$CD14$^-$CD19$^-$CD3$^+$CD45RO$^+$CD4$^-$CD8$^+$CD137$^+$CD69$^+$ (Extended Data Fig. 7a,c). For EBNA2$_{275-295}$- and EBNA2$_{279-290}$-specific T cells, cells were stained with CD14 VioBlue (1:200, clone REA599, Milenyti Biotec), CD19 VioBlue (1:100, clone REA675, Miltenyi Biotec), CD45RO BV510 (1:50, clone UCHL1, BD Biosciences), CD3 PE Cy5 (1:200, clone UCHT1, BioLegend), CD8a BV650 (1:100, clone RPA-T8, BioLegend), CD4 PerCPeFluor710 (1:200, clone SK3, eBioscience), CD69 APC (1:50, clone FN50, Miltenyi Biotec), CD154 PE (1:100, clone REA238, Miltenyi Biotec) and TCRVβ21.3 APC-Vio770 (1:100, clone REA894, Milteyi Biotec). CD4$^+$ or CD8$^+$ memory T cells and CD154$^+$ CD69$^+$ CD4$^+$ or CD8$^+$memory T cells were analysed for expression of TCRVβ21.3 (Extended Data Fig. 8c) using FlowJo software 10.8.1 (TreeStar).

## HLA haplotyping

DNA was extracted from whole blood or buccal swabs (eSwab, Copan) using the QIAamp DNA Micro kit (QIAGEN), from patients with MIS-C and heathy controls (children without MIS-C and adults used for the generation of virus-specific T cell libraries). The DNA was subjected to HLA genotyping for HLA-A, HLA-B, HLA-C, DRB1, DRB3, DRB4, DRB5, DQA1, DQB1 and DPB1. HLA typing was performed using the Protrans NGS for 7 Loci kit (Protrans Medical Diagnostics) by next generation sequencing. The isolated genomic DNA was used in eight multiplex PCR reactions and labelled with Nextera XT index primer kits (Illumina). After group-specific pooling, followed by CleanPCR Beads purification (Beckman Coulter) and normalization (Quant-iT PicoGreen, Thermo Fischer Scientific), a MiSEQ library was prepared and sequenced on an Illumina MiSeq platform. HiType software (Inno-Train) based on the most recent HLA database was used for data analysis.

## TCR library preparation of virus-specific T cells by ARTE assay

At least $1 \times 10^7$ total PBMCs were stimulated as mentioned above. After stimulation, cells were stained with TotalSeq anti-human Hashtags as previously mentioned, followed by CD154 MACS enrichment according to the manufacturer's protocol (CD154 MicroBead Kit, human; Miltenyi Biotec). After MACS, cells were incubated with human FcR-blocking reagent (1:50, Miltenyi Biotec) and stained for FACS enrichment with PE-anti Biotin (1:100, clone Bio3-18E7, Miltenyi Biotec), CD154 PE (1:40, clone REA238, Miltenyi Biotec), CD14 VioBlue (1:200, clone REA599, Milenyti Biotec), CD19 VioBlue (1:100, clone REA675, Miltenyi Biotec), CD3 PE Cy5 (1:200, clone UCHT1, BioLegend) and CD69 APC (1:100, clone REA824, Miltenyi Biotec). DAPI (Sigma-Aldrich, 0.1 µg ml$^{-1}$) was added and cells were sorted on a MA900 Multi-Application Cell Sorter (Sony Biotechnology) according to scatter properties,

viability (DAPI$^-$) and dump$^-$ (CD14 and CD19) specifically for CD3$^+$CD154$^+$CD69$^+$ antigen-specific cells (Extended Data Fig. 8a). A heat map was generated using the R package pheatmap.

## EBV killing assay

EBV-immortalized B cells (LCL) from healthy donors were generated as previously described[69]. In brief $5-10 \times 10^6$ PBMCs were seeded in RPMI1640 medium (Gibco), supplemented with 10% v/v FCS (Corning or Biowest) and 250 µl EBV supernatant 4 µg ml$^{-1}$ CpG and 0.5 µg ml$^{-1}$ Cyclosporin A and passaged subsequently. From the same donors, T cells were isolated from PBMC using the Human Pan T cell Isolation Kit (Miltenyi Biotec), and incubated with human FcR-blocking reagent (1:50, Miltenyi Biotec) and stained with CD3 FITC (1:100, clone UCHT1, in-house), CD8a BV650 (1:100, clone RPA-T8, BioLegend), CD4 PerCPe-Fluor710 (1:200, clone SK3, eBioscience), TCRVβ21.3 APC-Vio770 (1:100, clone REA894, Miltenyi Biotec). Prior to sorting, DAPI (Sigma-Aldrich, 0.1 µg ml$^{-1}$) was added. Cells were sorted on a MA900 Multi-Application Cell Sorter (Sony Biotechnology). Viable (DAPI$^-$) and VioBlue$^-$CD3$^+$ cells were further sorted according to CD4, CD8 and TCRVβ21.3 expression (Extended Data Fig. 8d). Forty thousand cells were seeded per well and polyclonally expanded using Dynabeads human T-Activator CD3/CD28 (ThermoFisher) at a bead-to-cell ratio of 1:1 to generate enough material for the killing assays. For CD4 cells, 50 U ml$^{-1}$ human recombinant IL-2 (Peprotech) was added and for CD8 expansion, 10 ng ml$^{-1}$ IL-15 (Peprotech) was added to the medium (TexMACS Medium (Miltenyi Biotec) with 5% human AB serum (Sigma-Aldrich) and 100 U ml$^{-1}$ Penicillin/Streptomycin (Gibco)). LCLs were marked with CFSE (1:2000, BioLegend). After expansion, viable, CFSE-marked LCLs and expanded T cells from each donor were counted after staining with DAPI and $1 \times 10^4$ viable LCLs were seeded with respective numbers of viable T cells from the same donor and centrifuged for 2 min at 200 rpm. LCLs were co-cultured for 4 h with CD8$^+$ T cells and 24 h with CD4$^+$ T cells at 37 °C 5% CO$_2$. To detect recent degranulation, a CD107a assay was used. For this assay, CD107a AlexaFluor647 (1:3000, clone H4A3, BioLegend) and Brefeldin A (5 µg ml$^{-1}$, BioLegend) were added for the last 4 h, as CD107a is only briefly expressed on the surface during degranulation[44]. Cells were stained with fixable viability dye ZombieAqua (1:400, BioLegend) or Viobility 405/452 Fixable Dye (1:100, Miltenyi Biotec), CD3 PE Cy5 (1:200, clone UCHT1, BioLegend), CD8a BV650 (1:100, clone RPA-T8, BioLegend), CD4 PerCPeFluor710 (1:200, clone SK3, eBioscience) and TCRVβ21.3 APC-Vio770 (1:100, clone REA894, Miltenyi Biotec). Afterwards, cells were fixated and permeabilized with the True Nuclear staining kit (BioLegend) and stained with active caspase-3 AlexaFluor647 (1:50, BD Bioscience). Viable LCLs were defined as CFSE$^+$, fixable viability dye$^-$ and active caspase-3$^-$ cells (Extended Data Fig. 8e). Cell counts were normalized to LCLs cultured without T cells and CD107a (Extended Data Fig. 8f) staining was normalized to T cells cultured without LCL to subtract background expression of CD107a.

## Antibody quantification

Serum IgG-antibody levels against HSV-1 (Abnova, KA0229), HSV2 (Abnova, KA0231), EBNA1 (abcam, ab108731), CMV (Abnova, KA1452), HHV-6 (Abnova, KA1457) and AdV (Creative Diagnostics, DEIA2382) and serum IgM antibody levels against HSV-1/2 (Abnova, KA4842), EBNA1 (Abnova, KA1449), CMV (Abnova, KA0228), HHV-6 (Creative Diagnostics, DEIABL57) and AdV (Creative Diagnostics, DEIA1767) were measured in first serum samples obtained from patients according to manufacturer's manuals. A provided working stock solution of the antibodies was used without further dilution. Arbitrary units were calculated based on reference controls. For age-matched comparison of seroprevalences, patients were grouped by age according to published age ranges[50,70–72]. Expanded humoral profiling data was obtained from a previously published MIS-C cohort[49]. Published cohorts or control cohorts, including healthy children and children with SARS-CoV-2 infection who did not get MIS-C, were adjusted to match the age distribution

of the test population (Supplementary Data 12). In hospital data for virus serology was determined by accredited Immunoblotting Assays for EBV-IgM and –IgG antibodies and enzyme immunoassays by automatic analysis (Liaison, Diasorin and Architect, Abbott).

## EBV transcript detection in single-cell dataset
For detection of EBV reactivation, unmapped reads from the transcriptome analysis were extracted from the .bam files from the transcriptome analysis and mapped to the human gamma herpesvirus 4 references (NCBI_Assemblies:GCF_002402265.1, GCF_000872045.1.). Specific binding as well as correct gene annotation was verified using blast with standard databases. EBV UMI counts were assigned to the corresponding cells in the single-cell transcriptome analysis by identical cellular barcodes.

## EBV load in cell-free plasma
EBV viral DNA (elution volume 50 µl) was extracted from 200 µl of plasma sample using an EasyMag extractor (bioMérieux, Marcy-l'Etoile, France) following the manufacturer's instructions. The presence and viral load of EBV were then determined using the EBV R-GENE kit (available for research use only, not for diagnostic, 69-002B; bioMérieux). Log of EBV DNA copy number per ml (log copies ml$^{-1}$) plasma are used to describe EBV viral load. The reported LOD was 1.6 log copies ml$^{-1}$.

## EBV reactivation assay
For EBV reactivation, LCLs were incubated in RPMI 1640 (Gibco 61870-044) with 10% v/v FCS (Biowest S1600-500) or 10% v/v patient's serum and 1% v/v Penicillin/Streptomycin (Gibco 15140-122). Recombinant human TGFβ1 (10 ng ml$^{-1}$; PeproTech, 100-21) was used as a positive control. If TGFβ was neutralized, the patient's serum was pre-incubated for 10 min with 50 µg ml$^{-1}$ of antibodies directed against TGFβ1, TGFβ2 and TGFβ3 (R&D Systems, MAB1835-SP). Cells were collected after 24 h and total RNA was extracted using the RNeasy Plus Micro Kit (Qiagen). cDNA was transcribed using the TaqMan reverse transcription kit using random hexamers (Life Technologies). TaqMan PCR was performed using human *HPRT1* primers and probes of TaqMan gene expression assays (Life Technologies) as well as the following combination of primers and probes for the transcription factor *BZLF1*, which induces the lytic EBV replication cycle, forward: 5′-CTCAACCTGGAGACAATTCTAC TGT-3′, reverse: 5′-TGCTAGCTGTTGTCCTTGGTTAG-3′, probe: 5′-FAM-CTG CTGCTGCTGTTTG-3′NFQ (Life Technologies). Samples were measured on a QuantStudio 5 Real-Time PCR System (Applied Biosystems).

## Statistical analysis and reproducibility
GraphPad Prism (v8.00 to v9.4.1) for Windows (GraphPad Software) was used for statistical analysis of the data. If applicable, all data were plotted as individual values and the median was given as data summary. For two-group comparisons of non-normal distributed data, a two-tailed Mann–Whitney *U*-test was used. If normal distribution could be assumed a two-tailed *t*-test was used. For paired samples, the paired-test equivalence was used. For multiple group comparisons, two-tailed non-parametric ANOVA (Kruskal–Wallis test) was used, followed by a Dunn's multiple comparison test with correction for multiple comparisons. Correlations were calculated using a Spearman correlation. Testing for significance in GSEA: if only positive enrichment was quantified, a two-tailed Mann–Whitney *U*-test was used; otherwise, to test for positive and negative enrichment in bimodal data a two-tailed Fisher's exact test was used. Testing for increased seroprevalences of latent virus infections: a one-tailed Fisher's exact test was used. The significance threshold for all tests was set to 0.05 and *P* values are provided with up to 4 significant figures where applicable.

## Reporting summary
Further information on research design is available in the Nature Portfolio Reporting Summary linked to this article.

## Data availability
Next generation sequencing datasets generated and analysed during the current study are available in the Gene Expression Omnibus (GEO) repository under accession GSE254179. Data were mapped using the human genome reference hg19. The published datasets used for GSEA are available in the Molecular Signatures Database (MSigDB) (https://www.gsea-msigdb.org/gsea/msigdb/), Hallmark[73] or Reactome[74], or were extracted from Witkowski et al.[4]. Activated B cell and plasmablast datasets used from previous studies can be found in the respective repositories: GSE253862 (ref. 57) and GSE158038 (ref. 30). Source data and additional supporting data are provided with this paper in Supplementary Data. Additional support for further analysis of the study findings is available upon request from the lead corresponding author or, for bioinformatics, contact P.D. (pawel.durek@drfz.de).

## Code availability
The software used in this study is open source; Cellranger from 10X Genomics (https://support.10xgenomics.com/single-cell-gene-expression/software) and Seurat packages (https://cloud.r-project.org/web/packages/Seurat/index.html). We used publicly accessible and widely used analysis pipelines, algorithms and script packages, which are regularly updated by the respective authors. We advise using the updated scripts provided by the authors' packages. Our used scripts are available upon request from P.D.

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

**Acknowledgements** The authors express their gratitude to the patients and their legal guardians for their consent in participating in this study. The authors thank the staff of Charité Department of Pediatric Respiratory Medicine, Immunology and Critical Care Medicine who took care of the patients; A. Portefaix and the CIC for the support on HPI COVID project; members of the Mashreghi and Kallinich laboratory, the Radbruch laboratory, G. Dannecker and the Charité Department of Pediatric Respiratory Medicine, Immunology and Critical Care Medicine for valuable discussions on the manuscript; L. D. Notarangelo and R. Goldbach-Mansky for setting up cross-continental collaborations; M. Abaandou, H. Kenney, N. Yoon and

S. Weber for their help with organization, documentation, sample storage, data cleanup and data entry; M. McGrath for critical reading of the manuscript and helpful comments; the staff at the DRFZ flow cytometry facility, J. Kirsch and T. Kaiser for their support in cell sorting. We acknowledge the excellent cooperation with the Central Biobank Charité (ZeBanC), which was responsible for storage and processing of biological samples. This work was supported by the following grants: the Federal Ministry of Education and Research (BMBF) with financing of the projects TReAT and CONAN to M.-F.M.; the state of Berlin and the European Regional Development Fund through the grant EFRE 1.8/11 and EFRE PersMedLab to M.-F.M.; the state of Berlin through 'Modulation of the mucosal immune response to prevent severe COVID-19 disease progression through commensal bacteria or vaccination' and 'Systematic decoding of unclear inflammations in children by single-cell sequencing and artificial intelligence'; the Leibniz Association through the Leibniz Collaborative Excellence TargArt project to M.-F.M. and T.K. and the ImpACt project to M.-F.M. and A.K.; the German Center for Child and Adolescent Health (DZKJ) grants 01GL2401A to T.K., M.A.M. and C.C.G. and 01GL2401C to M.-F.M.; the German Research Foundation (CRC 1449, project 431232613 and Clinical Research Unit KFO 5023 'BecauseY' and project 504745852 to A.K.) and the German Federal Ministry of Education and Research (82DZL009B1) to M.A.M. T.K. is supported by a BIH Charité Fellowship, C.C.G. is a fellow of the BIH Charité Clinician Scientist Program and L.E. is a fellow of the BIH Charité Junior Clinician Scientist Program, all funded by the Charité-Universitätsmedizin Berlin and the Berlin Institute of Health. L.E. is supported by a PhD Fellowship from the Research Foundation–Flanders (FWO) (grant 11E0123N). B.E.G. is funded by NIH R01 AI164709 and DE033907. This work was supported by National Institutes of Health/National Heart Lung and Blood Institute (1R01HL173059-01 to L.M.Y.). This work was supported by Hospices Civils de Lyon Fondation Hospices Civils de Lyon (HPI COVID) (A.B., S.T.-A. and E.J.), FONDECYT 1221802 (M.C.P.), ATE220061 (M.C.V.C. and M.C.P.), FONDECYT 1241530 (M.C.V.C.), and the Intramural Research Program of the National Institute of Allergy and Infectious Diseases, NIH (H.C.S.). A.B. and S.T.-A. were funded by ANR ANR-20-COVI-0064; A.B. and S.T.-A., were funded by the ANR-RHU Program ANR-21-RHUS-08 (COVIFERON) and A.B. was funded by the HORIZON-HLTH-2021-DISEASE-04 programme under grant agreement 01057100 (UNDINE).

**Author contributions** C.C.G., M. Massoud, S.F., G.M.G., M.F.-G., J.R.L., M.B., L.E., S.K.-P., E.J., S.P., S.T.-A., B. Sahin, M.V., M.H., A.I.H., L.-M.B., N.W., L.A., N.U., K.L., M. Matz, A.K., M. Maurer, B.E.G., A.B., A.R., T.K. and M.-F.M. performed or supervised experiments, and generated and analysed data. P.D., F.H. and C.C.G. performed computational analysis of data. T.K., A.B., A.S.L.v.S., S.W., S.O., Y.O., Y.Z., H.C.S., M.C.P., V.D., A.L.V., J.F.R., L.M.Y., K.D., C.A., Y.E., M.C.V.C., O.M.D., G.A.M.S., M. Magliocco, K.B., J.D., L.E., A.G., H.v.B., C.C.G., M.W., B. Sawitzki, L. Petrov, L. Peter, M.S.-H. and M.A.M. evaluated and recruited patients and/or controls. C.C.G., P.D., T.K. and M.-F.M. wrote the original draft with the help of the other co-authors. M.-F.M., T.K., C.C.G. and P.D. conceptualized the project. M.-F.M. directed the project and was responsible for funding acquisition and is the lead corresponding author.

**Funding** Open access funding provided by German Rheumatology Research Center, a Leibniz-Institute (DRFZ).

**Competing interests** M.-F.M., C.C.G., T.K. and P.D. filed a patent using TGFβ blockade for management of MIS-C and other post-COVID-19 sequelae. S.P. is an employee of bioMérieux. Y.O. received lecture fees from Pfizer on topics unrelated to this article. S.O. received speaker and/or consultancy fees from Novartis and Sobi on topics unrelated to this article. The authors declare no further competing interests.

**Additional information**
**Correspondence and requests for materials** should be addressed to Carl Christoph Goetzke, Tilmann Kallinich or Mir-Farzin Mashreghi.

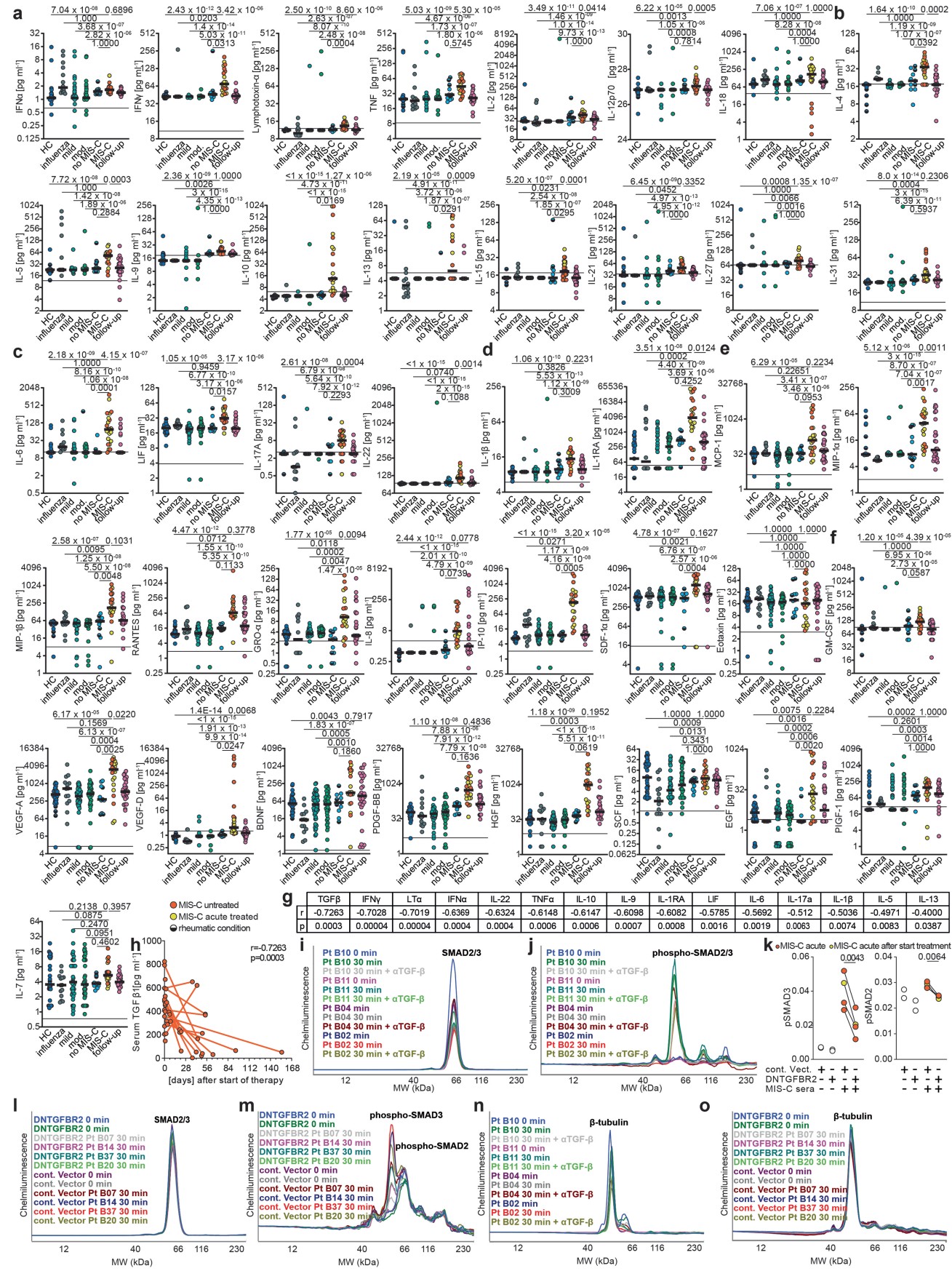

**Extended Data Fig. 1** | See next page for caption.

**Extended Data Fig. 1 | Multiplex cytokine profiling in MIS-C. a-f**, Cytokine, chemokine and growth factor profiles of serum from MIS-C patients during the first seven days of hospitalization ($n = 22$), untreated samples are marked in red, and at follow-up visits ($n = 28$ time points, 23 patients), paediatric patients during an acute infection with a asymptomatic to mild SARS-CoV-2 (mild; $n = 57$ of which $n = 13$ had no symptoms) or a moderate to severe (mod.; $n = 42$ of which teo had severe symptoms), of non-infected children ($n = 40$ of which five have an underlying rheumatic condition) and of children six weeks after SARS-CoV-2-infection, that did not develop MIS-C ($n = 11$ of which $n = 5$ have an underlying rheumatic condition). **a**, Type 1 cytokines, **b**, type 2 cytokines, **c**, type 3 cytokines, **d**, IL-1 family cytokines and **e**, chemokines and **f**, growth factors. **g**, Correlation of cytokine quantity versus time after hospitalization and initiation of treatment. **h**, Serum levels over time (0 indicates day of initiation of treatment) in individuals with MIS-C are indicated and decline in TGFβ1 serum levels was correlated to time after initiation of treatment. **i-j**, Graph view results of simple western size-based assay results for **i**, SMAD2/3 and **j**, phospho-SMAD2/3 (Ser465/467) (Fig. 1c) of primary T cells restimulated ex vivo with sera from indicated patients for 30 min with or without addition of a neutralizing anti-TGFβ prior to stimulation. **k-m**, HEK293T cells were transfected with a plasmid expressing a dominant negative mutant human *TGFBR2* or a non-expressing plasmid with Blasticidin-resistance. After positive selection with Blasticidin, cells were stimulated with MIS-C patients' sera (MIS-C; $n = 4$; yellow indicates sampling after start of treatment as indicated in **i**) for 30 min or left unstimulated, lysed and SMAD2/3 and phospho-SMAD2/3 (Ser465/467) levels were quantified by Simple Western Size-Based assay. **k**, Phospho-SMAD2/3 (Ser465/467) levels normalised to total SMAD2/3 levels. **l-m**, Graph view results of simple western size-based assay results for (**l**) SMAD2/3 and **m** phospho-SMAD2/3 (Ser465/467). **n-o**, Graph view results of Tubulin used as a loading control for (**n**) T cells and (**o**) transfected HEK293T cells. **a-f**, Lower limits of quantification are indicated by horizontal lines, short lines indicate medians. A non-parametric ANOVA (Kruskal-Wallis test) was used followed by a Dunn's multiple comparison test with correction for multiple comparisons, **g-h**, a two-tailed spearman-correlation was used, or **k**, a two-tailed ratio paired t-test was used.

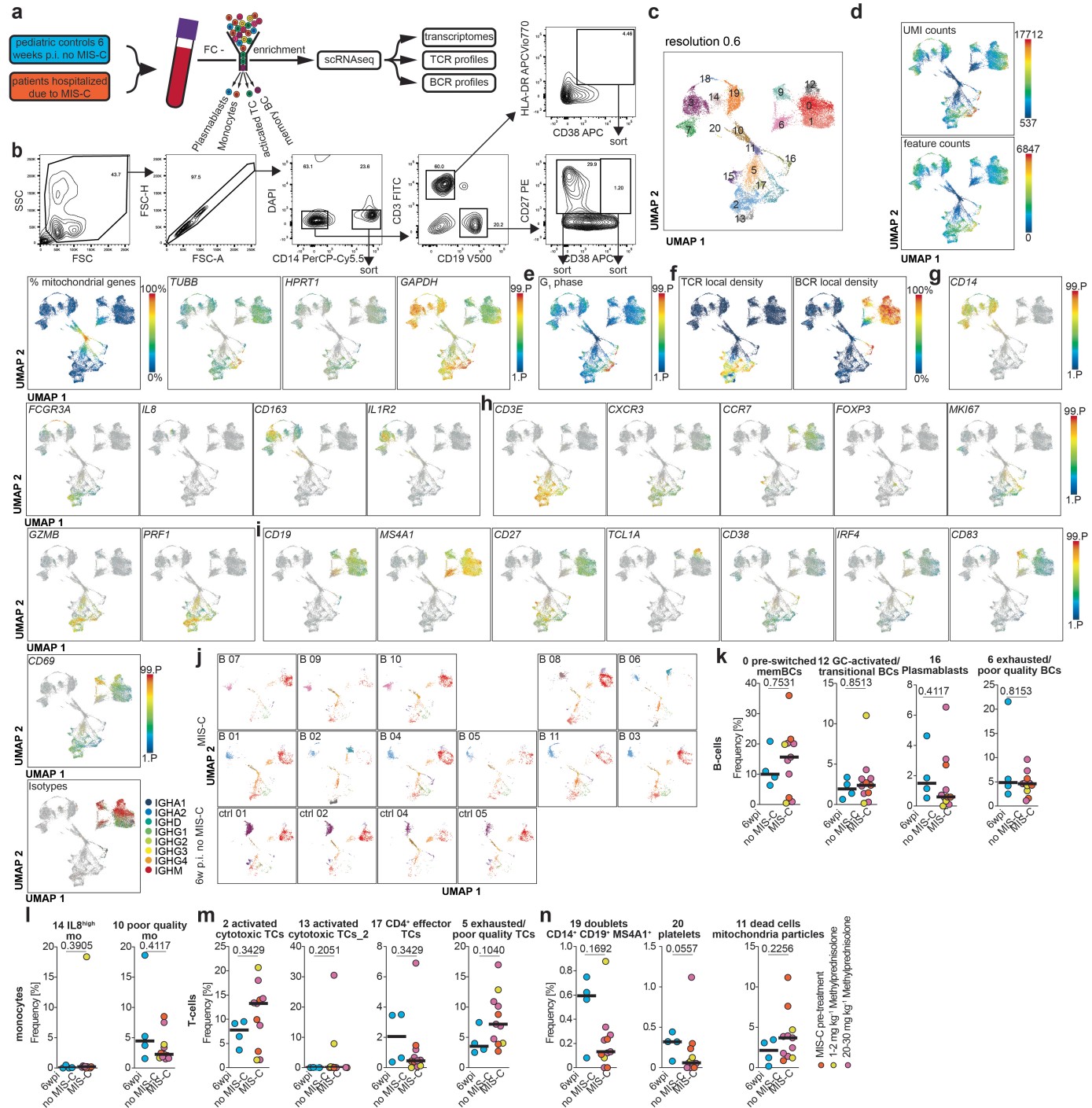

**Extended Data Fig. 2 | Single cell sequencing in MIS-C. a**, Schematic outline of the scRNAseq approach. **b**, Flow cytometry gating strategy. **c**, UMAP with clustering at resolution 0.6 of all patients and controls combined. **d**, Quality controls are depicted on this UMAP. UMI counts, feature counts and % mitochondrial genes are depicted for each cell. As additional quality controls, gene expression for the indicated housekeeping genes are depicted. **e**, To assess cell proliferation expression of $G_1$-phase cell cycle genes are plotted on the UMAP. **f**, Local density of TCR and BCR expressing cells is plotted. **g-i**, Relative expression of the indicated genes are plotted onto the UMAP to identify (**g**) monocyte, **h**, T-cell and (**i**) B-cell subsets including BCR isotypes. Cluster naming is based on marker gene expression and top differentially expressed genes (Supplementary Data 1). **j**, UMAP separated by individual

patients organised by duration and dosage of methylprednisolone treatment. B07 (MIS-C patient Berlin 07), B09 and B10 were captured before treatment with methylprednisolone, B08 was treated with 1-2 mg/kg methylprednisolone for two days, B06 was treated with 1-2 mg kg$^{-1}$ methylprednisolone for three days. Patients B01, B02 B04, B05 and B11 were treated with 20–30 mg kg$^{-1}$ methylprednisolone for one day and B03 was treated with 20–30 mg kg$^{-1}$ methylprednisolone for four days before capturing (Supplementary Data 8). Paediatric controls 6 wpi with SARS-CoV-2 (ctrl 1–4) did not receive medication. **k-n**, Frequencies of cells per cluster in both groups ($n$ = 4 no MIS-C and 11 MIS-C) for clusters not included in Fig. 1. Treatment with methylprednisolone is colour-coded in **n. k-n**, two-tailed Mann-Whitney-$U$-tests.

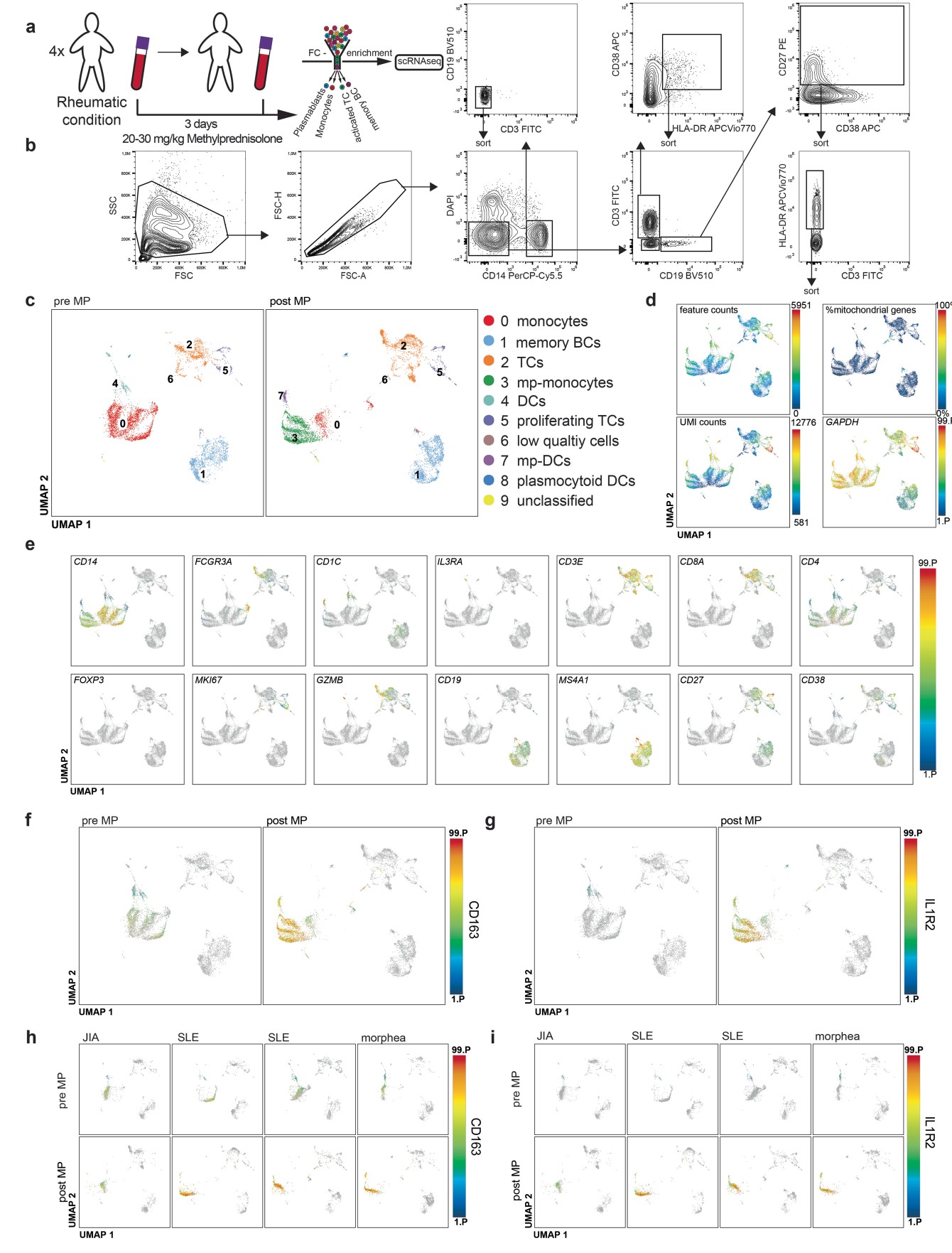

**Extended Data Fig. 3 |** See next page for caption.

**Extended Data Fig. 3 | Single cell analysis of methylprednisolone treated patients. a**, Schematic outline of the scRNAseq approach to identify the effect of methylprednisolone treatment. **b**, Gating strategy used for scRNAseq experiment. **c**, UMAP of a total of 13646 cells enriched by FACS for monocytes, HLA-DR[high] cells, CD38[+] T cells and CD27[+] B cells of $n$ = 4 patients prior (pre MP, 6839 cells) and after (post MP, 6807 cells) a three-day treatment with 20–30 mg kg[−1] day[−1] methylprednisolone. One patient was diagnosed with juvenile idiopathic arthritis (JIA), two patients with systemic lupus erythematosus (SLE) and one patient with morphea. **d**, Quality controls of the integrated data are depicted on this UMAP and UMI counts, feature counts and % mitochondrial genes are depicted for each cell. As an additional quality control, gene expression for the housekeeping gene *GAPDH* is depicted. **e**, Core marker gene expression for each cell is depicted and used for naming the clusters along with the top differentially expressed genes (Supplementary Data 2). **f**, *CD163* expression and (**g**) *IL1R2* which defines cluster 3 and 18 of the MIS-C cohort is depicted separately for pre MP and post MP. **h-i**, UMAP separated for individual patients' cells before and after treatment are depicted with (**h**) *CD163* expression and (**i**) *IL1R2* expression depicted separately.

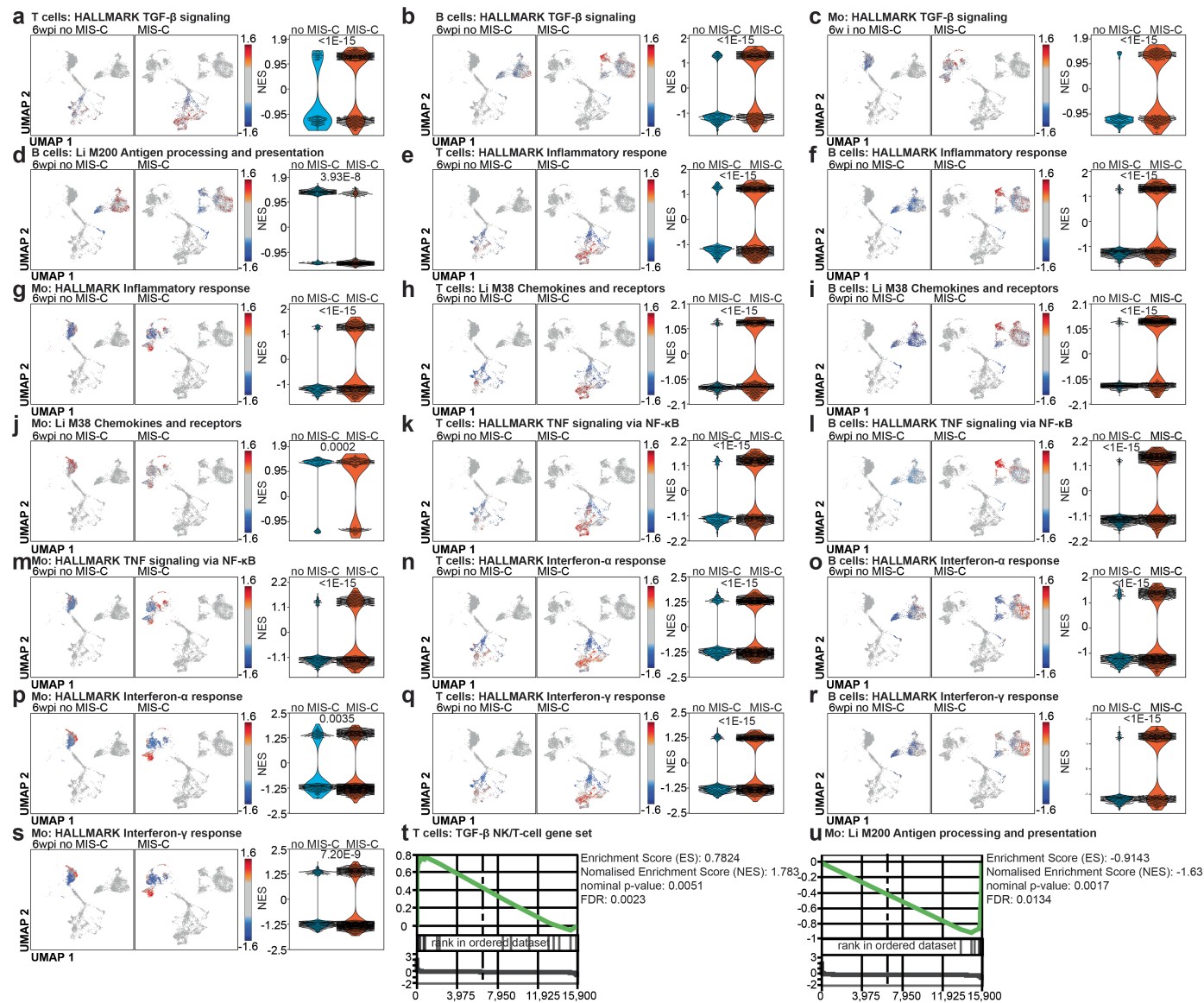

**Extended Data Fig. 4 | Gene Set Enrichment Analysis on a single cell level in MIS-C. a-s,** GSEA for indicated gene sets on indicated cell populations. For each GSEA the normalised enrichment score (NES) is depicted on the rarefied UMAP (see Fig.1d). The NES of all cells with significant positive or negative enrichment are plotted in a violin-plot. *P*-values are determined by two-tailed Fisher's exact test testing for positive versus negative enrichment. **t-u,** Pseudobulk GSEA for indicated gene sets on indicated cell populations. Enrichment Score, Normalised Enrichment Score, nominal *P*-value and FDR are indicated in next to the plot.

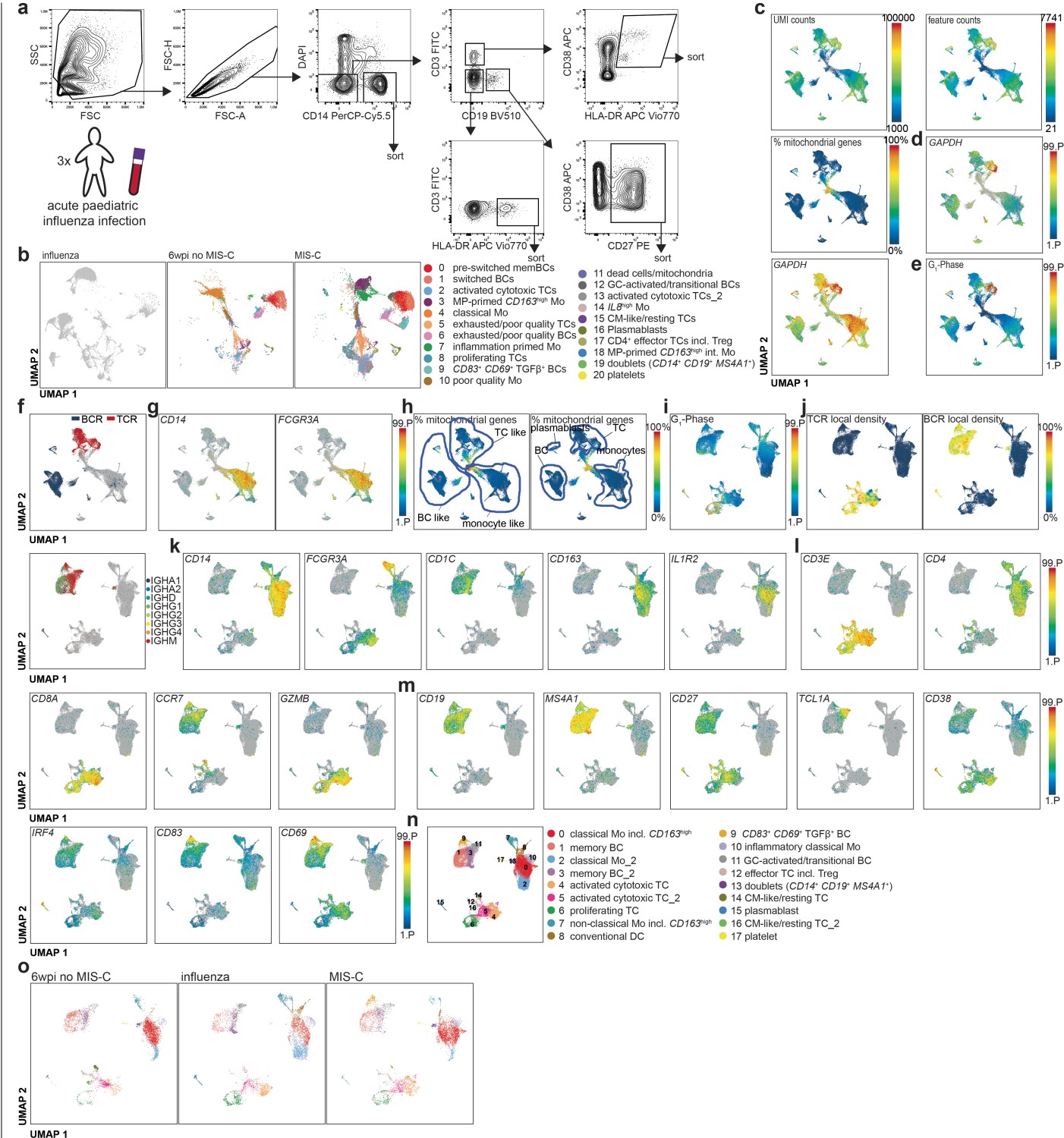

**Extended Data Fig. 5** | See next page for caption.

**Extended Data Fig. 5 | Single cell sequencing of additional influenza infected patient derived samples. a**, Gating strategy used for scRNAseq experiment of influenza infected patients. **b**, Projection of the clusters from Fig. 1d onto a split UMAP of non-integrated samples from the 3 influenza infected patients combined with data from Fig. 1d. **c**, Quality controls of the integrated data from three patients hospitalized due to influenza infection, eleven patients with MIS-C and four patients 6 wpi no MIS-C. A total of 44206 cells are depicted on this UMAP and UMI counts, feature counts and % mitochondrial genes are depicted for each cell. **d**, As additional quality controls, gene expression for the indicated housekeeping genes are depicted. **e**, To assess cell proliferation expression of $G_1$-phase cell cycle genes are plotted on the UMAP. **f**, Cells expressing a BCR and TCR are colour-coded on the UMAP to identify B cells and T cells. **g**, The relative expression of the indicated genes are plotted onto the UMAP to identify monocytes. **h**, Gating of B cell like (BC like), T cell like (TC like) and monocyte like cells is depicted on the UMAP low quality cells are included for calculating frequencies (left), whilst low quality cells were removed for all other following analysis (right). **i-o**, After removal of the low quality cells, monocyte like, B cell like and T cell like cells were integrated for further analysis. **i**, To assess cell proliferation expression of $G_1$-phase cell cycle genes are plotted on the UMAP. **j**, Local density of TCR, BCR and BCR isotypes are plotted on the UMAP. **k-m**, The relative expression of the indicated genes are plotted onto the UMAP to identify (**k**) monocyte, (**l**) T-cell and (**m**) B-cell subsets. **n**, Depicts the clustering at resolution 0.5 on the UMAP and (**o**) split by group. Cluster naming is based on marker gene expression (Extended Data Fig. 5j–m) and top differentially expressed genes (Supplementary Data 17). Frequencies of cells per cluster are in Supplementary Data 18.

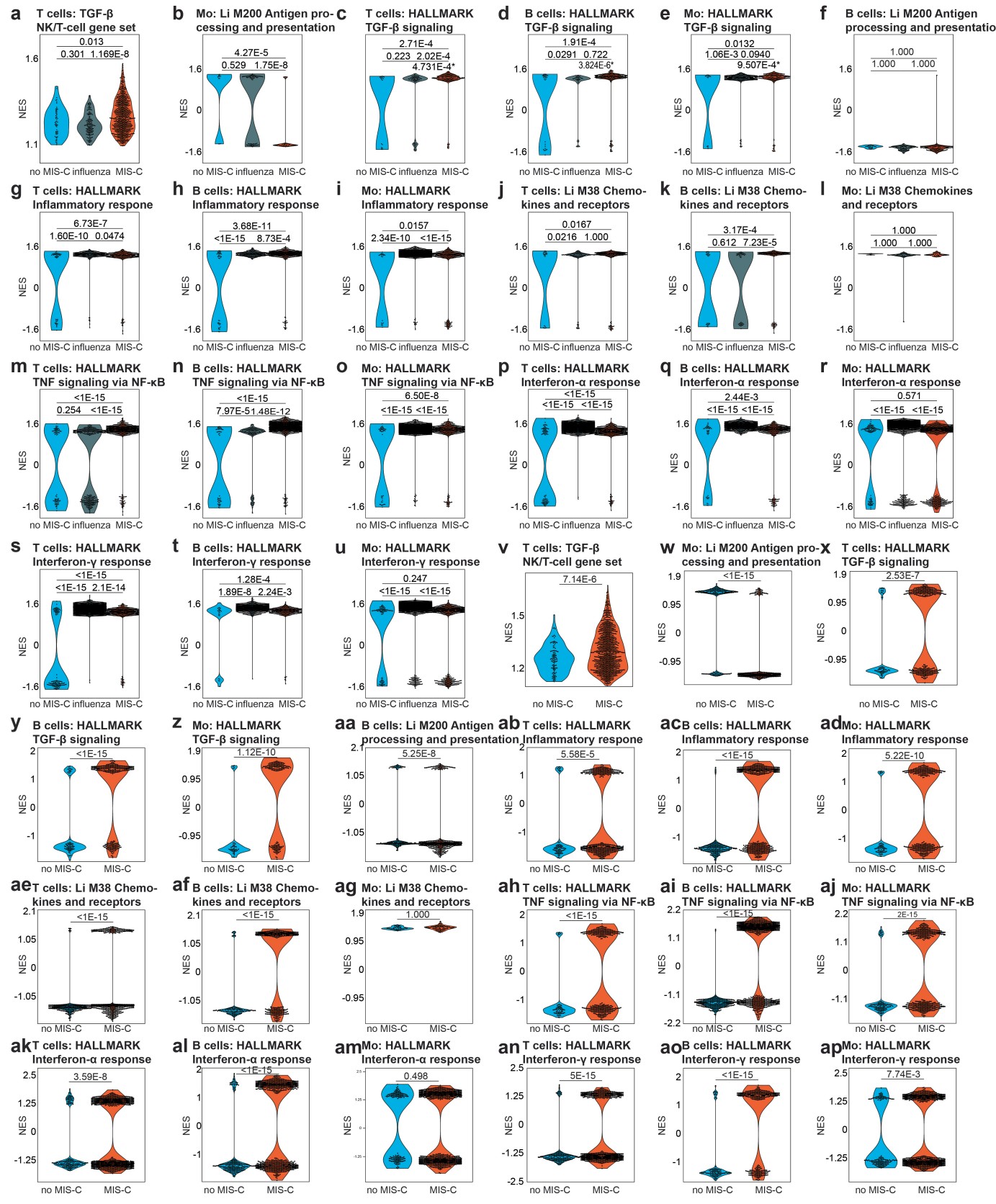

**Extended Data Fig. 6 | Gene Set Enrichment Analysis on a single cell level in MIS-C and influenza infection.** In contrast to Fig. 2 and Extended Data Fig. 4 a FDR of <0.25 was used for identifying significantly up- or downregulated genes. **a-u**, GSEA for indicated gene sets on indicated cell populations. The NES of all cells with significant positive or negative enrichment are plotted in a violin-plot. **v-ap**, GSEA for indicated gene sets on indicated cell populations. The NES of all cells with significant positive or negative enrichment are plotted in a violin-plot. *P*-values are determined by two-tailed Fisher's exact test testing for positive versus negative enrichment, or by two-tailed Mann-Whitney-*U*-tests to test for positivity of NES (for **a** + **v** and for **c-e**; marked by an *).

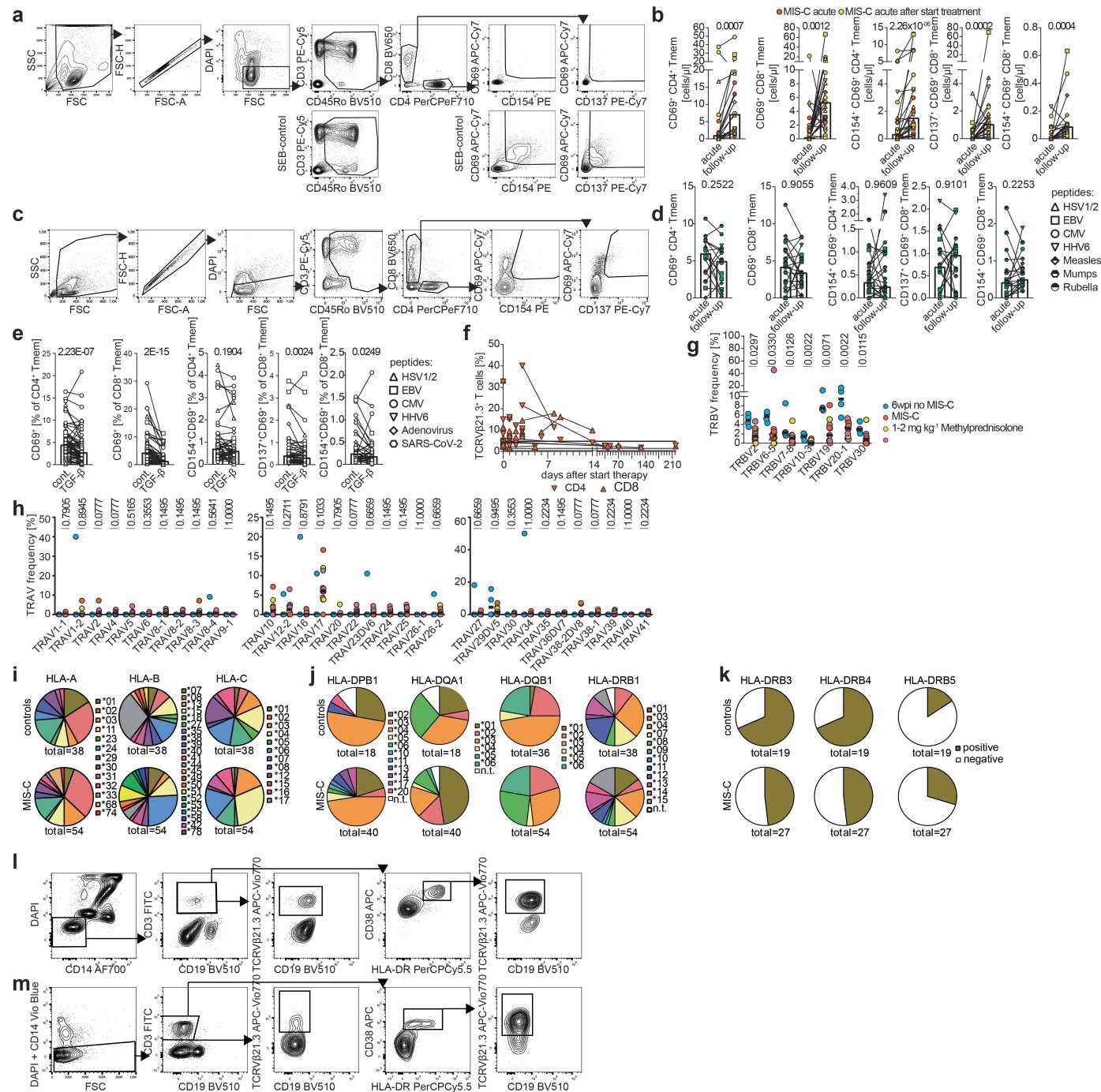

**Extended Data Fig. 7 | TGFβ impairs T-cell activation to viral epitopes.**
**a**, Gating strategy used in flow cytometry of the T cell reactivity assays
depicted in Fig. 2d–f. Cells were identified by size and granularity in a FSC-vs
SSC plot, followed by doublet exclusion in an FSC-A vs. FSC-H plot. Dump⁺
(DAPI, CD14 and CD19)⁺ cells were also excluded. As CD3 is downregulated after
T cell activation (SEB plot in second row), the gate was extended to include
CD3^low CD45RO⁺ cells. CD4⁺ epitope-specific T cells were identified as CD69⁺
CD154⁺ and CD8⁺ epitope specific T cells were identified as CD69⁺ CD137⁺ or as
CD69⁺ CD154⁺. SEB was used as a positive control for correct gating. **b**, Cell
counts for CD69⁺ or CD69⁺ and CD154⁺ or CD137⁺ memory T cells from Fig. 2d.
**c**, Gating strategy used in flow cytometry of the T cell reactivity assays
depicted in d. **d**, Frequencies of overall activated and antigen-specific
reactivated cells of CD4⁺ and CD8⁺ memory T cells from six children with a
confirmed infection with SARS-CoV-2 during the acute phase and follow-up
upon after resolution of symptoms. **e**, Frequencies of overall activated and

antigen-specific reactivated cells of CD4⁺ and CD8⁺ memory T cells from
healthy donors (n = 6) treated with 50 ng ml⁻¹ TGFβ1. **f**, Frequencies of
TCRVβ21.3⁺ on total T cells were quantified by Flow cytometry over time after
treatment start with IVIG and methylprednisolone. Horizontal lines indicate
normal range (0.9-4.9% for CD8⁺ T cells; 1.5-4-7% for CD4⁺ T cells) of TCRVβ21.3⁺
T cells (n = 25, children with MIS-C). **g**, Significantly regulated *TRBV* determined
by TCR sequencing of activated T cells. Dots indicate the frequency of specific
*TRBV* in each sample relative to all TCRs sequenced. **h**, Frequencies of *TRAV*
gene associated to *TRBV11-2*⁺ T cells not depicted in Fig. 3f. **i**, HLA-class I
haplotyping and (**j-k**) HLA-class-II haplotyping of our MIS-C cohort (n = 20
patients and n = 10 healthy controls including the 4 children used as a control
for the scRNAseq experiments). Additionally, HLA-haplotyping from a
previously published MIS-C cohort (n = 7 patients and 9 controls)[12] was
included. **l-m**, Sorting strategy for Fig. 3e. *P*-values for (**b** + **d-e** + **h**) were
determined by paired two-tailed Mann-Whitney-*U*-tests.

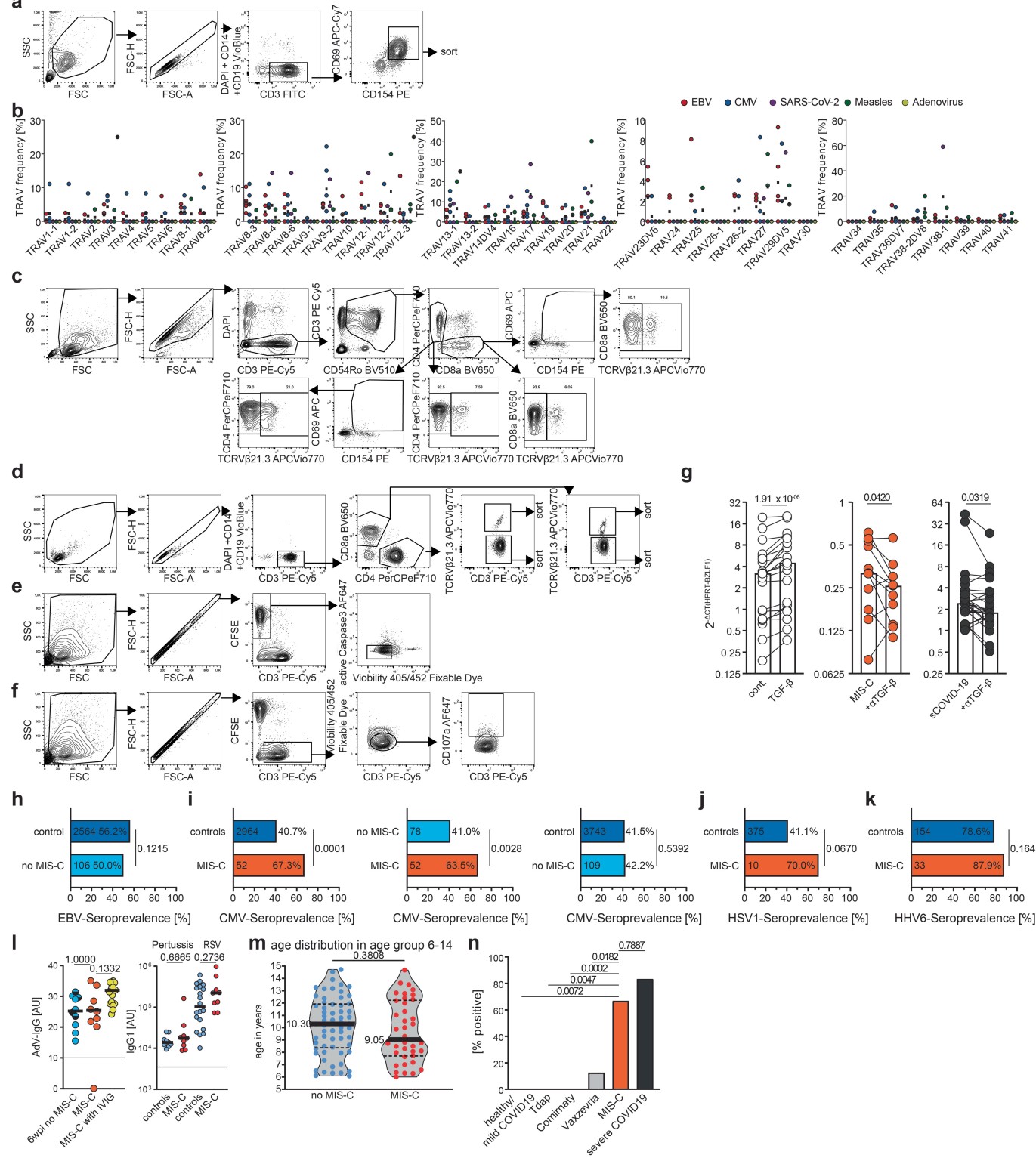

**Extended Data Fig. 8** | See next page for caption.

**Extended Data Fig. 8 | Antiviral immunity in MIS-C. a**, Flow cytometry gating strategies used for sorting of T cells obtained after the ARTE-assay and used for generation of a virus-specific TCR library (Fig. 4a, b). **b**, Frequencies of *TRAV* gene associated to *TRBV11-2*[+] T cells from either EBV-, CMV-, SARS-CoV-2-, Measles-, or AdV-specific T cells used for clustering in Fig. 4e. **c**, Flow cytometry gating strategy of PBMC stimulated with the EBNA2$_{275-294}$-peptide PRSPTVFY NIPPMPLPPSQL or the EBNA2$_{279-289}$-peptide TVFYNIPPMPL for Fig. 4f. Flow cytometry gating strategies used (**d**) for sorting of TCRVβ21.3-positive or -negative CD4[+]- or CD8[+] T cells for co-culture experiments (Fig. 5a), (**e**) analysing viable EBV infected B cells after co-culture with T cells for Fig. 5b,c and f analysing CD107a[+] T cells after co-culture with EBV-infected B cells (Fig. 5b,c). **g**, LCL from healthy donors (*n* = 6) were treated with TGF-β, or with serum from MIS-C (n = 5) or severe COVID-19 (*n* = 6), with or without pre-incubation with anti-TGFβ blocking antibodies. EBV reactivation was evaluated by qPCR by quantifying the reactivation transcription factor *BZLF1*. Results were calculated using the 2$^{-\Delta Ct}$ method. **h-k**, Seroprevalences for different virus were compared between MIS-C patients, patients that did not develop MIS-C after infection with SARS-CoV-2 and controls after age-matching (Supplementary Data 8–12+15). Group sizes and percentages of and antibody-positivity are indicated on bars. **h**, Age-matched controls for EBV[50] were compared to children that did not develop MIS-C after infection with SARS-CoV-2. **i**, CMV-seroprevalence was compared for children with MIS-C and age-matched controls[70] or age-matched children that did not develop MIS-C after infection with SARS-CoV-2, and children that did not develop MIS-C and age-matched controls[70]. **j**, Age-matched controls for HSV-1 seroprevalence[71] and (**k**) age matched controls for HHV6-seroprevalence[72] were compared to MIS-C patients. **l**, Antibody tires for anti-Adenovirus-IgG was compared between MIS-C patients (*n* = 9, of which 8 had a positive titre), children six weeks p.i. who did not develop MIS-C (*n* = 10) and MIS-C patients treated with IVIG (*n* = 15) from the Berlin cohort. For the Boston cohort[49] anti-virus-IgG1- titres were compared between MIS-C patients (*n* = 9) and pre-pandemic paediatric controls (*n* = 9 Pertussis, *n* = 20 RSV). **m**, Age-distribution within the largest age-group used for age matching for no MIS-C and MIS-C patients. **n**, Percentage of actively EBV-mRNA transcribing patients determined by scRNAseq of B cells and plasmablasts. *P*-values for (**g**) are calculated by a two-tailed Wilcoxon signed-rank test; for (**h-k**) are calculated by one-tailed Fisher's exact test to test if seroprevalences are increased in MIS-C or decreased in no MIS-C and for (**l**) by two-tailed Mann-Whitney U-tests comparing only the positive samples or determined by a non-parametric ANOVA (Kruskal-Wallis test) followed by a Dunn's multiple comparison test with correction for multiple comparisons, comparing all groups to MIS-C; **m**, are calculated using a two-tailed Mann-Whiteney-*U*-test and for (**n**) a one-tailed Chi-Squared tests to test if EBV mRNA positivity is enriched in MIS-C were calculated.

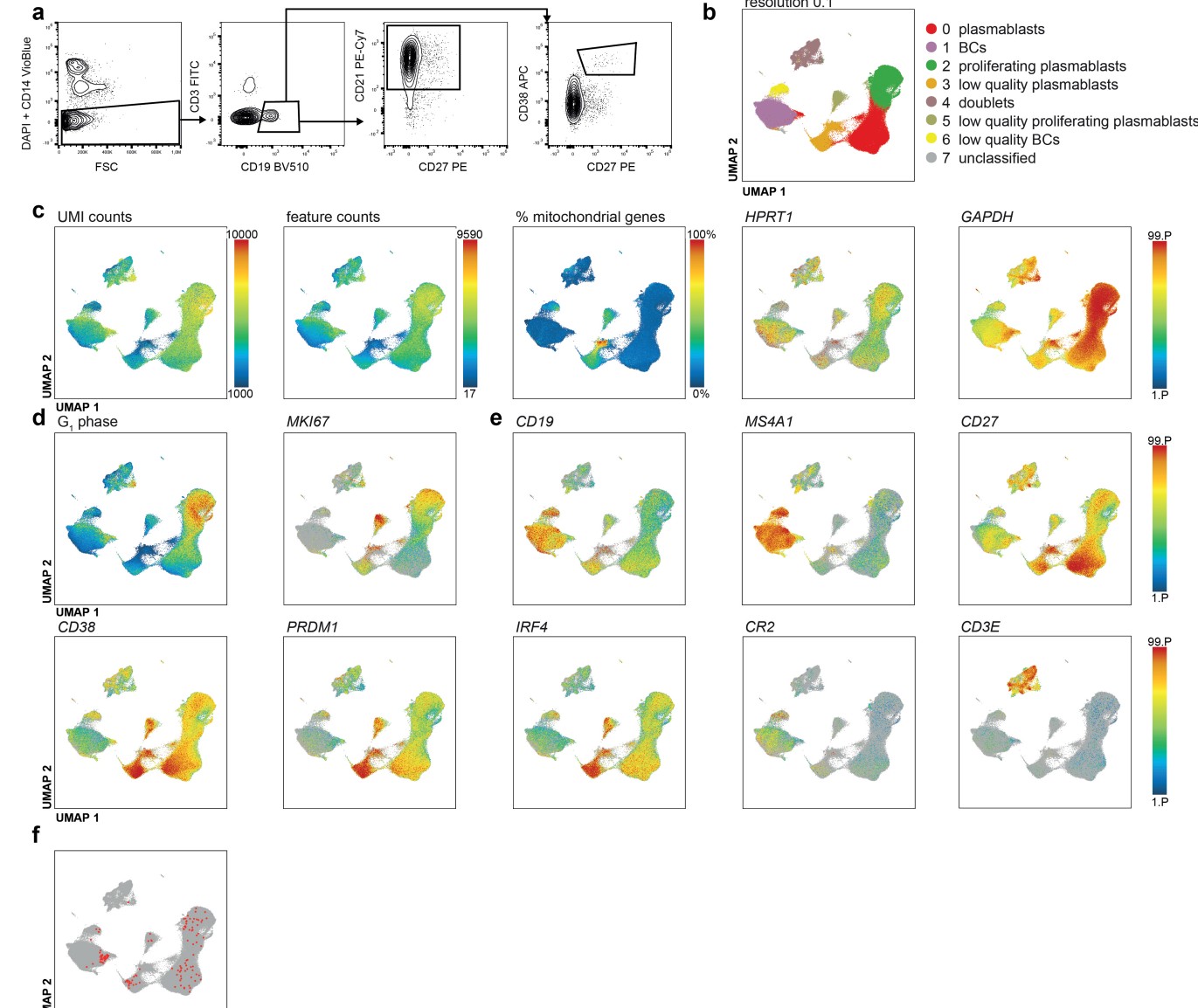

**Extended Data Fig. 9 | Detection of EBV-mRNA by transcriptomics. a**, Flow cytometry gating strategies used for sorting of B cells and plasmablasts for single cell transcriptomics. **b**, Integrated UMAP depicting 186,990 cells from healthy adults or adults with mild COVID-19 ($n$ = 6)[4], healthy adults on day seven post vaccination (Tdap vaccine, $n$ = 7 of which one had mild COVID-19 at time of sampling, Comirnaty vaccine $n$ = 15 or Vaxzevria vaccine $n$ = 8)[57], MIS-C ($n$ = 6) or adults with severe COVID-19 ($n$ = 12)[4]. Clustering at a resolution of 0.1 was used to show the position of B cells and plasmablasts on the UMAP.

**c**, For quality controls of the B cell and plasmablast UMAP UMI counts, feature counts and % mitochondrial genes are depicted for each cell. As additional quality controls, gene expression for the indicated housekeeping genes are depicted. **d**, To assess cell proliferation expression of $G_1$-phase cell cycle genes and *MKI67* are plotted on the UMAP. **e**, The relative expression of the indicated genes are plotted onto the UMAP to identify B cells, plasmablasts and doublets. **f**, Cells positive for EBV-mRNA that could be mapped onto the UMAP are highlighted in red.

**Extended Data Table 1 | Clinical characteristics of the MIS-C patient cohort included in this study**

| | Berlin cohort | | Included in serum studies | | Included in cell assays | | Included in single-cell sequencing | |
|---|---|---|---|---|---|---|---|---|
| Patients included | 39 | 100% | 28 | 73.7% | 20 | 52.6% | 11 | 28.9% |
| **Characteristics** | Median | IQR | Median | IQR | Median | IQR | Median | IQR |
| Age (years) | 8.7 | 5.5–10.2 | 8.1 | 4.9–10.3 | 8.3 | 5.5–9.8 | 8.1 | 4.4–9.6 |
| BMI (z-score) | 0.16 | -0.59–0.97 | 0.13 | -0.62–0.83 | -0.21 | -0.72–1.15 | 0.38 | -0.72–0.93 |
| | Female | Male | Female | Male | Female | Male | Female | Male |
| Sex at birth | 14 | 25 | 11 | 17 | 10 | 10 | 6 | 5 |
| **Ethnicity** | n | % | n | % | n | % | n | % |
| African | 3 | 7.7% | 3 | 10.7% | 3 | 15% | 2 | 18.2% |
| Asian | 1 | 2.6% | 1 | 3.6% | 1 | 5% | 0 | 0% |
| Caucasian | 20 | 51.3% | 10 | 35.7% | 8 | 40% | 3 | 27.3% |
| Middle east | 13 | 33.3% | 13 | 46.4% | 7 | 35% | 5 | 45.5% |
| Mixed/others | 2 | 5.1% | 1 | 3.6% | 1 | 5% | 1 | 9.1% |
| **Symptoms** | n | % | n | % | n | % | n | % |
| Fever | 39 | 100% | 28 | 100% | 20 | 100% | 11 | 100% |
| Mucocutaneous | 37 | 94.9% | 28 | 100% | 19 | 95.0% | 11 | 100% |
| Rash | 33 | 84.6% | 22 | 78.6% | 19 | 95.0% | 10 | 90.9% |
| Conjunctivitis | 29 | 74.4% | 22 | 78.6% | 17 | 85.0% | 9 | 81.8% |
| Mucositis | 21 | 53.8% | 17 | 60.7% | 8 | 40.0% | 5 | 45.5% |
| Cardiac | 25 | 64.1% | 17 | 60.7% | 15 | 75.0% | 9 | 81.8% |
| Hypotension | 22 | 56.4% | 7 | 25.0% | 13 | 65.0% | 8 | 72.7% |
| Reduced LVEF% | 12 | 30.8% | 15 | 53.6% | 8 | 40.0% | 5 | 45.5% |
| Neurological impairment | 1 | 2.6% | 1 | 3.6% | 1 | 5.0% | 0 | 0% |
| Abdominal | 33 | 84.6% | 25 | 89.3% | 19 | 95.0% | 11 | 100% |
| Abdominal pain | 23 | 59.0% | 18 | 64.3% | 14 | 70.0% | 9 | 81.8% |
| Ascites | 19 | 48.7% | 11 | 39.3% | 13 | 65.0% | 8 | 72.7% |
| Mesenteric lymphadenitis | 8 | 20.5% | 7 | 25.0% | 4 | 20.0% | 2 | 18.2% |
| Diarrhea | 16 | 41.0% | 15 | 53.6% | 10 | 50.0% | 6 | 54.5% |
| **Laboratory results** | Median | IQR | Median | IQR | Median | IQR | Median | IQR |
| Max. CrP (mg/l) | 141 | 106–226 | 164 | 111–235 | 126 | 92–162 | 118 | 86–164 |
| Max. NP-pro-BNP (ng/l) | 8393 | 4158–14398 | 8253 | 4074–15368 | 10193 | 6347–21306 | 11440 | 6031–29693 |
| Max. D-Dimer (mg/l) | 4.2 | 2.3–6.6 | 4.8 | 2.8–6.8 | 3.9 | 2.7–5.7 | 4.5 | 3.5–8.0 |
| Min. LVEF (%) | 53 | 47–57 | 53 | 50–56 | 52 | 44–55 | 50 | 41–55 |
| **Treatment** | | | | | | | | |
| Days hospitalised | 9 | 8–13 | 10 | 8–15 | 9 | 8–10 | 9 | 9–10 |
| Highest level of care | n | % | n | % | n | % | n | % |
| ICU | 13 | 33.3% | 8 | 28.6% | 7 | 35.0% | 4 | 36.4% |
| IMC | 17 | 43.6% | 14 | 50.0% | 9 | 45.0% | 6 | 54.5% |
| ward | 9 | 23.1% | 6 | 21.4% | 4 | 20.0% | 1 | 9.1% |
| Medication | | | | | | | | |
| IVIG | 39 | 100% | 28 | 100% | 20 | 100% | 11 | 100% |
| 1-2mg/kg MP | 18 | 46.2% | 13 | 46.4% | 5 | 25.0% | 3 | 27.3% |
| 20-30mg/kg MP | 19 | 48.7% | 12 | 42.9% | 15 | 75.0% | 8 | 72.7% |
| Anakinra | 5 | 12.8% | 2 | 7.1% | 2 | 10.0% | 1 | 9.1% |
| ASA | 38 | 97.4% | 28 | 100% | 19 | 95.0% | 11 | 100% |
| ionotropic | 12 | 30.8% | 8 | 28.6% | 7 | 35.0% | 4 | 36.4% |
| Respiratory support | 14 | 35.9% | 9 | 32.1% | 7 | 35.0% | 3 | 27.3% |
| intubation +ventilation | 1 | 2.6% | 1 | 3.6% | 1 | 5.0% | 0 | 0% |
| O2-supplement | 14 | 35.9% | 9 | 32.1% | 7 | 35.0% | 3 | 27.3% |

Characteristics of patients whose samples were included in serum-based studies (cytokine/chemokine measurements, in vitro testing of TGFβ experiments), from which cells were isolated during the acute phase and used for ex vivo studies. BMI was quantified by z-score relative to published paediatric percentiles for BMI[75]. Reduced LVEF was defined as a LVEF less than 50%. Only reported symptoms and reported treatment is included in this table. Inotropics used include adrenaline and milrinone. Not determined was treated as not present. (LVEF = left ventricle ejection fraction, CrP = C-reactive Protein, NP-pro-BNP=N-terminal prohormone of brain natriuretic peptide, ICU = intensive care unit, IMC = intermediate care unit, IVIG = intravenous immunoglobulins, MP = methylprednisolone, ASA = acetylsalicylic acid, IQR = interquartile range).

Tilmann Kallinich,
Mir-Farzin Mashreghi (lead)

# Reporting Summary

## Statistics

For all statistical analyses, confirm that the following items are present in the figure legend, table legend, main text, or Methods section.

| n/a | Confirmed | |
|-----|-----------|---|
| ☐ | ☒ | The exact sample size (*n*) for each experimental group/condition, given as a discrete number and unit of measurement |
| ☐ | ☒ | A statement on whether measurements were taken from distinct samples or whether the same sample was measured repeatedly |
| ☐ | ☒ | The statistical test(s) used AND whether they are one- or two-sided *Only common tests should be described solely by name; describe more complex techniques in the Methods section.* |
| ☒ | ☐ | A description of all covariates tested |
| ☐ | ☒ | A description of any assumptions or corrections, such as tests of normality and adjustment for multiple comparisons |
| ☐ | ☒ | A full description of the statistical parameters including central tendency (e.g. means) or other basic estimates (e.g. regression coefficient) AND variation (e.g. standard deviation) or associated estimates of uncertainty (e.g. confidence intervals) |
| ☐ | ☒ | For null hypothesis testing, the test statistic (e.g. *F*, *t*, *r*) with confidence intervals, effect sizes, degrees of freedom and *P* value noted *Give P values as exact values whenever suitable.* |
| ☒ | ☐ | For Bayesian analysis, information on the choice of priors and Markov chain Monte Carlo settings |
| ☒ | ☐ | For hierarchical and complex designs, identification of the appropriate level for tests and full reporting of outcomes |
| ☐ | ☒ | Estimates of effect sizes (e.g. Cohen's *d*, Pearson's *r*), indicating how they were calculated |

*Our web collection on statistics for biologists contains articles on many of the points above.*

## Software and code

Policy information about availability of computer code

| | |
|---|---|
| Data collection | Flow cytometry data was aquired using the software of the Multi-Application MA900 Cell Sorter (Sony Biotechnology) and MACSQuantify™ Software of the MACSQuant® Analyzer 16 Flow Cytometer (Miltenyi Biotec). Simple Western Size-Based assays were run on a Wes instrument (proteinsimple; Bio-Techne). |
| Data analysis | Raw sequence reads were processed using cellranger version 5.0.0. Statistics and data analysis was performed in R (version 4.1.2). Details of used packages are described in the Material and Methods section of the manuscript. Loupe Browser (version 5, 10x Genomics) was used to identify and define cells of interest by manual gating. Flow cytometry data was analysed using FlowJo software 10.8.1 (TreeStar). Simple Western Size-based assays were analysed using the Compass for SW (BioTechne; Protein Simple version 3.1.7). |

For manuscripts utilizing custom algorithms or software that are central to the research but not yet described in published literature, software must be made available to editors and reviewers. We strongly encourage code deposition in a community repository (e.g. GitHub). See the Nature Portfolio guidelines for submitting code & software for further information.

# Data

Policy information about availability of data

All manuscripts must include a data availability statement. This statement should provide the following information, where applicable:
- Accession codes, unique identifiers, or web links for publicly available datasets
- A description of any restrictions on data availability
- For clinical datasets or third party data, please ensure that the statement adheres to our policy

Next Generation Sequencing data sets are available in the GEO repository GSE254179. . Data was mapped using the human genome reference hg19 [https://www.10xgenomics.com/support/software/cell-ranger/downloads/cr-ref-build-steps]. data sets used for GSEA are available in the Molecular Signatures Database (MSigDB) [https://www.gsea-msigdb.org/gsea/msigdb/] and in the GEO repository under accession number GSE184329. Activated B cell and Plasmablast datasets used from previous studies can be found in the respective repositories: GSE253862 and GSE158038. Source data and additional supporting data are provided with this paper in the supporting information section.

# Human research participants

Policy information about studies involving human research participants and Sex and Gender in Research.

| Reporting on sex and gender | Participant's sex as self-reported. No sex- and gender-based analyses were performed. |
|---|---|
| Population characteristics | Pt B01 10 years female |

Pt B01 10 years female
Pt B02 9 years female
Pt B03 9 years male
Pt B04 8 years male
PT B05 10 years female
Pt B06 6 years male
Pt B07 4 years female
Pt B08 3 years female
Pt B09 8 years male
Pt B10 10 years female
Pt B11 4 years male
Pt B12 2 years male
Pt B13 15 years male
Pt B14 5 years male
Pt B15 9 years male
Pt B16 8 years female
Pt B17 7 years female
Pt B18 7 years male
Pt B19 10 years male
Pt B20 8 years male
Pt B21 12 years male
Pt B22 17 years male
Pt B23 10 years male
Pt B24 5 years male
Pt B25 8 years male
Pt B26 7 years male
Pt B27 10 years female
Pt B28 14 years male
Pt B29 1 years male
Pt B30 16 years male
Pt B31 9 years female
Pt B32 19 years male
Pt B33 14 years female
Pt B34 2 years male
Pt B35 2 years female
Pt B36 2 years female
Pt B37 15 years male
Pt B38 18 years male
Pt B39 8 years female

| | | |
|---|---|---|
| Pt L01 | 4 | years |
| Pt L02 | 17 | years |
| Pt L03 | 10 | years |
| Pt L04 | 8 | years |
| Pt L05 | 4 | years |
| Pt L06 | 11 | years |
| Pt L07 | 8 | years |
| Pt L08 | 8 | years |
| Pt L09 | 7 | years |
| Pt L10 | 9 | years |
| Pt L11 | 12 | years |

| ID | Sex | Age | Unit |
|---|---|---|---|
| Pt L12 | 9 | | years |
| Pt L13 | 6 | | years |
| Pt L14 | 1 | | years |
| Pt L15 | 10 | | years |
| Pt L16 | 13 | | years |
| Pt L17 | 17 | | years |
| Pt L18 | 11 | | years |
| Pt L19 | 7 | | years |
| Pt L20 | 13 | | years |
| Pt L21 | 3 | | years |
| Pt L22 | 17 | | years |
| Pt L23 | 13 | | years |
| Pt L24 | 5 | | years |
| Pt L25 | 12 | | years |
| Pt L26 | 7 | | years |
| Pt L27 | 10 | | years |
| Pt L28 | 9 | | years |
| Pt L29 | 7 | | years |
| Pt L30 | 12 | | years |
| Pt L31 | 8 | | years |
| Pt L32 | 16 | | years |
| Pt L33 | 8 | | years |
| Pt L34 | 15 | | years |
| Pt L35 | 14 | | years |
| Pt L36 | 9 | | years |
| Pt L37 | 12 | | years |
| Pt L38 | 15 | | years |
| Pt L39 | 6 | | years |
| Pt L40 | 11 | | years |
| Pt L41 | 11 | | years |
| Pt L42 | 9 | | years |
| Pt L43 | 15 | | years |
| Pt L44 | 9 | | years |
| Pt L45 | 5 | | years |
| Pt L46 | 8 | | years |
| Pt L47 | 6 | | years |
| Pt L48 | 9 | | years |
| Pt L49 | 10 | | years |
| Ankara_01 | Male | 5 | years |
| Ankara_02 | Male | 13 | years |
| Ankara_03 | Female | 15 | years |
| Ankara_04 | Male | 13 | years |
| Ankara_05 | Male | 11 | years |
| Ankara_06 | Female | 16 | years |
| Ankara_07 | Male | 13 | years |
| Ankara_08 | Male | 9 | years |
| Ankara_09 | Female | 8 | years |
| Ankara_10 | Male | 11 | years |
| Ankara_11 | Female | 6 | years |
| Ankara_12 | Female | 7 | years |
| Ankara_13 | Male | 11 | years |
| Ankara_14 | Male | 13 | years |
| Ankara_15 | Female | 11 | years |
| Ankara_16 | Female | 1 | years |
| Ankara_17 | Female | 3 | years |
| Ankara_18 | Male | 6 | years |
| Ankara_19 | Female | 2 | years |
| Ankara_20 | Male | 6 | years |
| Boston_1 | male | 1 | years |
| Boston_2 | female | 2 | years |
| Boston_3 | male | 3 | years |
| Boston_4 | male | 4 | years |
| Boston_5 | female | 7 | years |
| Boston_6 | male | 8 | years |
| Boston_7 | male | 10 | years |
| Boston_8 | male | 12 | years |
| Boston_9 | male | 14 | years |
| Chile_01 | Male | 4 | years |
| Chile_02 | Male | 8 | years |
| Chile_03 | Male | 13 | years |
| Chile_04 | Male | 13 | years |
| Chile_05 | Male | 4 | years |
| Chile_06 | Female | 3 | years |
| Chile_07 | Male | 8 | years |
| Chile_08 | Female | 6 | years |
| Chile_09 | Female | 2 | years |

| | | | |
|---|---|---|---|
| Chile_10 | Male | 1 | years |
| Chile_11 | Male | 12 | years |
| Chile_12 | Female | 3 | years |
| Chile_13 | Female | 11 | years |
| Chile_14 | Male | 2 | years |
| Italy_1 | Male | 5 | years |
| Italy_2 | Male | 5 | years |
| COV Pat01 | male | 10 | years |
| COV Pat02 | female | 18 | years |
| COV Pat03 | female | 8 | years |
| COV Pat04 | male | 15 | years |
| COV Pat05 | male | 9 | years |
| COV Pat06 | male | 15 | years |
| COV Pat07 | male | 8 | years |
| COV Pat08 | male | 12 | years |
| COV Pat09 | female | 11 | years |
| COV Pat10 | female | 7 | years |
| COV Pat11 | male | 17 | years |
| COV Pat12 | male | 13 | years |
| COV Pat13 | male | 12 | years |
| COV Pat14 | female | 5 | years |
| COV Pat15 | male | 1 | years |
| COV Pat16 | male | 12 | years |
| COV Pat17 | female | 6 | years |
| COV Pat18 | female | 13 | years |
| COV Pat19 | male | 11 | years |
| COV Pat20 | female | 9 | years |
| COV Pat21 | female | 6 | years |
| COV Pat22 | male | 6 | years |
| COV Pat23 | female | 2 | years |
| COV Pat24 | male | 6 | years |
| COV Pat25 | female | 10 | years |
| COV Pat26 | female | 7 | years |
| COV Pat27 | male | 12 | years |
| COV Pat28 | male | 5 | years |
| COV Pat29 | male | 4 | years |
| COV Pat30 | male | 2 | years |
| COV Pat31 | male | 1 | years |
| COV Pat32 | female | 4 | years |
| COV Pat33 | male | 1 | years |
| COV Pat34 | male | 5 | years |
| COV Pat35 | male | 5 | years |
| COV Pat36 | female | 13 | years |
| COV Pat37 | female | 6 | years |
| COV Pat38 | male | 16 | years |
| COV Pat39 | male | 3 | years |
| COV Pat40 | female | 8 | years |
| COV Pat41 | female | 12 | years |
| COV Pat42 | female | 17 | years |
| COV Pat43 | female | 6 | years |
| COV Pat44 | female | 9 | years |
| COV Pat45 | female | 5 | years |
| COV Pat46 | male | 4 | years |
| COV Pat47 | male | 6 | years |
| COV Pat48 | female | 10 | years |
| COV Pat49 | male | 11 | years |
| COV Pat50 | female | 13 | years |
| COV Pat51 | female | 14 | years |
| COV Pat52 | male | 1 | years |
| COV Pat53 | male | 8 | years |
| COV Pat54 | female | 11 | years |
| COV Pat55 | male | 10 | years |
| COV Pat56 | male | 13 | years |
| COV Pat57 | male | 2 | years |
| COV Pat58 | male | 1 | years |
| COV Pat59 | female | 13 | years |
| COV Pat60 | female | 9 | years |
| COV Pat61 | male | 2 | years |
| COV Pat62 | male | 2 | years |
| COV Pat63 | male | 9 | years |
| COV Pat64 | male | 16 | years |
| COV Pat65 | male | 11 | years |
| COV Pat66 | male | 9 | years |
| COV Pat67 | female | 8 | years |
| COV Pat68 | female | 6 | years |
| COV Pat69 | male | 10 | years |

| | | | |
|---|---|---|---|
| COV Pat70 | female | 10 | years |
| COV Pat71 | female | 2 | years |
| COV Pat72 | female | 8 | years |
| COV Pat73 | male | 11 | years |
| COV Pat74 | male | 9 | years |
| COV Pat75 | female | 16 | years |
| COV Pat76 | male | 6 | years |
| COV Pat77 | male | 5 | years |
| COV Pat78 | male | 7 | years |
| COV Pat79 | female | 11 | years |
| COV Pat80 | female | 13 | years |
| COV Pat81 | male | 12 | years |
| COV Pat82 | male | 10 | years |
| COV Pat83 | male | 6 | years |
| COV Pat84 | female | 12 | years |
| COV Pat85 | male | 10 | years |
| COV Pat86 | female | 5 | years |
| COV Pat87 | female | 6 | years |
| COV Pat88 | male | 12 | years |
| COV Pat89 | female | 10 | years |
| COV Pat90 | male | 7 | years |
| COV Pat91 | female | 15 | years |
| COV Pat92 | male | 11 | years |
| COV Pat93 | female | 8 | years |
| COV Pat94 | male | 11 | years |
| COV Pat95 | female | 6 | years |
| COV Pat96 | female | 13 | years |
| COV Pat97 | female | 2 | years |
| COV Pat98 | female | 4 | years |
| COV Pat99 | female | 6 | years |
| HC Pat01 | female | 7 | years |
| HC Pat02 | male | 16 | years |
| HC Pat04 | male | 9 | years |
| HC Pat05 | male | 9 | years |
| HC Pat06 | female | 8 | years |
| HC Pat07 | female | 2 | years |
| HC Pat08 | female | 10 | years |
| HC Pat09 | female | 6 | years |
| HC Pat10 | male | 9 | years |
| HC Pat11 | female | 9 | years |
| HC Pat12 | male | 7 | years |
| HC Pat13 | female | 5 | years |
| HC Pat14 | female | 7 | years |
| HC Pat15 | female | 3 | years |
| HC Pat16 | male | 10 | years |
| HC Pat17 | female | 10 | years |
| HC Pat18 | male | 6 | years |
| HC Pat19 | female | 6 | years |
| HC Pat20 | female | 15 | years |
| HC Pat21 | female | 15 | years |
| HC Pat22 | female | 5 | years |
| HC Pat23 | male | 10 | years |
| HC Pat24 | male | 6 | years |
| HC Pat25 | male | 14 | years |
| HC Pat26 | female | 8 | years |
| HC Pat27 | female | 3 | years |
| HC Pat28 | male | 15 | years |
| HC Pat29 | male | 9 | years |
| HC Pat30 | female | 9 | years |
| HC Pat30 | male | 9 | years |
| HC Pat31 | female | 4 | years |
| HC Pat32 | male | 9 | years |
| HC Pat33 | female | 3 | years |
| HC Pat34 | female | 6 | years |
| HC Pat35 | female | 17 | years |

Recruitment

All patients and/or their legal guardians treated for MIS-C at the Charité Universitätsmedizin Berlin Department of Pediatric Respiratory Medicine, Immunology and Critical Care Medicine were asked by the treating physicians if they would take part in this study. All that agreed were included in the study. Similarily, 49 patients were recruited in Lyon and 20 patients were recruited in Ankara, Turkey, 2 patients in Turin, Italy and 114 patients in Santiago, Chile. Blood samples were taken from the acute phase of MIS-C and during follow-up visits in the out-patient clinic. Additionally, samples from 36 healthy children, 57 children with asymptomatic or mild SARS-CoV-2 infection and 39 children with a moderate and 2 with a severe infection with SARS-CoV-2 during acute and follow up and 11 children 6 weeks post infection with SARS-CoV-2 were used as controls. The

controls were recruited in Berlin in outpatient practices and in paediatric departments.

| Ethics oversight | This study was approved by the local institutional review boards (IRB) of: the Charité (Pa-COVID-19 and EA2/178/22) (Berlin, Germany), the Comité de Protection des Personnes Sud Méditerranée I, Marseille, France) (ID-RCB: 2020-A01102-37) for the French patients and by the ethical committee of Hospices civils de Lyon (Lyon University Hospitals, France) N° 23_5231 for French controls, by the Hacettepe University Ethical Committee (2021/09-45, Ankara, Turkey) for MIS-C patients from Turkey, by Mass General Brigham IRB 2020P000955 for the patients and controls from Boston, USA and by the Ethics Committee Federico II in Naples as a collaborative study protocol with the NIH in 2020 (158/2020); and by the ethical committee of Clínica Alemana Universidad del Desarrollo (IRB ID: 202098) for the samples from Chile. Written and informed consent was provided by all legal representatives of the patients that participated in this study. |

Note that full information on the approval of the study protocol must also be provided in the manuscript.

# Field-specific reporting

Please select the one below that is the best fit for your research. If you are not sure, read the appropriate sections before making your selection.

☒ Life sciences ☐ Behavioural & social sciences ☐ Ecological, evolutionary & environmental sciences

For a reference copy of the document with all sections, see nature.com/documents/nr-reporting-summary-flat.pdf

# Life sciences study design

All studies must disclose on these points even when the disclosure is negative.

| Sample size | Blood sample size was determined by weight and age of patients included in this study according to "ETHICAL CONSIDERATIONS FOR CLINICAL TRIALS ON MEDICINAL PRODUCTS CONDUCTED WITH THE PAEDIATRIC POPULATION". Number of study participants was determined by the number of patients hospitalised for MIS-C who volunteered to take part in this study. |
| --- | --- |
| Data exclusions | Single-cell sequencing data from one patient was excluded as this patient was diagnosed with Kawasaki-Disease instead of MIS-C. For allocating EBV-positive cells to patients, samples where all viral RNA detected was within lytic cells or less than 50 intact total cells per patient were sequenced, were excluded for testing frequencies of positivity, but were included in the overall analysis. |
| Replication | We report consistent results from the analysis of different subjects. For TGF-β-activity assay two different T-cell donors and 4 different MIS-C patient sera were used. For scRNAseq data 11 MIS-C patients and 4 controls and 3 patients with influenza were used. For T-cell reactivation assay PBMC from 6 different healthy donors with 6 different peptides and treated with serum from 7 different MIS-C patients, or during acute phase and follow up 8 patients with MIS-C with 5 different peptides, or 6 children infected with SARS-CoV-2 with 6 different peptides during the acute phase and during follow-up. For testing of TCRVb21.3 T cells among activated T cells samples from 32 patients (13 Chile, 19 Europe) were used. For the TCR-atlas 5 donors and 2 different peptides and 2 sets of 3 donors with 1 peptide each were used. For EBNA2-peptide reactivation assay 7 different donors in two independet expriments were used. For killing assay of LCL by T cells 4 donors each analyzed twice with blood drawn at different time points were used. For age matched comparison of seroprevalence Age-matched group sizes: MIS-C=62, 64, or 27; control=2573; no MIS-C=75; pCOVID-19 high TGF-β=6 patients or controls were included. For activated B cell/plasmablast datasets (6 healthy/mild COVID-19, 7 Tdap, 15 Comirnaty, 8 Vaxzevria, 32 MIS-C, 12 severe COVID-19 donors were used. |
| Randomization | Serum samples and PBMCs from patients with MIS-C used in this study were randomly used after a preselection of the samples with high enough cell counts or high enough quantities of serum needed for the experiments. |
| Blinding | Blinding was not possible in this study due to the inherent nature of the research and the sample processing workflow. Specifically, the study involved analyzing samples from patients with MIS-C and control individuals, obtained from differnt hospitals, outpatient practices and biobanks, each using different tubes and pseudonyms. To minimize potential bias, the researchers performing bioinformatics analysis or analysis of in vitro studies did not have access to patient information during the experimental procedures. Clinical history and additional patient data were only integrated into the dataset after the analysis to contextualize the results. |

# Reporting for specific materials, systems and methods

We require information from authors about some types of materials, experimental systems and methods used in many studies. Here, indicate whether each material, system or method listed is relevant to your study. If you are not sure if a list item applies to your research, read the appropriate section before selecting a response.

## Materials & experimental systems

| n/a | Involved in the study |
|---|---|
| ☐ | ☒ Antibodies |
| ☐ | ☒ Eukaryotic cell lines |
| ☒ | ☐ Palaeontology and archaeology |
| ☒ | ☐ Animals and other organisms |
| ☒ | ☐ Clinical data |
| ☒ | ☐ Dual use research of concern |

## Methods

| n/a | Involved in the study |
|---|---|
| ☒ | ☐ ChIP-seq |
| ☐ | ☒ Flow cytometry |
| ☒ | ☐ MRI-based neuroimaging |

# Antibodies

**Antibodies used**

Flow cytometry anti-human antibodies:
FITC anti-CD3 (1:100, UCHT1, in house), PerCP-Cy5.5 anti-CD14 (1:50, TÜK4, BioLgend), APC anti-CD38 (1:25, HIT2, Biolegend), APC-Vio770 anti-HLA-DR (1:100, AC122, Miltenyi Biotec), V500 anti-CD19 (1:200, HIB19, BD Biosciences), PE anti-CD27 (1:100, M-T271, BioLegend), CD14 VioBlue (1:200, REA599, Milenyti Biotec), CD19 VioBlue (1:100, REA675, Miltenyi Biotec), CD45Ro BV510 (1:50, UCHL1, BD Biosciences), CD3 PE Cy5 (1:200, UCHT1, BioLegend), CD8a BV650 (1:100, RPA-T8, BioLegend), CD4 PerCPeFlour710 (1:200, SK3, eBioscience), CD69 APC-Cy7 (1:100, FN50, BioLegend), CD154 PE (1:100, REA238, Miltenyi Biotec) CD137 PE Cy7 (1:100, 4B4-1, BioLegend), PE-anti Biotin (1:100, Bio3-18E7, Miltenyi Biotec), CD154 PE (1:40, REA238, Miltenyi Biotec), CD14 VioBlue (1:200, REA599, Milenyti Biotec), CD19 VioBlue (1:100, REA675, Miltenyi Biotec), CD3 PE Cy5 (1:200, UCHT1, BioLegend), CD69 APC (1:100, REA824, Miltenyi Biotec),CD8a BV650 (1:100, RPA-T8, BioLegend), CD4 PerCPeFlour710 (1:200, SK3, eBioscience), TCR Vβ21.3 APCVio770 (1:100, REA894, Miltenyi Biotec), CD107a AlexaFlour647 (1:3000, H4A3, BioLegend), CD3 PE Cy5 (1:200, UCHT1, BioLegend), CD8a BV650 (1:100, RPA-T8, BioLegend), CD4 PerCPeFlour710 (1:200, SK3, eBioscience). active caspase-3 AlexaFlour647 (1:50, BD Bioscience), HLA-DR PerCP-Cy5.5 (1:50, clone L243, BioLegend), CD21 PE-Vio770 (1:75, clone HB5, Miltenyi Biotec), or CD14 AlexaFluor700 (1:500, clone TM1, in house).

CITE-Seq anti-human antibodies:
Hashtag 1, clone LNH-94; 2M2, GTCAACTCTTTAGCG, BioLegend, Cat. 394661; Hashtag 2, clone LNH-94; 2M2, TGATGGCCTATTGGG, BioLegend, Cat. 394663; Hashtag 3, clone LNH-94; 2M2, TTCCGCCTCTCTTTG, BioLegend, Cat. 394665; Hashtag 4, clone LNH-94; 2M2, AGTAAGTTCAGCGTA, BioLegend, Cat. 394667; Hashtag 5, clone LNH-94; 2M2, AAGTATCGTTTCGCA, BioLegend, Cat. 394669; Hashtag 6, clone LNH-94; 2M2, GGTTGCCAGATGTCA, BioLegend, Cat. 394671; Hashtag 7, clone LNH-94; 2M2, TGTCTTTCCTGCCAG, BioLegend, Cat. 394673; Hashtag 8, clone LNH-94; 2M2, CTCCTCTGCAATTAC, BioLegend, Cat. 394675; Hashtag 9, clone LNH-94; 2M2, CAGTAGTCACGGTCA, BioLegend, Cat. 394677; Hashtag 10, clone LNH-94; 2M2, ATTGACCCGCGTTAG, BioLegend, Cat. 394679. All used accodring to manufacturer's reccommended dilutions.

ELISA detection anti-human antibodies:
IgG: HSV1 (Abnova, KA0229), HSV2 (Abnova, KA0231), EBNA1 (abcam, ab108731), CMV (Abnova, KA1452), HHV6 (Abnova, KA1457), Adenovirus (Creative Diagnostics, DEIA2382). All working stock sultions provided and used undiluted.
IgM: HSV1/2 (Abnova, KA4842), EBNA1 (Abnova, KA1449), CMV (Abnova, KA0228), HHV-6 (Creative Diagnostics, DEIABL57), Adenovirus (Creative Diagnostics, DEIA1767) All working stock sultions provided and used undiluted.

In hospital data for virus serology was determined by accredited Immunoblotting Assays for EBV-IgM and –IgG antibodies and enzyme immunoassays by automatic analysis (Liaison, Diasorin and Architect, Abbott).

Neutralising Antibody:
TGF-β1, TGF-β2 and TGF-β3 (50μg/ml, R&D Systems, MAB1835-SP)

antibodies for polyclonal stimulation of T-cells:
Dynabeads™ human T-Activator CD3/CD28 (ThermoFisher)

antibodies for ARTE Assay:
CD40 Antibody, anti-human, pure-functional grade,130-094-133, Miltenyi Biotec
CD28 Antibody, anti-human, pure-functional grade,130-093-375, Miltenyi Biotec

antibodies for size-based protein assays:
SMAD2/3 (clone: D7G7) XP® Rabbit mAb #8685 and Phospho-SMAD2 (Ser465/467)/SMAD3 (Ser423/425) (clone: D27F4) Rabbit mAb #8828 (both 1:100; both Cell Signaling)
anti-β-Tubulin (polyclonal antibody: NB600-936; 1:100; Novus Biologicals)
Anti-Rabbit Secondary HRP Antibody (042-206; Bio-Techne; provided as working stock dilution from manufacturer)

**Validation**

All purchased antibodies were validated by their manufacturers and further in-house testing for flow cytometry antibodies was done. Titration tests using healthy donors were performed prior to experiments to find optimal working dilutions.

Miltenyi Biotec
"All our antibodies are rigorously tested and validated before release. In the application section on the product page, you can find examples of typical performance data. In addition, we provide extended validation data highlighting details of antibody performance, specificity, and fixation compatibility. All antibodies for which any of these datasets are already available will be indicated with the extended validation stamp."
Validation of antibody specificity by:
- Counterstaining

- Knockout of target protein
- Epitope competition assay
- siRNA knockdown
- Stimulation of cells
- Overexpression of target protein
- Binding to purified antigen (latex bead coating)
- Cross-reactivity

BioLegend Flow Cytometry
"Specificity testing of 1-3 target cell types with either single- or multi-color analysis (including positive and negative cell types). Once specificity is confirmed, each new lot must perform with similar intensity to the in-date reference lot. Brightness (MFI) is evaluated from both positive and negative populations.
Each lot product is validated by QC testing with a series of titration dilutions."
BioLegend TotalSeq™ Antibodies
"Bulk lots are tested by PCR and sequencing to confirm the oligonucleotide barcodes. They are also tested by flow cytometry to ensure the antibodies recognize the proper cell populations.
Bottled lots are tested by PCR and sequencing to confirm the oligonucleotide barcodes."

BD Biosciences
"The specificity is confirmed by using multiple applications that may include a combination of flow cytometry, immunofluorescence, immunohistochemistry or western blot to test a combination of primary cells, cell lines or transfectant models."
"Once our research and development (R&D) team completes evaluation of a new product, the developed process is transferred to our manufacturing teams, including Quality Control."
"Quality control testing of new, manufactured lots are performed side-by-side with a previously accepted lot as a control, helping to serve as a reference for comparison and assuring that performance of the new lot is both reliable and consistent."
"Our strict adherence to these guidelines helps ensure that different lots of conjugated reagents are performing consistently."

Cell Signaling Technology antibodies:
"Antibody Validation for Western Blotting
Western blotting remains one of the most common scientific methods for monitoring protein expression in cells or tissue. The accuracy of western blot results relies heavily of the quality of the primary antibody employed in the immunoblotting. Cell Signaling Technology (CST) provides the highest quality primary and secondary antibodies available for western blotting. CST™ antibodies are produced in-house and validated extensively according to a rigorous protocol.

Validation Steps Include
Examination of several cell lines and/or tissues of known expression levels allows accurate determination of species cross-reactivity and verifies specificity.
Treatment of cell lines with growth factors, chemical activators or inhibitors, which induce or inhibit target expression, verifies specificity. Phosphatase treatment confirms phospho-specificity.
The use of siRNA transfection or knockout cell lines verifies target specificity.
Side-by-side comparison of lots to ensures lot-to-lot consistency.
Optimal dilutions and buffers are predetermined, positive and negative cell extracts are specified, and detailed protocols are already optimized, saving valuable time and reagents."

Novus Biologicals Antibodies:
"Antibody Reproducibility Initiative
Novus recognizes the need for highly validated, high quality antibodies in the life sciences community. The research community faces ongoing concerns about data reproducibility and especially the validity of antibody-based assays. A recent article in Nature discusses the variable standards and performance of antibodies and antibody suppliers in the market. Novus is committed to addressing this problem and to helping our customers attain the best possible results with our products.

To that end, we actively seek high quality, highly validated products and provide support to ensure that our customers have the tools to properly validate their own assays. We are also collaborating with several global initiatives that help life science researchers choose antibodies with proven results. Of the five pillars of validation established by these initiatives, genetic knockout validation provides the most reliable control for assessing antibody specificity."

Antibodies for Size-based Protein assay were checked for external validation using the Simple Western Antibody Database. (https://www.bio-techne.com/resources/simple-western-antibody-database).

Abcam EBNA1 ELISA:
"All kit components have been formulated and quality control tested to function successfully as a kit."

Abnova ELSIA kits:
"The manufacturer guarantees the applicability of the kit as a whole. "

Creative Diagnostics ELISA kits:
"Creative Diagnostics is an evolving biotech company providing highly purified protein/ recombinant antigens worldwide. The protein/ recombinant antigens are rigorously tested to meet the research and development demand for excellent quality, uncompromising biological activity at competitive prices."

# Eukaryotic cell lines

Policy information about <u>cell lines and Sex and Gender in Research</u>

| | |
|---|---|
| Cell line source(s) | Lymphoblastic cell line derived from healthy donors included in this study.<br>And HEK293T cells originally obtained from ATCC (CRL-3216 ™) |
| Authentication | By Flowcytometry |
| Mycoplasma contamination | Cells were not tested for mycoplasma contamination during the time of the study. |
| Commonly misidentified lines<br>(See <u>ICLAC</u> register) | no commonly misidentified cell lines were used in this study. |

# Flow Cytometry

## Plots

Confirm that:

☒ The axis labels state the marker and fluorochrome used (e.g. CD4-FITC).

☒ The axis scales are clearly visible. Include numbers along axes only for bottom left plot of group (a 'group' is an analysis of identical markers).

☒ All plots are contour plots with outliers or pseudocolor plots.

☒ A numerical value for number of cells or percentage (with statistics) is provided.

## Methodology

| | |
|---|---|
| Sample preparation | Pereipheral blood samples for cell sorting:<br>PBMC were isolated from peripheral blood by Ficoll-Paque™ PLUS (Cytiva) density gradient centrifugation at room temperature. Cells were either used directly for analysis or stored at −80°C in heat-inactivated foetal bovine serum (FCS; Corning, 35-079-CV) with 10% v/v dimethylsulfoxide before analysis. Serum samples were stored at −80°C before analysis. 2µg/mL actinomycin D was added to the buffer used during the first centrifugation. Enriched cells were incubated with Fc Blocking Reagent (Miltenyi Biotec) following manufacturer's instructions and subsequently stained for 30 min at 4°C with fluorophore-coupled anti-human antibodies and/or fluorophore-coupled proteins. To stop the staining, cells were washed with PBS/1%BSA. DAPI was added before sorting to allow dead cell exclusion. |
| Instrument | Sortings were preformed using a MA900 Multi-Application Cell Sorter (Sony Biotechnology), a BD Ariall (BD Biosciences) or a Cytek Aurora (Cytek Biosciences). Cell counting was performed using a MACSQuant16 flow cytometer (Miltenyi Biotec). Flow cytometry analysis was performed using a MACSQuant16 flow cytometer (Miltenyi Biotec). |
| Software | Flow cytometry data was aquired using the software of a Multi-Application MA900 Cell Sorter (Sony Biotechnology) and MACSQuantify™ Software of the MACSQuant® Analyzer 16 Flow Cytometer (Miltenyi Biotec), or<br>BD FACSDiva™ Software on a BD ARIA II  (BD Biosciences) or  SpectroFlo® on a Cytek Aurora (Cytek Biosciences) machine, and analysed using FlowJo software 10.8.1 (TreeStar). |
| Cell population abundance | Cell population abundance was highly variable among subjects. Sorted population purity was analyzed during post-sorting cell counting using a MACSQuant flow cytometer (Miltenyi Biotec). |
| Gating strategy | For sequencing:<br>cells were identified by FSC-SSC gating and doublets were excluded based on SSC-A vs SSC-H or FSC-A vs FSC-H plots.<br>Activated T-cells were identified as DAPI-, CD14-, CD19-, CD3+, CD38+ and HLA-DRhigh,<br>memory B-cells were identified as DAPI-, CD14-, CD3-, CD19+, CD27+ (and CD38high for plasmablasts),<br>monocytes were identified as DAPI-, CD14+,<br>lineage- HLA-DRhigh cells were identified as DAPI-, CD14-, CD19-, CD3-, and HLA-DRhigh<br>B cells were identified as DAPI-, CD14-, CD3-, CD19+ and CD21+<br>antigen-specific T-cells:<br>Cells were identified by size and granularity in a FSC-vs SSC plot, followed by doublet exclusion in an FSC-A vs. FSC-H plot. Dump (DAPI, CD14 and CD19)+ cells were also excluded. As CD3 is downregulated after T-cell activation (SEB plot in second row), the gate was extended to include CD3low CD45Ro+ cells. CD4+ epitope-specific T-cells were identified as CD69+ CD154 + and CD8+ epitope specific T-cells were identified as CD69+ CD137+. SEB was used as a positive control for correct gating.<br><br>T-cells for co-culture with LCL:<br>Viable (DAPI-) and VioBlue-, CD3+ cells were further sorted according to CD4, CD8 and TCR Vβ21.3 expression<br><br>Viable LCL were defined as CFSE+, fixable viability dye- and active caspase-3- cells |

☒ Tick this box to confirm that a figure exemplifying the gating strategy is provided in the Supplementary Information.

