## [Peer Review File · Nature]

TGFβ links EBV to multisystem inflammatory syndrome in children

Corresponding Author: Dr Mir-Farzin Mashreghi

A version of this paper was originally rejected for publication by Nature, however that decision was reconsidered after appeal by the authors.

Version 0:

Reviewer comments:

Referee #1

(Remarks to the Author)

In this manuscript, Goetzke, Massoud and colleagues investigate the role of TGFβ and EBV in MIS-C. They seek to establish that patients with MIS-C exhibit levels of TGFβ that paralyze EBV-reactive T cells, leading to EBV reactivation that causes the hyperinflammatory phenotype.

The evidence presented are as follows:

1. Patients with MIS-C exhibit levels of TGFβ as high as severe adult COVID patients and a population of activated monocytes (here, cluster 7) and Ki-67+ proliferating T cells; multiple lineages exhibit a transcriptomic pattern of TGFβ response.
2. CD4 and CD8 T cells from MIS-C patients are (modestly) hyporesponsive, a phenotype likely due to TGFβ since it is reproduced by MIS-C and COVID sera in a manner blocked by anti-TGFβ.
3. As noted by others, the authors find expansion of TCRVβ11-2 T cells in MIS-C, noting that this population increases with treatment (Ext Fig 6a, which seems a critical part of the argument and so perhaps should be in the main figures); these cells seem largely to correspond to the activated T cells noted above, suggesting that they may be reacting specifically to something. Multiple alpha chains are also expanded (it is suggested in line 245 that this weighs against the superantigen hypothesis, though since many alpha chains are expanded, I would think the opposite). Using an assay that is difficult to follow from the text or legend (what does it mean “frequency of TRAVs associated to TCRVβ11-2 positive cells?”), it is suggested that the TCRα/β combinations seen in MIS-C echo those from EBV patients, though the numbers of samples is relatively small. (HLA allele distribution is discussed but it is unclear what we should make of this information.) In vitro assays using TCRVβ21.3 + or depleted cells provide evidence that the CD8 and possible CD4 cells participate to kill EBV-infected cells.
4. Evidence for excess EBV in MIS-C patients includes higher seropos rate than according to a pre-pandemic pediatric control (**age distributions of the two populations not provided and are entirely critical here since EBV seropositivity increase with age; it is stated that they are “age-matched” but details are essential**) together with other indicators better assessed by an EBV expert than by this reviewer. TGFβ was applied directly to LCL cells showing induction of the lytic cycle.

Overall, these data are intriguing, but do not seem sufficient to prove the model proposed.

a. Measurements of TGFβ are in patients acutely ill with MIS-C. The model requires that TGFβ excess precedes MIS-C, since the hyperinflammatory response is not TGFβ-mediated but instead represents a reaction to reactivated EBV or CMV. Yet patients with MIS-C have typically recovered from COVID, with often relatively mild disease, and feel generally well before the syndrome strikes.

b. MIS-C has clear epidemiological and clinical features that are not typical of EBV or CMV viremia, such as abdominal pain, rapidly reversible cardiac dysfunction, and association with obesity.

c. The effects of TGFb shown for viral defense are relatively modest – are they sufficient to lead to such an extreme phenotype?

d. Responsive T cells should be truly oligoclonal – that is, they should have the same TCRa/b receptor, not just share the same a/b families. The term oligoclonal is used here in a sense that seems overly loose (i.e. incorrect). To my knowledge, multiple investigators have looked for true oligoclonality and this has not been found.

e. A key point of the argument is that TGFb renders TCRVB11-2 T cells insufficient to contain EBV. While TGFb impairs them somewhat, as expected, it is not shown that they are incompetent. Further, no evidence is present that MIS-C patients actually develop high-titer viremia.

f. Adults with severe COVID have the same TGFb levels and are all but uniformly EBV infected, so the model would seem to predict that they should get MIS-C as well, but MIS-A (while reported) remains very rare.

These issues strike me as serious and reasonably obvious flaws, and it worries me that the authors write the manuscript as though their data confirm their model, when I think they are at best potentially consistent with it.

Additional points:

1. Extended data table 1 lacks demographic information about the patients studied, including age and sex, and if available body mass index and racial/ethnic group. How many samples were obtained pre-treatment, and could the data obtained with these samples be somehow highlighted?
2. Line 161 states that the T cell proliferation indicates activation via the TCR. No reference is provided, and it seems non-obvious that this is the case, given the much higher prevalence of similar cells in HLH not thought to be driven by a T cell antigen (see e.g. PMID: 33512385).
3. Line 239 - it is not clear what high-resolution analysis is being discussed as a potential test.
4. Is the Figure 3c label TRBV11-1 a typo (should it be 11-2)?
5. The manuscript and methods section do not allow me to fully understand what was done for the antigen reactive T cell enrichment assays.
6. Line 252ff - The statements about ankylosing spondylitis assume that HLA-B27 causes disease an antigen presenting molecule. My impression is that this is an old view. HLA-B27-mediated AS-like disease in animals requires neither the B2M chain nor CD8 T cells, suggesting that HLA-B27 may instead cause disease when it mis-fold/mis-traffic. In any case the area remains controversial in a manner not properly reflected in the text
6. Line 258ff – EBV reactivation has a long and undistinguished history of being invoked for every form of chronic, poorly explained syndrome. I strongly suggest caution in legitimating the idea that EBV contributes long COVID until the evidence is stronger.

Referee #2

(Remarks to the Author)

This study from Goetzke and colleagues analyzed pediatric age patients with a post SARS-CoV-2 MIS-C diagnosis in order to better understand the pathogenesis of MIS-C (multisystem inflammatory disease in children). Their studies conclude that SARS-CoV-2 infection in genetically predisposed patients resulted in high levels of TGF- β that contributed to impairment of pre-existing adaptive immunity against viruses especially against EBV. The authors link the development of EBV reactivation to the development of MIS-C.

The authors have performed extensive serum cytokine analyses, single cell RNA sequencing studies on circulating immune cells, TCR repertoire analyses, and some functional studies on oligoclonally expanded T cells in terms of responsiveness to EBV

Over all the studies are convincing though they are some gaps in terms of details that are only to be expected in a human study. How exactly EBV reactivation leads to the symptomatology of MIS-C is not made clear.

1. Evidence is provided not just for serum increases in TGF beta but, on the basis of changes in the transcriptome, also for in vivo responses to TGF-beta by MIS-C patient lymphoid and myeloid cells. Alterations linked to treatment were thoughtfully identified and excluded. These data that suggest that TGF beta induction and activation both do occur in MIS-C patients
2. Evidence is provided for oligoclonal TRVB11-2 + (encoding TCR Vbeta 21.3) cytotoxic CD4+ and CD8+ T cell expansion (with a subset of TCR V alpha genes) in MIS-C patients that continues after treatment, making superantigen driven expansion unlikely.
3. The demonstration that the sorted cytotoxic TCR Vbeta 21.3+ CD8+ and CD4+ T cells were able to more be activated effectively by EBV infected cells as assessed by a degranulation assay using CD107a expression) supports their general hypothesis.

Referee #3

(Remarks to the Author)

The paper by Goetzke et al shows that in children with MIS-C, elevated levels of TGF-beta results in suppressed reactivation of memory T cells and reduced antigen presentation by monocytes leading to an impairment in the control of EBV reactivation. These results provide new mechanistic insights into the potential immunological mechanisms underlying MIS-C. However several concerns must be addressed:

1. A major concern is that throughout the paper, is that MIS-C patient samples were isolated during acute and follow up visits, but the control samples were isolated 6 weeks post SARS-CoV-2 infection. The cytokine response at 6 weeks is likely to be lower than what is observed during acute infection, and this makes the comparison with the MIS-C samples problematic. The authors should compare the response of COVID infected children during the acute phase.

2. What is the sensitivity of the assays for the various cytokines. Type 1 IFN α , IL-17A and several other cytokines are detected at ~1-2 picograms per ml, which seems to be at or below of the threshold of detection of many cytokine assays.

3. In Table 1, please include age and sex of subjects, as well as age and sex of control subjects. Only 11 control subjects are used. The authors should use additional subjects to achieve more robust conclusions.

4. The authors must show directly the functional activity of TGF-beta in serum by culturing sera with primary cells and demonstrating induction of SMAD2 by western blotting. In addition, to demonstrate that this induction is indeed through the TGF β receptor, the authors should culture the serum with a cell line expressing the dominant negative TGF β receptor and demonstrate lack of induction of SMAD2.

5. The authors perform experiments aimed at assessing the impact of hyperinflammation on memory T cells previously generated in response to prior infection with other viruses.

A technical problem lies in the fact the *ex vivo* stimulation of isolated mononuclear cells to assess memory T cell responses uses PBMCs, rather than sorted memory T cells, as described in the methods section. Since, during an acute infection, there is a major expansion of effector T cells and plasmablasts, the frequency of the bulk memory T cell subset in PBMCs will be reduced, and therefore it is not surprising that upon peptide restimulation that there will be lower frequencies of antigen-specific T cells. However, this is simply the effect of reduced frequencies of the bulk memory T cell population in PBMCs, rather than any deleterious effects of TGF-beta on memory T cells.

6. In Figure 2 d - f, the data seems very heterogeneous. For example in Fig 2d, approximately half the children seem to have impaired T cell responses, whilst the others have enhanced T cell responses. This does not support the conclusion that there is impaired reactivation of memory T cells in MIS-C patients.

Also, in this figure it is very difficult to distinguish between the lines representing results with the different antigens. The authors should color code the antigens or separate them out in different graphs.

Version 1:

Reviewer comments:

Referee #1

(Remarks to the Author)

I thank the authors for their responses to my comments.

For their hypothesis about MIS-C to be true, then:

- 1) There must be elevated TGF β levels preceding the onset of inflammation, because inflammation results from TGF β -suppressed EBV reactive T cells.
- 2) As a result, there is massive, uncontrolled EBV infection.
- 3) This EBV expansion triggers an exuberant T cell reaction (despite the T cell suppression that launched it), triggering expansion of EBV-reactive clones and a host reaction that is HLH-like.

I see no convincing evidence for any of these points. Rather they show that:

1. When already sick, patients with MIS-C have TGF β levels like patients sick with COVID.

2. Patients have many activated T cells. To my previous concern that their claim that the CD38/HLADR signature "indicates activation of these T cells via the T cell receptor" (line 176), the authors cite evidence that upregulation of these marker follows from TCR activation – i.e. that activating T cells is sufficient to upregulate the markers. They do not show however

that it is necessary (which it is not, a counterexample being HLH). These markers show simply that T cells have been activated, not how. There is also no direct evidence to suggest these T cells have anything to do with EBV. Aside from inference from MHC data, many of the supportive experiments were performed using cells from healthy controls after in vitro activation and therefore it is not clear how much the findings reflect the mechanism of MIS-C.

3. Many cells show a TGFb signature, which simply follows from #1 and does not provide further evidence that this signature plays a causal role. The TGFb signature is also among many cytokine signatures in MIS-C, as shown by the authors and by previous studies.

4. TNFb suppresses T cell function in vitro (as is known) – again this is expected as a function of #1 and does not show that the suppression plays a causal role.

5. There is expansion of TCRVB21.3+ T cells in many patients, along with non-random expansion of some alpha chains. Cells with this TCRVB are relatively enriched for EBV reactivity. The term “oligoclonal” is used but no sequence evidence is present that these represent expanded individual T cells with particular TCRs – that is what “clonal” denotes, and it cannot be extrapolated from categories of alpha and beta chains. The claim is made that these T cells are “dysfunctional” (line 326) but I am not clear what is meant or what the evidence is. Are they dysfunctionally hyperactivated or dysfunctionally unresponsive? The data to back up this point are in vitro studies of expanded TCRVB21.3T cells and LGL. The connection to MIS-C remains speculative.

6. MIS-C patients exhibit higher seroreactivity against EBV than age-matched controls (71.4% vs. 51%). No n is provided in the Figure 4 legend; the author response indicates that (for another part of the study) they had access to only 9 pre-IVIG samples, which seems consistent with Ext Fig 9. “71.4%” is however not easily derived from 9 samples, nor is age-matching by categories identifiable in Extended Table 10, where many more than 9 children are noted. Differential EBV seropositivity could be interesting supportive data, but it would need to be clear who these children were and how age-matching could be accomplished in their sample set. Note that only 12% of MIS-C patients had EBV in saliva, which again is not compelling evidence of cause. Do the titers reflect low levels of reactivation in the setting of immunosuppression or high levels of seen with primary EBV infection? What about MIS-C patients that are negative for EBV? Do they have similar disease manifestations and expansion of TCRVB21.3+ T cells?

I do not dispute that these datapoints could be connected in a way that is compatible with the authors’ hypothesis, which remains interesting. However, they do not prove it, or even make it especially plausible, in my view. A more likely story is that sick patients upregulate many cytokines, including immunosuppressive ones, in an inadequate attempt to quell inflammation. The resulting immunosuppression causes incidental reactivation of EBV or other herpesviruses in a subset of patients (as the authors note happens commonly in severe adult COVID); this reactivation passes uneventfully, as it does many times during a normal lifetime.

I am grateful to the authors for pointing me toward the fascinating HLA-B27 studies – these had eluded me and I look forward to reading them.

Referee #2

(Remarks to the Author)

I do not have additional comments and am satisfied with the responses

Shiv Pillai

Referee #3

(Remarks to the Author)

The authors have satisfactorily addressed my comments, and the paper is much improved.

Version 2:

Reviewer comments:

Referee #1

(Remarks to the Author)

My understanding of the authors’ claim is as follows: MIS-C reflects TGF-b-mediated viral reactivation, usually but potentially not always from EBV. This face validity of this hypothesis is uncertain, because: (1) MIS-C is a pediatric disease, whereas (as the authors show) TGFb levels are as high in COVID-infected adults, adults are almost all EBV-positive, and late-phase MIS-C in adults is extremely rare; and (2) MIS-C has essentially vanished, whereas COVID and background EBV seropositivity have not. Thus it is important to look at each step of the process to see whether the data overcome a reasonable burden of proof. These, and my comments, are:

1. Patients are infected with COVID.

2. Patients then develop high levels of TGFb, which leads to impairment of T cell responses.

a. Whether this elevation is common or rare is not defined or tested (this is not a criticism and nothing the authors need to address, in my view)

b. Data presented in support of Step 2 are TGFb levels during the acute phase of COVID. Since the proposed mechanism requires high levels of TGFb before MIS-C begins, and TGFb elevation is perhaps not surprising as a manifestation of severe illness, the relevance of these data is uncertain (though arguably suggestive in favor of the authors' thesis). It is understandable that pre-MIS-C samples are not available, yet the lack of information here renders this step of the hypothesis difficult to test. High levels of IgA2 antibodies (Figure 5E) may provide further support, but these are compared against healthy controls rather than children with COVID but no MIS-C, so the finding is of uncertain weight.

3. In patients previously infected with EBV, TGFb elevation results in EBV reactivation.

a. The authors show expansion of a T cell V-beta population that is enriched for responsiveness to peptides from EBV and ability to be activated / and to kill LCL cells (Figures 3, 4, 5). This is proposed as evidence for EBV reactivation. This is a reasonable thought, since once would have expected EBV reactivation to trigger a temporary activation of these cells. I can't help but wonder whether there could be other explanations (e.g. expansion for other reasons of a very large and diverse TCR-bearing population that also happens to have some EBV reactivity, or incidental expansion of this population during EBV reactivation that occurred with acute illness but was unrelated to that illness).

b. 5F shows evidence for EBV genome in a small fraction of B cells and plasmablasts have EBV. This is helpful information; it should be specified in the legend that the control samples were similarly gated/sorted, if this is the case; if not, the comparison is problematic. Of note, such evidence had previously been sought in MIS-C and not identified: <https://pubmed.ncbi.nlm.nih.gov/33891889/>.

c. Figure 13J shows EBV PCR in MIS-C, and is helpful data. These could perhaps be brought into the main figures, for expositional clarity.

d. The most compelling evidence for a connection with EBV is the higher seroprevalence for EBV in MIS-C patients (Fig 5D). Despite the large amount of information assembled, the dataset from which this claim is drawn is relatively small. Every age group of MIS-C is <10 individuals except for the 6-14 age group, which has 38, of which 82% are positive, compared to 62% in two control populations listed in Table 13. However, 6-14 is a large range, during which EBV seropositivity will transition from negative to positive in the majority of people. If the MIS-C patients were skewed older and the controls groups skewed younger, then the observation could reflect epidemiology not biology. This concern could be addressed by providing histograms of the distributions, and/or by performing more precise age matching rather by these very large categories. (It could be that the authors have address this in their response, but frankly this was difficult for me to understand and must be clear and evident in any published manuscript.)

4. EBV reactivation results in a hyperinflammatory response

a. As far as I can tell, no evidence is provided to support this claim, and conceptually there may be a little difficulty reconciling the hyperinflammatory reaction with the idea that TGFb levels were so high that T cells were too impaired to contain the EBV reactivation.

My conclusion from review of the data is that the hypothesis is consistent with the data but not confirmed – it remains possible that EBV reactivation, even where present, is a bystander effect, a possibility that is plausible since it is also seen in severe adult COVID. (Of course it could be that EBV reactivation participates in disease severity in both, an idea suggested in the rebuttal letter, but this is yet another claim that would need to be shown, especially since the phenotypes of MIS-C and severe adult COVID diverge substantially.) If the work is published in this prestigious venue, my hope is that the epidemiological concern with respect to EBV seroprevalence (which make that critical piece of data uninterpretable) are addressed and that the limitations and gaps in the story should be stated clearly.

Point-by-point response to reviewer's comments

We wish to express our sincere appreciation to the three reviewers for their valuable contributions. We think that their input has helped us significantly enhance the quality of our manuscript and thereby strengthened the conclusions made in the primary version. In the following sections, we will address each of their comments with a concise and detailed point-by-point response.

Reviewer comments:

Referee #1 (Remarks to the Author): In this manuscript, Goetzke, Massoud and colleagues investigate the role of TGFb and EBV in MIS-C. They seek to establish that patients with MIS-C exhibit levels of TGFb that paralyze EBV-reactive T cells, leading to EBV reactivation that causes the hyperinflammatory phenotype. The evidence presented are as follows:

- 1. Patients with MIS-C exhibit levels of TGFb as high as severe adult COVID patients and a population of activated monocytes (here, cluster 7) and Ki-67+ proliferating T cells; multiple lineages exhibit a transcriptomic pattern of TGFb response.*
- 2. CD4 and CD8 T cells from MIS-C patients are (modestly) hyporesponsive, a phenotype likely due to TGFb since it is reproduced by MIS-C and COVID sera in a manner blocked by anti-TGFb.*
- 3. As noted by others, the authors find expansion of TCRVB11-2 T cells in MIS-C, noting that this population increases with treatment (Ext Fig 6a, which seems a critical part of the argument and so perhaps should be in the main figures);*

This is a valid point and we included this finding in Fig. 3c in the main manuscript.

these cells seem largely to correspond to the activated T cells noted above, suggesting that they may be reacting specifically to something. Multiple alpha chains are also expanded (it is suggested in line 245 that this weighs against the superantigen hypothesis, though since many alpha chains are expanded, I would think the opposite).

We agree that a superantigen as a driver for TCRVβ21.3⁺ T-cell expansion cannot be ruled out with full certainty. Nonetheless, our data suggests that not all T cells bearing the β-chain exhibit expansion, as evidenced by the differing distribution of TRAVs to TRBV in our MIS-C and control cohorts.

Thus, we redid literature research looking for expansion and predominance of TCRVβ21.3⁺ T cells. We found a study analysing clonotypic architecture of EBV-specific CD4⁺ T cells¹. T cells derived from patients with infectious mononucleosis, show different epitope specific TCR usage compared to HLA-matched healthy controls. Strikingly, the EBNA2 peptide 279–295 (PRSPVTFYNIPPMPLPSSL) showed a profound TCRVβ21.3⁺ repertoire bias (16% if EBNA2₂₇₉₋₂₉₅) (Meckiff et al. figure 5)¹. Others have shown that a subset of this EBNA2 peptide (namely EBNA2 280-290 TVFYNIPPMPL) is highly promiscuous and can be presented effectively on many MHC class II alleles².

Additionally, SARS-CoV-2 spike protein has been shown to have no intrinsic superantigen-like activity, as shown for Jurkat T-cell lines as well as in primary human CD4⁺- and CD8⁺-T cells³. Also, others have shown, that TCRVβ21.3⁺-T cells are not within the SARS-CoV-2-specific T-cell population in MIS-C patients⁴. A different antigen (which yet has to be determined, potentially an EBV-derived superantigen) has to be the trigger.

In our study however, we provide evidence that this T-cell subset can be considered as EBV-specific by showing unbiased clustering of MIS-C patients' TCRVβ21.3⁺ T cells together with EBV-specific TCRVβ21.3⁺ T cells (Fig. 3g) and enhanced killing capacity for EBV infected B cells by TCRVβ21.3⁺ T cells compared to all other T cells that had been depleted of TCRVβ21.3⁺ T cells (Fig. 3i-j). And as mentioned above, others have shown an association between TCRVβ21.3⁺-T cells and an EBNA2-peptide in infectious mononucleosis¹.

Thus, we have included additional references and changed the paragraph in manuscript text (lines 267–268 and 310-312 in the version without marked changes; and in the redline version lines 288-290 and 337-338) accordingly.

Using an assay that is difficult to follow from the text or legend (what does it mean “frequency of TRAVs associated to TRVB11-2 positive cells?”), it is suggested that the TCRa/b combinations seen in MIS-C echo those from EBV patients, though the numbers of samples is relatively small.

We regret any confusion caused by the previous presentation of the experiment involving the enrichment of antigen-reactive T cells within the memory T-cell population reactivated with viral peptide pools. It is important to note that this method allows for the detection of very rare antigen-specific T cells, which can be as rare as 1 cell in 10^7 (memory) T cells⁵. To address this, we have introduced a clear schematic overview of the experimental setup, along with an in-depth analysis, including a UMAP representation depicting 22,344 antigen-reactivated memory T cells specific for EBV, CMV, SARS-CoV-2, or measles peptide pools derived from 17 separate antigen reactive T cell enrichment (ARTE⁵) assays (see Extended Data Fig. 8 in the supplement). It is important to note that we have expanded the dataset to incorporate additional control data and now encompasses additional TCR repertoires from three measles-specific T-cell libraries. For adenovirus peptide-specific memory T cells, we have exclusively conducted TCR sequencing. Furthermore, we have refined the figure legend to provide a more comprehensive description of the experimental procedures.

To generate a dataset of virus-reactive T-cell receptor (TCR) sequences, we utilised ARTE technology⁵ for reactivating and enriching antigen-responsive memory T cells. In brief, we collected blood samples from healthy donors, which were tested positive for a past infection with EBV, CMV, SARS-CoV-2 or AdV, or had a confirmed history of at least one measles vaccination. Subsequently, peripheral blood mononuclear cells (PBMCs) were isolated and incubated with specific peptide pools derived from the aforementioned viruses as specified in the materials section. Within the PBMCs, autologous antigen-presenting cells are thereby loaded with the specific peptides. To co-stimulate T cells, we introduced an activating anti-CD28 antibody. This led to an upregulation of CD69 and CD154 (also known as CD40 ligand) in antigen-reactive T cells. Additionally, we used an anti-CD40 antibody to prevent CD154 internalisation upon activation⁶.

This approach allowed us to sort antigen-specific T cells by magnetic enrichment for CD154, followed by flow sorting of CD154 and CD69 positive CD3 positive cells. We implemented strict gating criteria to minimise the sorting of non-specific T cells (see Extended Data Fig. 8b in the supplement). Subsequently, we conducted single-cell transcriptomics coupled with TCR sequencing at the single-cell level, revealing 22 344 T cells, which are visualised in the UMAP in Extended Data Fig. 8c in the supplement. Among these cells, we identified a *TRVB* in 18 010 cells and successfully sequenced the full TCR from 15 496 cells. All *TRBV11-2*⁺ cells (as shown in Extended Data Fig. 8d in the supplement) were selected for further analysis, as this TCR V beta chain characterises the expanded activated T cells in all MIS-C patients analysed (Fig. 3d).

For each donor and each virus specificity, we determined the usage frequencies of all *TRAV* genes (Extended Data Fig. 8f in the supplement). Since all these cells share the same *TRBV* gene, we referred to them as *TRBV11-2*⁺ T cells. The usage of these *TRAV* genes compared to the children, which did not suffer from MIS-C 6 weeks after infection with SARS-CoV-2 are depicted in the Extended Data Fig. 7e in the supplement. The hierarchical clustering analysis, as demonstrated in Extended Data Fig. 8a in the supplement, involved TCR repertoire data from each donor and every virus-specific TCR library. For this analysis, we specifically focused on the frequency of each *TRAV* associated to the *TRBV11-2*⁺ cells. The clustering was performed using the *r*-package heatmap as described in material and methods section.

To summarise “frequency of *TRAVs* associated to *TRBV11-2*⁺ cells” means the distribution of *TRAVs* associated to *TRVB11-2* for each sample, reflecting both chains of the TCR (changes made in lines 447

and 453 in the version without marked changes; and in the red-line version lines 491 and 498). The distribution of TRAVs associated with TRBVB11-2 observed in more than 22,000 antigen-specific cells indicates a striking resemblance between the TCR repertoires in MIS-C and those in EBV-specific T cells.

(HLA allele distribution is discussed but it is unclear what we should make of this information.)

We added our interpretation of the results to the manuscript aiming to address the following two questions:

1. Is there a difference in HLA-haplotypes that could explain altered TCR-usage?
2. Is there a specific HLA-class I or II is necessary for the expansion of the *TRBV11-2*⁺ T cells?

Our findings suggest that significant differences in HLA-haplotypes do not seem to underlie the disparities in TCR composition. Furthermore, the presence of similarities in HLA-haplotypes enables a valid and meaningful comparison between the control group and MIS-C patients.

In addition, none of the MIS-C patients exhibited a single class I or class II MHC molecule that was universally present among them. The most common HLA-haplotypes were also detected in the control population at high frequencies. This suggests a broader, non-MHC-class subtype-specific expansion of T cells.

Additionally, we specified in the methods section that our controls included healthy donors we used for the generation of antigen specific T cell libraries were included in the control group. The minor differences observed made us neglect MHC-haplotypes for clustering of virus specific T cells and MIS-C specific T cells in figure 3g.

To summarise, we do not observe a clear association between MIS-C and HLA alleles.

In vitro assays using TCRVb21.3 + or depleted cells provide evidence that the CD8 and possible CD4 cells participate to kill EBV-infected cells.

*4. Evidence for excess EBV in MIS-C patients includes higher seropos rate than according to a pre-pandemic pediatric control (**age distributions of the two populations not provided and are entirely critical here since EBV seropositivity increase with age; it is stated that they are "age-matched" but details are essential**) together with other indicators better assessed by an EBV expert than by this reviewer. TGFb was applied directly to LCL cells showing induction of the lytic cycle.*

We included a separate Extended Data Table 10 in the supplement for better transparency, which gives details of the calculations used for the age matching of the EBV-seroprevalence and added details in the materials and methods section. Our patients (n=35 tested for EBV serology) were 1.3 to 19.5 years old (median 8.5 years, IQR 5.7-12.4 years; mean 9.1 years, SD 4.8 years). In brief, patients were grouped in age groups ranging from less than 3 years; 3 to 5 years; 6 to 14 years and 15 to 16 years. We used these age ranges, as they were used in the publication from which we extracted the control group. (Thesis is written in German but relevant features could be translated upon request. The data itself is taken from "Tabelle 2" (table 2) on page 31.).

We additionally tested 35 healthy children during the pandemic (age range: 2.1-17.2 years; median 8.7 years, IQR 6.1-10.0 years; mean: 8.5 years, SD 3.8 years) to assess effects of social distancing and lockdowns on EBV-frequency. No significant difference was observed in EBV frequency between the pre-pandemic and pandemic cohorts (age-matched 34 (adolescent aged 17.2 years old was not matched, as data lacked for this age in the control group) vs 2745; 44.1% EBV-seropositive vs. 55.8% EBV-seropositive; p=0.1172). Secondly, we added a further 99 children who were infected with SARS-CoV-2 and did not develop MIS-C. Comparing children with MIS-C and children infected with SARS-

CoV-2 who did not develop MIS-C showed increased EBV seroprevalence as well (MIS-C: 71.4% vs. Control: 51%; $p=0.0403$). Overall, both EBV and CMV-seroprevalences are significantly higher in MIS-C than in control or in the no MIS-C groups. Together, over 85% of children tested in the MIS-C group were either seropositive for EBV, CMV or both. We incorporated all the mentioned data in the revised manuscript (Fig. 4a, and Extended Data Fig. 9a-d and Extended Data Table 10 in the supplement and lines 350-351 and in the redline version lines 379-380).

Overall, these data are intriguing, but do not seem sufficient to prove the model proposed.

a. Measurements of TGF β are in patients acutely ill with MIS-C. The model requires that TGF β excess precedes MIS-C, since the hyperinflammatory response is not TGF β -mediated but instead represents a reaction to reactivated EBV or CMV. Yet patients with MIS-C have typically recovered from COVID, with often relatively mild disease, and feel generally well before the syndrome strikes.

We appreciate the reviewer's comments and would like to emphasise several key findings in our study. We demonstrate that (i) TGF- β 1 in MIS-C patients induces a general T-cell dysfunction (as shown in Fig. 2d+e); (ii) Viral reactivation is a common occurrence in MIS-C, as evidenced by Extended Data Fig. 9j in the supplement. (iii) This viral reactivation is induced by high levels of TGF- β 1 in MIS-C patient sera, as depicted in Fig. 1b+4b. (iv) Additionally, under normal conditions (i.e., without increased active TGF- β 1), TCRV β 21.3+ T cells from EBV-positive donors exhibit enhanced killing of EBV-infected B-cells, as illustrated in Fig. 3i-j.(v) We also highlight a connection between the EBNA2-peptide 276-295 and TCRV β 21.3+ T cells, as demonstrated by others¹.

Furthermore, TGF- β can be induced by SARS-CoV-2^{8,9} and occurs during the acute phase in severe COVID-19 in adults^{10,11}. Additionally, microbiota¹² can induce TGF- β . Thus, viral persistence in the gut, which others have shown for MIS-C¹³ potentially could also induce high TGF- β 1 levels directly or via alterations of the microbiome.

To further address the reviewer's comment we performed additional experiments to get a better insight into the events leading up to the increased TGF- β 1-levels. To this end, we examined stored samples from 99 children in the Central Biobank Charité who had acute SARS-CoV-2 infections, confirmed by high viral load in nasal or throat swab PCR. Notably, none of these patients developed MIS-C, and to our knowledge, no other biobank holds samples from MIS-C patients during the acute phase of SARS-CoV-2 infection. Unfortunately, this aspect cannot be fully explored experimentally. However, we hardly observed increased TGF- β 1 during the acute phase in paediatric COVID-19, which is especially evident for moderate to severe cases (Fig. 1b).

Thus, paediatric COVID-19 diverges from its adult counterpart in regards to early TGF- β 1 induction. This variance may clarify why T-cell dysfunction and EBV reactivation can manifest in severe COVID-19 cases in adults, while paediatric patients tend to experience a time lag of 4-8 weeks¹⁴ between the initial infection and the onset of T-cell dysfunction and EBV reactivation.

Elevated TGF- β 1 levels were uncommon in paediatric COVID-19, but we identified six patients with TGF- β 1 levels exceeding 250 pg/ml. For these cases, we investigated EBV prevalence, and the results revealed a substantial reduction compared to children who did develop MIS-C (16.7% vs. 72.7%, $p=0.04$ as depicted in Fig. 4a). This suggests that the combination of elevated TGF- β 1 and EBV prevalence is of significance in MIS-C.

We acknowledge the concerns raised by reviewer 1 and, accordingly, have refined our argumentation (lines 79, 267-268 and 383-384 in the version without marked changes; and in the redline version lines 80-83, 288-290 and 422-424). We have also incorporated the possibility of a viral persistence¹³ to our explanation (lines 130 in the version without marked changes; and in the redline version lines 137).

b. MIS-C has clear epidemiological and clinical features that are not typical of EBV or CMV viremia, such as abdominal pain, rapidly reversible cardiac dysfunction, and association with obesity.

We could not detect an association with obesity in our cohort (see Extended Data Table 1 in the supplement), as z-scores for BMI were well within the normal range within our cohort. Likewise, others also could not detect an association of obesity with MIS-C (only 6.4% in a study including 598 MIS-C patients from multiple centres)¹⁵.

However, we do agree with reviewer 1 that the clinical features do not represent EBV or CMV primary infection. Yet, as MIS-C is a hyperinflammatory condition and clinical findings more likely mimics hemophagocytic lymphohistiocytosis (HLH)/ macrophage activation syndrome (MAS) which can be frequently triggered by EBV¹⁶. The symptoms in MIS-C are generally thought to be caused by the hyperinflammation. In HLH, a (genetic or acquired) NK- or T-cell dysfunction¹⁷ induces hyperinflammation mostly after infectious triggers¹⁸. In our model, TGF- β 1 induces an acquired T-cell dysfunction with reactivation of a latent infection with most likely EBV, but potentially also CMV or other latent infections, which then trigger the hyperinflammation. A combination of more than one virus might also influence the clinical manifestations. We modified the text of the revised manuscript accordingly (lines 350-352 and in the redline-version lines 379-381).

c. The effects of TGF β shown for viral defense are relatively modest – are they sufficient to lead to such an extreme phenotype?

In response to this comment, we reanalysed our data and restructured Figure 2 to enhance its interpretability and better highlight the dysfunction induced by TGF- β . Furthermore, we incorporated a new extended data table containing numerical results (Extended Data Table 3). Our findings underscore the extent of T-cell dysfunction. Notably, during the acute phase of MIS-C, we observed that in 13 out of 22 reactivations, the signal for CD8+ memory T cells was either at or below the background level (Fig. 2d). However, at the follow-up stage, these combinations consistently produced a clearly positive signal, as confirmed by additional analysis utilizing CD69 and CD154 (Fig. 2d-f, Extended Data Fig. 6b-c) from the previous dataset. This suggests an absence of specific reactivation in the samples during the acute phase. Similarly, adding MIS-C sera to PBMC from healthy donors resulted in antigen-specific memory T-cell reactivation at or below background in 8 out of 24 reactions (CD154 CD69 analysis on CD8+ memory T cells) (Fig. 2e). The median fold-changes in antigen-specific T-cell reactivity varied from approximately a 2-fold to a 10-fold reduction, whether during the acute phase of MIS-C or upon the addition of MIS-C sera to samples obtained from healthy donors without neutralizing TGF- β .

In response to reviewer 1's additional point 1, we discovered that we mistakenly assumed one patient's sample was taken during active disease, whilst we had only follow-up samples from this one patient. Thus, we had to remove this patient from the analysis (for Fig. 2d).

We want to highlight some aspects of the assay used to put this data into better perspective. Overall, using the ARTE-assay, we can identify very rare antigen-reactive T cells. Using flow cytometry only, the limit of detection usually is at frequencies of 0.01 to 0.1% of memory T cells⁵ given background and limited events recorded by flow cytometry. Our assay specific detection range was 0.04 to 0.05%. Thus, effects seen are expected to appear smaller compared to methods analysing more common cell populations.

For the cells to be defined as antigen-specific, every staining brighter than the negative staining controls of CD69, CD154 and CD137 were considered positive (see also flow-cytometry gating in Extended Data Fig. 6a in the supplement). This makes the method very accurate in detecting these very rare cells and thus we expect the significant results observed to be relevant for virus control.

In our in vitro assays of memory T-cell reactivation, we have demonstrated that the reactivation of virus-reactive T cells is hindered during the acute phase of infection but significantly improves once children have recovered. Furthermore, our experiments involving the targeted blockade of TGF- β in the serum of children with MIS-C and patients with severe COVID-19 revealed that the reactivation of

healthy virus-specific memory T cells is notably dependent on TGF- β . These findings held true for both CD4⁺- and CD8⁺ memory T cells in all instances tested.

However, it is worth noting that for CD69⁺CD137⁺CD8⁺ virus-reactive T cells, the evidence was relatively weaker and not as robust. This could be due to our initial omission of CD137 staining in many samples (Fig. 2e). In contrast, CD154 staining was consistently performed, and it is known to be upregulated on antigen-specific CD8⁺ T cells^{5,6} (and Extended data Fig. 8e in the supplement). We therefore re-analysed all samples for CD154 expression on CD8⁺ memory T cells. In these analyses, we observed a pronounced and statistically significant reduction in specific T cell-activation attributable to TGF- β in the sera of MIS-C patients.

Further evidence is given by the significantly reduced frequencies of switched memory B cells in our single cell RNA sequencing data set (Fig. 1g). The reduced frequencies are highly suggestive of a relevant disruption of T-B-cell interaction *in vivo* (lines 177–181 and 310-312 and in the redline version lines 192-196 and 337-338).

Despite the limitation of functional human studies, these *in vitro* generated stimulation assays, including the cited data, demonstrate that TGF- β is highly capable of inducing T-cell dysfunction that can lead to severe phenotypes.

In addition to the point above, TGF- β expression in the microenvironment of EBV-related malignancies is a well-known immune evasion mechanism¹⁹. By overexpression of a dominant negative TGF- β receptor in T-cells, others could demonstrate an increased anti-tumour activity both in human subjects²⁰, as well as in a humanised mouse model²¹. Also for sepsis, it is well known, that TGF- β dampens immune reactions²². We included the relevant citations and discussion in the manuscript (lines 196 and 330-333 in the version without marked changes; and in the redline version lines 213-214 and 360-363).

d. Responsive T cells should be truly oligoclonal – that is, they should have the same TCR α / β receptor, not just share the same α / β families. The term oligoclonal is used here in a sense that seems overly loose (i.e. incorrect). To my knowledge, multiple investigators have looked for true oligoclonality and this has not been found.

We previously used the term "oligoclonal" to describe the variation in the expansion of T cells with the TCR V-beta chain *TRBV11-2*. This term was intended to convey that not all T cells with this particular TCR V-beta chain had equal expansion, as opposed to a fully "polyclonal" scenario where all T cells would exhibit uniform expansion or exhibit expansion with fewer variations. But as correctly stated by reviewer 1, there is no single clone nor a subset of highly limited clones expanded in MIS-C. We therefore agree that oligoclonal was somewhat imprecise. As a result, we have amended the manuscript to use more precise terms like "TRBV11-2 specific expansion", or "expansion of a subset of TRBV11-2 positive T-cells" as needed.

e. A key point of the argument is that TGF β renders TCRV11-2 T cells insufficient to contain EBV. While TGF β impairs them somewhat, as expected, it is not shown that they are incompetent. Further, no evidence is present that MIS-C patients actually develop high-titer viremia.

As mentioned above, our data clearly show that T cells are dysfunctional (Fig. 2c-e). In addition, we would again highlight that others have shown in *in vivo* settings, that transferring EBV-specific T cells insensitive to TGF- β in EBV-driven malignancies have increased anti-tumour activity in a human study²⁰ and a humanized mouse model when compared to non-manipulated EBV specific T cells²¹. For this, they also tested specific lysis of EBV infected B-cells by primary T cells in the presence of or absence of TGF- β . They could show a profound TGF- β dependent reduction in lysis of EBV transformed B-cells by EBV-specific T cells (Aaron Foster et al. 2008 Figure 2a)²¹. (Lines 330-334 in the version without marked changes; and in the red-line version lines 357-362)

Usually, the conventional method to detect EBV-reactivation involves assessing viral load in patients' saliva, given its superior sensitivity and specificity. A recently published study found that only in roughly 1/3 of patients with a positive PCR from saliva, was there relevant viral DNA in sera²³. To answer this reviewer question, we analysed cell-free plasma (as we expect only high viremia to be detectable in cell-free plasma and because we lacked saliva samples from our patients) from 109 samples. Our cohort consisted of 59 MIS-C patients, 14 paediatric patients with asymptomatic or mild SARS-CoV-2 infection, 27 patients with severe paediatric COVID-19, and 9 paediatric patients with severe viral infections unrelated to SARS-CoV-2. The true positive detection limit of the assay is ≥ 1.6 log copies/ml and is indicated in the graph in the Extended Data Fig. 9j. We could only detect EBV virus load in one severe paediatric COVID-19 patient (3.7%) whilst we could detect EBV in the cell-free sera from 12.5% of MIS-C patients, indicative of a high virus load at least in a subset of the MIS-C patients. No viral load could be detected in children with non-SARS-CoV-2 (and non-EBV) severe viral infections or children with only a mild infection with SARS-CoV-2 (see Extended Data Fig. 9j in the supplement).

Taken together with the reactivation found in whole blood (prevalence 45%) and plasmablasts expressing EBV lytic genes (prevalence 67 % for MIS-C, and 83% for severe COVID-19 in adults (Extended Data Fig. 9j in the supplement), we have evidence for an EBV reactivation in MIS-C. Overall, however, we believe that a high viral load is not a requisite, which is consistent with other hyperinflammatory conditions, such as HLH²⁴.

f. Adults with severe COVID have the same TGF β levels and are all but uniformly EBV infected, so the model would seem to predict that they should get MIS-C as well, but MIS-A (while reported) remains very rare.

Hyperinflammatory conditions such as HLH²⁵ or Still's disease²⁶ are more common in children than in adults. Other hyperinflammatory conditions, like Kawasaki-Disease²⁷ only affect children, which is especially surprising as plenty of evidence suggests infectious triggers for Kawasaki-disease²⁸. The reason for these striking age-specific differences in hyperinflammatory syndromes in children versus adults remains unknown. Additionally, the initial response to viral infection is different. For example, increased expression of pattern recognition receptors in children's upper airway innate immune cells compared to adults, along with a higher level of interferon priming, likely accounts for the overall stronger initial innate antiviral responses in children, potentially explaining the observed differences in immune responses against SARS-CoV-2^{29 30}.

In this regard, we have noticed a significant reactivation of EBV in nearly all adult patients who were hospitalised with severe COVID-19. Specifically, when re-evaluating our previously published dataset³¹, we found that EBV-envelope mRNA was present in 10 out of 12 patients, with approximately one in 2000 plasmablasts affected, as opposed to the absence of EBV-envelope mRNA in all six healthy or mildly affected COVID-19 patients. At the same time, we do not observe EBV reactivation in mildly affected children and only very rarely in severely affected paediatric COVID-19 patients (roughly 4 times less frequent than in MIS-C) (Extended Data Fig. 9j in the supplement). Thus, a primary distinguishing factor between children with MIS-C and adults experiencing severe COVID-19 is the timing of the hyperinflammatory response, which in our dataset can be timely associated with high levels of TGF- β 1 (Fig. 1b). Another explanation might be that as most people have had EBV-infection during youth or young adulthood³², primary EBV-infection and SARS-CoV-2 infection are likely to coincide. As EBV-incubation time is approximately 6 weeks³³, this aligns with the incubation time for MIS-C after SARS-CoV-2 infection¹⁴. Potentially, adults therefore get an earlier reactivation of EBV, already during severe COVID-19 infection (Extended Data Fig. 9j in the supplement), instead of after an incubation period of 4-8 weeks. This discrepancy may however also be attributed to several factors, such as lower initial TGF- β 1 levels during acute SARS-CoV-2 infection in children (as depicted in Fig. 1b) or the persistence of SARS-CoV-2 in children¹³, which might lead to a prolonged TGF- β 1 response culminating in MIS-C. Furthermore, an altered microbiome might play a role as well^{12,34} and is known to differ between children and adults³⁵.

To summarise, we think that adult COVID-19 patients experience immediate hyperinflammation and EBV reactivation, whereas the children due to a differently balanced immune response against SARS-CoV-2 can more frequently develop hyperinflammation timely separated from COVID-19, i.e. MIS-C.

These issues strike me as serious and reasonably obvious flaws, and it worries me that the authors write the manuscript as though their data confirm their model, when I think they are at best potentially consistent with it.

We trust that the revisions made to the manuscript in response to the major points provided by the reviewer have effectively addressed these concerns and we thank the reviewer for pointing us in this direction.

Additional points:

1. Extended data table 1 lacks demographic information about the patients studied, including age and sex, and if available body mass index and racial/ethnic group.

We added BMI z-scores (compared to paediatric percentile curves³⁶) and ethnicity. Previously, age and gender were available in the additional extended data table, listing each patient from the Berlin cohort individually. Now, we have also included these details in Extended Data Table 1 in the supplement for improved accessibility.

How many samples were obtained pre-treatment, and could the data obtained with these samples be somehow highlighted?

In total, we obtained sera from nine patients before treatment, which were used for cytokine studies and testing the effect of TGF- β in patients' sera on cells. Additionally, we had PBMCs from five patients collected prior to initiation of treatment. We employed colour coding to highlight treatment-related information whenever applicable and, when necessary, indicated it in the extended data tables.

2. Line 161 states that the T cell proliferation indicates activation via the TCR. No reference is provided, and it seems non-obvious that this is the case, given the much higher prevalence of similar cells in HLH not thought to be driven by a T cell antigen (see e.g. PMID: 33512385).

As stated above, HLH is caused by NK-cell and/or T-cell dysfunction¹⁷. In this setting, CD38^{bright} and HLA-DR⁺ T cells are very common³⁷. However, as mentioned by Vandana Chaturvedi et. al. "In vivo antigen-activated human T cells have been shown to be best identified by a CD38^{bright}, HLA-DR⁺ phenotype, peaking 1 to 2 weeks after infection or vaccination"^{38,39}. This highlights potential converging mechanisms following different inductors of the T-cell dysfunction, leading to hyperinflammation (see above).

We added references as suggested (lines 175-177 in the version without marked changes; and in the redline version lines 190-192).

3. Line 239 - it is not clear what high-resolution analysis is being discussed as a potential test.

The analysis of TCRV β 21.3 frequencies was not performed on whole blood but was instead focused on CD38⁺ HLA-DR⁺ T cells. This is now clearly stated in the revised text for clarity.

4. Is the Figure 3c label TRBV11-1 a typo (should it be 11-2)?

We thank the reviewer for spotting this typographical error and have corrected it accordingly.

5. The manuscript and methods section do not allow me to fully understand what was done for the antigen reactive T cell enrichment assays.

See response to the third comment above. We hope that amendments to the text, material and methods and adding an Extended Data Fig. 8 in the supplement clarified our approach.

6. Line 252ff - The statements about ankylosing spondylitis assume that HLA-B27 causes disease an antigen presenting molecule. My impression is that this is an old view. HLA-B27-mediated AS-like disease in animals requires neither the B2M chain nor CD8 T cells, suggesting that HLA-B27 may instead cause disease when it mis-fold/mis-traffic. In any case the area remains controversial in a manner not properly reflected in the text

We agree with the reviewer that this area still remains controversial and we added the misfolding/mistrafficking as an argument. However, we politely disagree, that this is an old view as most recently new evidence for the antigen presentation hypothesis has been published in Nature (2022)⁴⁰, in Science (2023)⁴¹ and most recently in a one patient study in Nature Medicine (2023)⁴².

6. Line 258ff – EBV reactivation has a long and undistinguished history of being invoked for every form of chronic, poorly explained syndrome. I strongly suggest caution in legitimating the idea that EBV contributes long COVID until the evidence is stronger.

As suggested, we have taken steps to exercise greater caution in our argumentation (as indicated in lines 287-290). Yet we wish to highlight the study led by Akiko Iwasaki^{43,44}, which we initially cited as a preprint. In this large multiomics study, high titres of antibodies directed against EBV were found to be a hallmark of long COVID. It is important to note that, as of now, there is no established pathological mechanism explaining long COVID, so associations with EBV and other herpesviridae could potentially be coincidental.

Referee #2 (Remarks to the Author):

This study from Goetzke and colleagues analyzed pediatric age patients with a post SARS-CoV-2 MIS-C diagnosis in order to better understand the pathogenesis of MIS-C (multisystem inflammatory disease in children). Their studies conclude that SARS-CoV-2 infection in genetically predisposed patients resulted in high levels of TGF- β that contributed to impairment of pre-existing adaptive immunity against viruses especially against EBV. The authors link the development of EBV reactivation to the development of MIS-C.

The authors have performed extensive serum cytokine analyses, single cell RNA sequencing studies on circulating immune cells, TCR repertoire analyses, and some functional studies on oligoclonally expanded T cells in terms of responsiveness to EBV

Over all the studies are convincing though they are some gaps in terms of details that are only to be expected in a human study. How exactly EBV reactivation leads to the symptomatology of MIS-C is not made clear.

We thank the reviewer for the positive evaluation and agree, that we cannot close all gaps in this human study as we do not have any samples leading up to the symptoms of the patients developing MIS-C after SARS-CoV-2 infection. Nevertheless, we performed substantial additional experiments including recruiting additional control groups to test our hypothesis and strengthen our conclusions. Please also see the answers to the other reviewers on this subject. (Reviewer 1 point 4 (evidence for excess EBV), and comment a and e; and reviewer 3 comment 1).

1. Evidence is provided not just for serum increases in TGF beta but, on the basis of changes in the transcriptome, also for in vivo responses to TGF-beta by MIS-C patient lymphoid and myeloid cells. Alterations linked to treatment were thoughtfully identified and excluded. These data that suggest that TGF beta induction and activation both do occur in MIS-C patients

In response to reviewer 1 and 3, we added further evidence for the role of active TGF- β (we tested induction of SMAD2-phosphorylation in T-cells (Fig. 1c)) and furthermore highlighted treatment in all instances possible for better interpretability of our data.

2. Evidence is provided for oligoclonal TRVB11-2 + (encoding TCR Vbeta 21.3) cytotoxic CD4+ and CD8+ T cell expansion (with a subset of TCR V alpha genes) in MIS-C patients that continues after treatment, making superantigen driven expansion unlikely.

3. The demonstration that the sorted cytotoxic TCR Vbeta 21.3+ CD8+ and CD4+ T cells were able to more be activated effectively by EBV infected cells as assessed by a degranulation assay using CD107a expression) supports their general hypothesis.

Shiv Pillai

In response to reviewer 3, we additionally repeated the CD107a staining experiment for CD4⁺ T cells, as effects on CD4⁺ T cells are weaker and thus initially lacked power for statistical significance. With additional samples analysed we now observe a significantly ($p=0.0391$) higher CD107a positivity in TCRV β 21.3⁺ T cells (see Fig. 3j).

Referee #3 (Remarks to the Author):

The paper by Goetzke et al shows that in children with MIS-C, elevated levels of TGF-beta results in suppressed reactivation of memory T cells and reduced antigen presentation by monocytes leading to an impairment in the control of EBV reactivation. These results provide new mechanistic insights into the potential immunological mechanisms underlying MIS-C. However several concerns must be addressed:

1. A major concern is that throughout the paper, is that MIS-C patient samples were isolated during acute and follow up visits, but the control samples were isolated 6 weeks post SARS-CoV-2 infection. The cytokine response at 6 weeks is likely to be lower than what is observed during acute infection, and this makes the comparison with the MIS-C samples problematic. The authors should compare the response of COVID infected children during the acute phase.

Our rationale of comparing the MIS-C subjects to controls that did not get MIS-C was to be able to account for past and potentially still persisting reactions to SARS-CoV-2^{4,13,45,46}. We understand the additional benefits of also comparing MIS-C with an acute infection to SARS-CoV-2, as especially in adults, hyperinflammation during severe COVID-19 shares some features with MIS-C including the EBV-reactivation and TGF- β -induced T-cell dysfunction (Fig. 2f). Therefore, in response to this comment, we recruited an additional consecutively sampled 99 children with acute SARS-CoV-2 infection (defined by a positive PCR from a throat-swab) and 35 children not infected (SARS-CoV-2 PCR from throat swab negative) as additional controls. Of these patients, 57 had no or only mild symptoms and 42 had moderate to severe symptoms (Extended Data Table 8 in the supplement). As suggested we did compare cytokine levels (Fig. 1b and Extended Data Fig. 1) and found very little differences between healthy children, children infected with SARS-CoV-2 and children 6 weeks after an infection with SARS-CoV-2.

2. What is the sensitivity of the assays for the various cytokines. Type 1 IFN α , IL-17A and several other cytokines are detected at ~1-2 picograms per ml, which seems to be at or below of the threshold of detection of many cytokine assays.

Please find the assay specific lower limit of quantification (LLOQ) for the individual cytokines, chemokines and growth factors in Extended Data Table 9 in the supplement.

We agree with the reviewer that several cytokines and growth factors, including IL-23, IL-1 alpha, and FGF-2, yielded values that were below, close to, or at the lower limit of quantification (LLOQ). Given

the significant potential for noise to overshadow the signal at such low levels, we made the decision to omit the graphs for IL-1 alpha, IL-23, and FGF-2 from the figure since most of their values fell below the LLOQ (as evident below). These cytokines might still be important in MIS-C and/or COVID-19, however, as IL-1 α has a strong paracrine function in inflammation⁴⁷, serum levels might be too far diluted to be properly detectable. For all cytokines, we have included the LLOQ in the graphs to emphasize that, in the case of certain cytokines such as IL-17a; this highlights the fact that most control groups exhibited minimal detectable cytokine levels, while we frequently measured quantifiable amounts in MIS-C sera.

3. In Table 1, please include age and sex of subjects, as well as age and sex of control subjects. Only 11 control subjects are used. The authors should use additional subjects to achieve more robust conclusions.

Age, sex distribution, BMI and ethnicity are now included in the Extended Data Table 1 in the supplement. Individual patient characteristics can be found in Extended Data Table 7 in the supplement.

As suggested, we analysed an additional 99 samples from patients with acute SARS-CoV-2 infection (defined by a positive PCR from a throat-swab) and 35 children not infected (SARS-CoV-2 PCR from throat swab negative) as additional controls to increase the number of patients and the robustness of our data.

4. The authors must show directly the functional activity of TGF-beta in serum by culturing sera with primary cells and demonstrating induction of SMAD2 by western blotting. In addition, to demonstrate that this induction is indeed through the TGF β receptor, the authors should culture the serum with a cell line expressing the dominant negative TGF β receptor and demonstrate lack of induction of SMAD2.

We performed the suggested experiments with human T cells from healthy donors and HEK293T cells transfected with a dominant negative TGFBR2⁴⁸ (lines 517 - 540 in the version without marked changes; and in the red-line version lines 567-590) (see Fig. 1c and Extended Data Fig. 2b-h). Adding MIS-C patients' sera does induce SMAD2 phosphorylation in T cells and HEK293T cells. We added a discussion of these results in lines 140-145. Furthermore, inhibition of TGF- β using a neutralising antibody dampens SMAD2-phosphorylation in T cells and overexpression of a dominant negative TGFBR2 similarly dampens SMAD2-phosphorylation.

A technical problems lies in the fact the ex vivo stimulation of isolated mononuclear cells to assess memory T cell responses uses PBMCs, rather than sorted memory T cells, as described in the methods section. Since, during an acute infection, there is a major expansion of effector T cells and plasmablasts, the frequency of the bulk memory T cell subset in PBMCs will be reduced, and therefore it is not surprising that upon peptide restimulation that there will be lower frequencies of

antigen-specific T cells. However, this is simply the effect reduced frequencies of the bulk memory T cell population in PBMCs, rather than any deleterious effects of TGF-beta on memory T cells.

We want to clarify that we did normalise frequencies of antigen-reactive T cells to total amount of either CD4⁺- or CD8⁺ memory T cells (CD45RO⁺) to avoid misinterpreting relative changes in memory T cell frequencies as changes in frequencies of antigen-specific T cells. We also calculated total numbers of antigen-reactive T cells. Both total counts (see Extended Data Fig. 6b in the supplement) and counts normalised to either CD4⁺- or CD8⁺ memory T cells show T-cell dysfunction during acute MIS-C. We specified the relevant parts in the material methods section (lines 622 and 646 – 653 and in the redline-version lines 682 and 706-713) to avoid this confusion.

We would also like to repeat our response to reviewer 1's comment c on the effects observed:

Overall, using the ARTE-assay, we can identify very rare antigen-reactive T cells. Using only flow cytometry, the limit of detection is usually at frequencies of 0.01 to 0.1% of memory T cells⁵, given background and limited events recorded by flow cytometry, our assay specific detection range was 0.04 to 0.05%. Thus, effects seen are expected to appear smaller compared to methods analysing more common cell populations.

For the cells to be defined as antigen-specific, every staining brighter than the negative staining controls of CD69, CD154 and CD137 were considered positive (see also flow-cytometry gating in Extended Data Fig. 6a). For the cells to be defined as antigen-specific, every staining brighter than the negative staining controls of CD69, CD154 and CD137 were considered positive (see also flow-cytometry gating in Extended Data Fig. 6a). This makes the method very accurate in detecting these very rare cells and thus we expect the significant results observed to be relevant for virus control.

In our *in vitro* assays of memory T-cell reactivation, we have demonstrated that the reactivation of virus-reactive T cells is hindered during the acute phase of infection but significantly improves once children have recovered. Furthermore, our experiments involving the targeted blockade of TGF- β in the serum of children with MIS-C and patients with severe COVID-19 revealed that the reactivation of healthy virus-specific memory T cells is notably dependent on TGF- β . These findings held true for both CD4⁺- and CD8⁺ memory T cells in all instances tested.

However, it's worth noting that for CD69⁺CD137⁺CD8⁺ virus-reactive T cells, the evidence was relatively weaker and not as robust. This could be due to our initial omission of CD137 staining in all samples (Fig. 2e). In contrast, CD154 staining was consistently performed, and it is known to be upregulated on antigen-specific CD8⁺ T cells^{5,6} (and Extended Data Fig. 8e in the supplement). We therefore re-analysed all samples for CD154 expression on CD8⁺ memory T cells. In these analyses, we observed a pronounced and statistically significant reduction in specific T cell reactivation attributable to TGF- β in the sera of MIS-C patients.

Further evidence is given by the significantly reduced frequencies of switched memory B cells in our single cell RNA sequencing data set (Fig. 1g). The reduced frequencies are highly suggestive for a relevant disruption of T-B-cell interaction *in vivo*⁴⁹.

Despite limitation of functional human studies, these *in vitro* generated stimulation assays, including the cited data, demonstrate that TGF- β is highly capable of inducing profound T-cell dysfunction that can lead to extreme phenotypes.

6. In Figure 2 d - f, the data seems very heterogenous. For example in Fig 2d, approximately half the children seem to have impaired T cell responses, whilst the others have enhanced T cell responses. This does not support the conclusion that there is impaired reactivation of memory T cells in MIS-C patients. Also, in this figure it is very difficult to distinguish between the lines representing results with the different antigens. The authors should color coder the antigens or separate them out in different graphs.

Please see also our response to the comment c from reviewer 1 and comment 5 by reviewer 3 above. In brief, TGF- β expression in the microenvironment of EBV-related malignancies is a well-known immune evasion mechanism¹⁹. By overexpression of a dominant negative TGF- β receptor in T-cells, others could demonstrate an increased anti-tumour activity both in human subjects²⁰, as well as in a humanised mouse model²¹. Also for sepsis, it is well known, that TGF- β dampens immune reactions²². We included the relevant citations and discussion in the manuscript (lines 196 and 330-334 in the version without marked changes; and in the redline version lines 212 and 357-362).

Additional, in response to this comment, we have added an Extended Data Table 3 in the supplement for transparency so that different antigens can be better assessed. As indicated above (Reviewer 1 comment c), we had to remove one patient from the analysis (for Fig. 2d), as we wrongly assumed the sampling to be during active disease, whilst we only had sample during follow-ups. Additionally, we reanalysed the data set of CD8⁺ memory T cells, treated with patients' sera (see Fig. 2d-f and Extended Data Fig. 6b in the supplement). As expected in studies with primary cells, there are some biological variations. However, we did find significantly improved T-cell reactivity when blocking only TGF- β in the sera and during acute vs. follow-ups. It is also noteworthy, that all experiments show effects in the same direction. We therefore would like to clarify that our data does not have a half-half distribution, as it is evident from the statistical analysis that we observe a very consistent and significant impaired T-cell response, which as stated above is also biologically relevant. We changed bars from mean to median to better represent the dataset and rearranged the figure for easier comparison between different set ups (Fig. 2d-f and Extended Data Fig. b-c in the supplement). We hope this clarifies the concerns mentioned.

References for rebuttal:

- 1 Meckiff, B. J. *et al.* Primary EBV Infection Induces an Acute Wave of Activated Antigen-Specific Cytotoxic CD4⁺ T Cells. *The Journal of Immunology* **203**, 1276-1287 (2019). <https://doi.org/10.4049/jimmunol.1900377>
- 2 Omiya, R., Buteau, C., Kobayashi, H., Paya, C. V. & Celis, E. Inhibition of EBV-Induced Lymphoproliferation by CD4⁺ T Cells Specific for an MHC Class II Promiscuous Epitope1. *The Journal of Immunology* **169**, 2172-2179 (2002). <https://doi.org/10.4049/jimmunol.169.4.2172>
- 3 Amormino, C. *et al.* SARS-CoV-2 Spike Does Not Possess Intrinsic Superantigen-like Inflammatory Activity. *Cells* **11** (2022). <https://doi.org/10.3390/cells11162526>
- 4 Yonker, L. M. *et al.* Multisystem inflammatory syndrome in children is driven by zonulin-dependent loss of gut mucosal barrier. *The Journal of Clinical Investigation* **131** (2021). <https://doi.org/10.1172/JCI149633>
- 5 Bacher, P. & Scheffold, A. Flow-cytometric analysis of rare antigen-specific T cells. *Cytometry A* **83**, 692-701 (2013). <https://doi.org/10.1002/cyto.a.22317>
- 6 Bacher, P. *et al.* Antigen-Reactive T Cell Enrichment for Direct, High-Resolution Analysis of the Human Naive and Memory Th Cell Repertoire. *The Journal of Immunology* **190**, 3967-3976 (2013). <https://doi.org/10.4049/jimmunol.1202221>
- 7 Beer, A. D. *Die Prävalenz des Epstein-Barr-Virus im Stadt-Land-Gefälle bei Kindern und Jugendlichen bis zum vollendeten 16. Lebensjahr* MD thesis, Charité Universitätsmedizin Berlin, (2017).
- 8 Wang, W. *et al.* SARS-CoV-2 N Protein Induces Acute Kidney Injury via Smad3-Dependent G1 Cell Cycle Arrest Mechanism. *Adv Sci (Weinh)* **9**, e2103248 (2022). <https://doi.org/10.1002/advs.202103248>
- 9 Zhao, X., Nicholls, J. M. & Chen, Y. G. Severe acute respiratory syndrome-associated coronavirus nucleocapsid protein interacts with Smad3 and modulates transforming growth factor-beta signaling. *J Biol Chem* **283**, 3272-3280 (2008). <https://doi.org/10.1074/jbc.M708033200>

- 10 Stefan Frischbutter, P. D., Mario Witkowski, Stefan Angermair, Sascha Treskatsch, Marcus Maurer, Andreas Radbruch, Mir-Farzin Mashreghi. Serum TGF β as a predictive biomarker for severe disease and fatality of COVID-19 *European Journal of Immunology* (2023).
- 11 Witkowski, M. *et al.* Untimely TGF β responses in COVID-19 limit antiviral functions of NK cells. *Nature* **600**, 295-301 (2021). <https://doi.org/10.1038/s41586-021-04142-6>
- 12 Beller, A. *et al.* Specific microbiota enhances intestinal IgA levels by inducing TGF- β in T follicular helper cells of Peyer's patches in mice. *Eur J Immunol* **50**, 783-794 (2020). <https://doi.org/10.1002/eji.201948474>
- 13 Buonsenso, D. *et al.* Viral persistence in children infected with SARS-CoV-2: current evidence and future research strategies. *The Lancet Microbe* **4**, e745-e756 (2023). [https://doi.org/10.1016/S2666-5247\(23\)00115-5](https://doi.org/10.1016/S2666-5247(23)00115-5)
- 14 Belot, A. & Levy-Bruhl, D. Multisystem Inflammatory Syndrome in Children in the United States. *N Engl J Med* **383**, 1793-1794 (2020). <https://doi.org/10.1056/NEJMc2026136>
- 15 Sperotto, F. *et al.* Clinical phenotypes and outcomes in children with multisystem inflammatory syndrome across SARS-CoV-2 variant eras: a multinational study from the 4CE consortium. *eClinicalMedicine* **64** (2023). <https://doi.org/10.1016/j.eclinm.2023.102212>
- 16 Marsh, R. A. Epstein-Barr Virus and Hemophagocytic Lymphohistiocytosis. *Front Immunol* **8**, 1902 (2017). <https://doi.org/10.3389/fimmu.2017.01902>
- 17 Lee, H. *et al.* Natural Killer Cell Function Tests by Flowcytometry-Based Cytotoxicity and IFN- γ Production for the Diagnosis of Adult Hemophagocytic Lymphohistiocytosis. *Int J Mol Sci* **20** (2019). <https://doi.org/10.3390/ijms20215413>
- 18 Griffin, G., Sheno, S. & Hughes, G. C. Hemophagocytic lymphohistiocytosis: An update on pathogenesis, diagnosis, and therapy. *Best Pract Res Clin Rheumatol* **34**, 101515 (2020). <https://doi.org/10.1016/j.berh.2020.101515>
- 19 Tauriello, D. V. F. *et al.* TGF β drives immune evasion in genetically reconstituted colon cancer metastasis. *Nature* **554**, 538-543 (2018). <https://doi.org/10.1038/nature25492>
- 20 Bollard, C. M. *et al.* Tumor-Specific T-Cells Engineered to Overcome Tumor Immune Evasion Induce Clinical Responses in Patients With Relapsed Hodgkin Lymphoma. *J Clin Oncol* **36**, 1128-1139 (2018). <https://doi.org/10.1200/jco.2017.74.3179>
- 21 Foster, A. E. *et al.* Antitumor activity of EBV-specific T lymphocytes transduced with a dominant negative TGF-beta receptor. *J Immunother* **31**, 500-505 (2008). <https://doi.org/10.1097/CJI.0b013e318177092b>
- 22 Döcke, W. D. *et al.* Monocyte deactivation in septic patients: restoration by IFN-gamma treatment. *Nat Med* **3**, 678-681 (1997). <https://doi.org/10.1038/nm0697-678>
- 23 Sufiawati, I. *et al.* Detection of Human Herpesviruses in Sera and Saliva of Asymptomatic HIV-Infected Individuals Using Multiplex RT-PCR DNA Microarray. *Pathogens* **12**, 993 (2023).
- 24 Kelesidis, T. *et al.* Epstein-Barr virus-associated hemophagocytic lymphohistiocytosis in Los Angeles County. *J Med Virol* **84**, 777-785 (2012). <https://doi.org/10.1002/jmv.23267>
- 25 Kuron, D. *et al.* Epidemiology of haemophagocytic lymphohistiocytosis at the population level in Germany. *British Journal of Haematology* **201**, 285-289 (2023). <https://doi.org/https://doi.org/10.1111/bjh.18617>
- 26 Tomaras, S., Goetzke, C. C., Kallinich, T. & Feist, E. Adult-Onset Still's Disease: Clinical Aspects and Therapeutic Approach. *J Clin Med* **10** (2021). <https://doi.org/10.3390/jcm10040733>
- 27 Nakamura, Y. Kawasaki disease: epidemiology and the lessons from it. *International Journal of Rheumatic Diseases* **21**, 16-19 (2018). <https://doi.org/https://doi.org/10.1111/1756-185X.13211>
- 28 Principi, N., Rigante, D. & Esposito, S. The role of infection in Kawasaki syndrome. *J Infect* **67**, 1-10 (2013). <https://doi.org/10.1016/j.jinf.2013.04.004>
- 29 Loske, J. *et al.* Pre-activated antiviral innate immunity in the upper airways controls early SARS-CoV-2 infection in children. *Nature Biotechnology* **40**, 319-324 (2022). <https://doi.org/10.1038/s41587-021-01037-9>
- 30 Yoshida, M. *et al.* Local and systemic responses to SARS-CoV-2 infection in children and adults. *Nature* **602**, 321-327 (2022). <https://doi.org/10.1038/s41586-021-04345-x>

- 31 Ferreira-Gomes, M. *et al.* SARS-CoV-2 in severe COVID-19 induces a TGF- β -dominated chronic immune response that does not target itself. *Nat Commun* **12**, 1961 (2021). <https://doi.org/10.1038/s41467-021-22210-3>
- 32 Fourcade, G. *et al.* Evolution of EBV seroprevalence and primary infection age in a French hospital and a city laboratory network, 2000-2016. *PLoS One* **12**, e0175574 (2017). <https://doi.org/10.1371/journal.pone.0175574>
- 33 Dunmire, S. K., Grimm, J. M., Schmeling, D. O., Balfour, H. H., Jr. & Hogquist, K. A. The Incubation Period of Primary Epstein-Barr Virus Infection: Viral Dynamics and Immunologic Events. *PLoS Pathog* **11**, e1005286 (2015). <https://doi.org/10.1371/journal.ppat.1005286>
- 34 Suskun, C. *et al.* Intestinal microbiota composition of children with infection with severe acute respiratory syndrome coronavirus 2 (SARS-CoV-2) and multisystem inflammatory syndrome (MIS-C). *Eur J Pediatr* **181**, 3175-3191 (2022). <https://doi.org/10.1007/s00431-022-04494-9>
- 35 Radjabzadeh, D. *et al.* Diversity, compositional and functional differences between gut microbiota of children and adults. *Scientific Reports* **10**, 1040 (2020). <https://doi.org/10.1038/s41598-020-57734-z>
- 36 Kromeyer-Hauschild, K. *et al.* Perzentile für den Body-mass-Index für das Kindes- und Jugendalter unter Heranziehung verschiedener deutscher Stichproben. *Monatsschrift Kinderheilkunde* **149**, 807-818 (2001). <https://doi.org/10.1007/s001120170107>
- 37 Chaturvedi, V. *et al.* T-cell activation profiles distinguish hemophagocytic lymphohistiocytosis and early sepsis. *Blood* **137**, 2337-2346 (2021). <https://doi.org/10.1182/blood.2020009499>
- 38 Miller, J. D. *et al.* Human effector and memory CD8+ T cell responses to smallpox and yellow fever vaccines. *Immunity* **28**, 710-722 (2008). <https://doi.org/10.1016/j.immuni.2008.02.020>
- 39 Chandele, A. *et al.* Characterization of Human CD8 T Cell Responses in Dengue Virus-Infected Patients from India. *J Virol* **90**, 11259-11278 (2016). <https://doi.org/10.1128/jvi.01424-16>
- 40 Yang, X. *et al.* Autoimmunity-associated T cell receptors recognize HLA-B*27-bound peptides. *Nature* **612**, 771-777 (2022). <https://doi.org/10.1038/s41586-022-05501-7>
- 41 Zhai, Y. *et al.* Cysteine carboxyethylation generates neoantigens to induce HLA-restricted autoimmunity. *Science* **379**, eabg2482 (2023). <https://doi.org/10.1126/science.abg2482>
- 42 Britanova, O. V. *et al.* Targeted depletion of TRBV9+ T cells as immunotherapy in a patient with ankylosing spondylitis. *Nature Medicine* (2023). <https://doi.org/10.1038/s41591-023-02613-z>
- 43 Klein, J. *et al.* Distinguishing features of Long COVID identified through immune profiling. *medRxiv* (2022). <https://doi.org/10.1101/2022.08.09.22278592>
- 44 Klein, J. *et al.* Distinguishing features of Long COVID identified through immune profiling. *Nature* (2023). <https://doi.org/10.1038/s41586-023-06651-y>
- 45 Brodin, P. Exaggerated responses to a virus long gone. *Science* **379**, 538-539 (2023). <https://doi.org/10.1126/science.adg2776>
- 46 Brodin, P. SARS-CoV-2 infections in children: Understanding diverse outcomes. *Immunity* **55**, 201-209 (2022). <https://doi.org/https://doi.org/10.1016/j.immuni.2022.01.014>
- 47 Malik, A. & Kanneganti, T. D. Function and regulation of IL-1 α in inflammatory diseases and cancer. *Immunol Rev* **281**, 124-137 (2018). <https://doi.org/10.1111/imr.12615>
- 48 Bordignon, P. *et al.* Dualism of FGF and TGF- β Signaling in Heterogeneous Cancer-Associated Fibroblast Activation with ETV1 as a Critical Determinant. *Cell Rep* **28**, 2358-2372.e2356 (2019). <https://doi.org/10.1016/j.celrep.2019.07.092>
- 49 Elsner, R. A. & Shlomchik, M. J. Germinal Center and Extrafollicular B Cell Responses in Vaccination, Immunity, and Autoimmunity. *Immunity* **53**, 1136-1150 (2020). <https://doi.org/10.1016/j.immuni.2020.11.006>

Point-by-point response to referee comments

We would like to thank referees #2 and #3 for their approval of the revised version of the manuscript, and referee #1 for the valuable comments which have helped to still further improve the manuscript. In response to her/his comments, we have now added new data to support our claims further and have adapted our claims. Furthermore, we have included additional 45 patients from **four new centres**: Ankara (Turkey), Boston (USA), Santiago (Chile), and Turin (Italy). Our study now comprises 145 MIS-C patients and 221 paediatric and 134 adult controls from six centres across four continents. These additions have allowed us to reproduce and validate our findings in a geographically diverse patient population. Please find below our point-by-point response to the comments of referee #1.

Referee #1 (Remarks to the Author):

I thank the authors for their responses to my comments.

For their hypothesis about MIS-C to be true, then:

- 1) There must be elevated TGF β levels preceding the onset of inflammation, because inflammation results from TGF β -suppressed EBV reactive T cells.
 - 2) As a result, there is massive, uncontrolled EBV infection.
 - 3) This EBV expansion triggers an exuberant T cell reaction (despite the T cell suppression that launched it), triggering expansion of EBV-reactive clones and a host reaction that is HLH-like.
- I see no convincing evidence for any of these points.

Rather they show that:

1. When already sick, patients with MIS-C have TGF β levels like patients sick with COVID.

Response: Indeed, it is impossible to define the onset of enhanced TGF- β expression in the preclinical phase of MIS-C. However, we have shown that at the time of clinical manifestation, systemic TGF- β expression is elevated significantly, latent EBV is reactivated as evidenced by increased EBV genome copy number, and immunosurveillance of virus-infected cells is blocked, i.e. we observe EBV viremia. For severe COVID-19 in adults, we have shown before the kinetics of TGF- β ¹. Our hypothesis is based on published and cited evidence that (a) SARS-CoV-2 can induce TGF- β production in infected cells, (b) TGF- β can reactivate EBV, (c) that EBV surveillance is based on specific, cytotoxic T cells, (d) that this immunosurveillance is blocked by TGF- β , and (e) that EBV viremia is associated with severe inflammatory pathologies. It is beyond the scope of this manuscript to decipher the mechanism behind EBV-driven inflammatory pathogenesis. However, our hypothesis presents a novel approach by specifically identifying TGF- β and EBV as a therapeutic target to treat MIS-C (and severe COVID-19 in adults).

ad 1) (Hopefully) convincing evidence for elevated TGF- β levels: We do not claim that TGF- β causes the inflammation, but that it aggravates inflammation. Thus, its elevation before onset of inflammation is not relevant, rather its elevation during acute inflammation. Enhanced TGF- β levels during clinical manifestation of MIS-C are shown in Fig. 1B.

ad 2) Evidence for EBV viremia is now shown in the **new Fig. 5F** and **new Extended Data Fig. 13J**. This now shows data from 32 MIS-C patients, as compared to 3 in the original revised version. Evidence that virus-infected cells are not killed by specific T and NK cells in the presence of TGF- β , has been shown

by others and us before^{1,2}. For MIS-C patients, we show that T cell recognition of EBV-infected cells is impaired likewise (Fig. 2D).

ad 3) although we speculate that the EBV viremia induces reactivation of specific T cells, a likely scenario, we report just the expansion of specific T cells in MIS-C patients in the clinical phase of the disease. Our claim that the expanded TCRV β 21.3⁺ T cells are EBV-specific so far had been based on evidence from the literature and the data presented in Fig. 4E and Fig. 5B and C. We now present additional data (**new Fig. 4F**), confirming that TCRV β 21.3⁺ T cells are specific for the EBNA2₂₇₅₋₂₉₄ peptide.

We fully agree with the referee's interpretation of our data that "...patients with MIS-C have TGF[β] levels like [adult] patients [severely] sick with COVID[-19]". Indeed, we extend the MIS-C observation to patients with adult severe COVID-19 and controls with serological as well as functional data from acute symptomatic paediatric COVID-19, paediatric influenza and healthy children for added value.

We have now adapted the graphical abstract to avoid the unnecessary speculation about timing of TGF- β induction in MIS-C.

2. Patients have many activated T cells. To my previous concern that their claim that the CD38/HLADR signature "indicates activation of these T cells via the T cell receptor" (line 176), the authors cite evidence that upregulation of these marker follows from TCR activation – i.e. that activating T cells is sufficient to upregulate the markers. They do not show however that it is necessary (which it is not, a counterexample being HLH). These markers show simply that T cells have been activated, not how. There is also no direct evidence to suggest these T cells have anything to do with EBV. Aside from inference from MHC data, many of the supportive experiments were performed using cells from healthy controls after in vitro activation and therefore it is not clear how much the findings reflect the mechanism of MIS-C.

Response: The pathomechanism of MIS-C addressed here is the immune reaction against EBV, more specific the expansion of EBV specific T cells. As we show, TCRV β 21.3⁺ T cells are significantly expanded and activated (Fig. 3D and **new Fig. 3E**) in MIS-C patients, and we also show now that these T cells are specific for the EBNA2₂₇₅₋₂₉₄ peptide (**new Fig. 4F**). EBNA2 is a major EBV-encoded oncogene that is expressed upon EBV lytic reactivation as well as in newly infected and transformed B-cells. It therefore is an important antiviral T-cell target, the control of which is crucial to prevent the outgrowth of EBV oncogene driven B-cells as well as virion producer cells^{3,4}. We now validated in two independent cohorts from Europe and South America consisting of a total of 32 patients with MIS-C, that TCRV β 21.3⁺ cell frequencies were always higher when analysing CD38⁺ HLA-DR⁺ T cells compared to all T cells (**new Fig. 3E**).

Regarding "activation" it is less important, how the T cells have been activated in the children. We refer to expression of CD38 and HLA-DR as markers of activation. That they are markers of T cell antigen receptor-mediated activation has been suggested by references 5-7 (see below)⁵⁻⁷, including Chaturvedi et al., who state that "*[e]xperimental evidence has suggested that T-cell activation in HLH occurs in response to antigen presentation [...]*"⁸.

3. Many cells show a TGF β signature, which simply follows from #1 and does not provide further evidence that this signature plays a causal role. The TGF β signature is also among many cytokine

signatures in MIS-C, as shown by the authors and by previous studies.

Response: For TGF- β , proof of activity (Fig. 1C) and specification of signatures induced (Fig 2A and Extended Data Table 4), is providing additional evidence, that it is indeed involved in pathogenesis. We show that serum TGF- β in MIS-C (and severe COVID-19 in adults) is active (Fig. 1C + Extended Data Fig. 2B-H, Fig. 2A + Extended Data Fig. 5A-C), that this TGF- β by itself is responsible for the impairment of T-cell recognition of infected cells (Fig. 2E and F and Extended Data Fig. 10B and C), provide additional references to indicate the biological relevance of TGF- β -impaired T-cell cytotoxicity (lines 425-431 in the revised manuscript) and show that TGF- β in the sera of patients can drive EBV reactivation (Extended Data Fig. 13D). As mentioned above, TGF- β is a major trigger for EBV reactivation. Taken together, these functional data are a critical foundation of our concept, that TGF- β in the sera of MIS-C patients is causing the EBV viremia observed.

To further support the distinct role of TGF- β in MIS-C, we have now added an additional inflammatory control group to our MIS-C dataset: 14 consecutive children with acute influenza infection. In these patients, we do not see upregulation of TGF- β , nor do we see a significant EBV viremia (**new Fig. 1B**). Other cytokines are upregulated during influenza infection alike in MIS-C (**new Extended Data Fig. 1**).

Also, proinflammatory imprinting of these cytokines as observed by GSEA on the enriched immune cells was most prominent during influenza infection, whilst TGF- β imprinting was strongest in MIS-C and the only cytokine imprinting that was stronger in MIS-C than in influenza infection in all cell subsets (**new Extended Data Fig. 7 and 8**).

4. TGF β suppresses T cell function in vitro (as is known) – again this is expected as a function of #1 and does not show that the suppression plays a causal role.

Response: In a situation of EBV reactivation, suppression of immunosurveillance is highly likely to enable viremia, and thus in all likelihood TGF- β -mediated suppression of anti-viral T cell cytotoxicity plays a causal role in MIS-C. MIS-C is characterised by the increase of many pro- and anti-inflammatory cytokines, as also revealed in the present manuscript, in Extended Data Fig. 1 and Extended Data Fig. 5. This may also enable expansion of the pool of latently EBV infected B cells, in whom EBNA2 is a major driver of B cell proliferation, expanding the reservoir of cells that can reactivate. Here we demonstrate that despite the increase of other anti- and pro-inflammatory molecules present in the sera of MIS-C patients, it is TGF- β in the serum and its imprint on activated immune cells, which discriminates MIS-C from other inflammatory conditions, but makes it similar to severe COVID-19 in adults. Originally, we show here that serum TGF- β of MIS-C patients activates herpesviruses and inactivates their immunosurveillance. Although individual effects of TGF- β have been described in other situations, their combination and synergy is novel and, as we show here, specific for MIS-C and severe COVID-19 in adults (Fig 2D and F).

Additionally, to show that this effect is specific for MIS-C and does not happen during any inflammation, we have repeated the experiment with PBMC samples from children with acute symptomatic SARS-CoV-2, which do not display high TGF- β levels and samples from the same children after full recovery. In these children no effect of the inflammatory environment on T-cell functionality could be observed, increasing the robustness and specificity of our data showing TGF- β inducing an impaired T cell cytotoxicity T cells in MIS-C (**new Extended Data Fig. 10C and D**).

5. There is expansion of TCRVB21.3+ T cells in many patients, along with non-random expansion of some alpha chains. Cells with this TCRVB are relatively enriched for EBV reactivity. The term “oligoclonal” is used but no sequence evidence is present that these represent expanded individual T cells with particular TCRs – that is what “clonal” denotes, and it cannot be extrapolated from categories of alpha and beta chains. The claim is made that these T cells are “dysfunctional” (line 326) but I am not clear what is meant or what the evidence is. Are they dysfunctionally hyperactivated or dysfunctionally unresponsive? The data to back up this point are in vitro studies of expanded TCRVB21.3T cells and LCL. The connection to MIS-C remains speculative.

Response: Since our data do not show a truly “oligoclonal” expansion, we now have eliminated this term from the manuscript. The term “dysfunctional” here specifically means “compromised cytotoxicity”, as is demonstrated by the data presented. We clarify this in the revised manuscript (line 68, 78, 264, 365, 415, 431 and 515 in the revised manuscript). For patients with MIS-C and those with severe COVID-19 in adults, we obviously had to perform the functional studies on T cell cytotoxicity *ex vivo*, rather than *in vivo*, since the use of TGF- β blocking drugs in those patients is not approved. Nevertheless, the *ex vivo* data point to the molecular mechanism, a deficit in recognizing and docking on to the target cells, with no reason why this should be different *in vivo*, and the observed viremia is reporting a defective immunosurveillance *in vivo* by itself.

6. MIS-C patients exhibit higher seroreactivity against EBV than age-matched controls (71.4% vs. 51%). No n is provided in the Figure 4 legend; the author response indicates that (for another part of the study) they had access to only 9 pre-IVIG samples, which seems consistent with Ext Fig 9. “71.4%” is however not easily derived from 9 samples, nor is age-matching by categories identifiable in Extended Table 10, where many more than 9 children are noted.

Response: To further strengthen this point we collaborated with Prof. Lael Yonker from Boston (USA) and Seza Özen from Ankara (Türkiye) and obtained additional samples of MIS-C. We now have 64 patients from 4 countries and 3 continents enrolled into our serological analysis. Overall, the seroprevalence for EBV in those MIS-C patients is 79.7% (**new Fig. 5D** and **new Extended Data Table 13**).

As a side note, we clearly indicated n-values in the figure itself in the bars and stated “n-values after age-matching and frequencies are depicted in the bar graphs” (line 648 in the revised manuscript) (**new Fig. 5 D**) in the figure legend. We also include the n numbers in the figure legend itself in the revised version of the manuscript (line 753-754 and 1624-1831 in the revised manuscript).

As stated in the manuscript “only samples collected before initiation of IVIG treatment were taken into consideration to complement the routinely tested serology” (line 338-340 of the previous revised version). These samples from the in-hospital data from Berlin and Lyon make up n=26 for EBV and n=23 for CMV. Additionally, we examined EBV and CMV seroprevalence in nine children with MIS-C prior to the start of IVIG treatment in our laboratory. As stated in the material and methods section (lines 718-722 of the previous revised version), the raw data for Extended Data Table 13 (previously, Extended Data Table 10) is in Extended Data Table 8-10 and in reference Beer A. 2017⁹. Additionally, the new serological data from Türkiye and the USA is included in the **new Extended Data Table 11 and 12**.

Age-matching by categories is provided in the new Extended Data Table 13 (old Extended Data Table 10 with added data). The top left box (cells A2-C8 (referring to excel sheet “MIS-C vs controls”)) lists the number of patients with MIS-C with positive or negative EBV-serology binned by age-ranges (0-3 years, 3-5 years, 6-14 years, 15-16 years and additionally 17 and older) predefined by the control group. Individual patients can be identified in Extended Data Table 8-12. The top right box (cells K2-M8) and cells M11 to P17 list the children that we analysed from Extended Data Table 8 and from Beer A. 2017⁹ respectively. These two groups are added bin-wise. The result is listed in cells C14 to E17 next to the n numbers for each bin from the MIS-C patients (cells B14-B17) (irrespective of the result of EBV-serology). In cells F13-H17 the result of the function to match the frequency of patients in each bin for MIS-C and the control group named “healthy” is presented. For cells I14-K17 we rarefied the total patients per bin, so that no patient or control was counted twice, resulting in the final counts listed in I18-K18 which are used for the subsequent statistical analysis.

Differential EBV seropositivity could be interesting supportive data, but it would need to be clear who these children were and how age-matching could be accomplished in their sample set. Note that only 12% of MISC patients had EBV in saliva, which again is not compelling evidence of cause.

Response: Unfortunately, we did not have access to saliva at all. 12% of our MIS-C patient had free EBV-DNA in cell-free blood plasma, but this is significantly less sensitive for EBV reactivation than salivary shedding. For instance, EBV viral load analysis is not licensed for the diagnosis of acute EBV infection/infectious mononucleosis as it is not sufficiently sensitive as serology. We therefore. We consider this a drastic underestimation of EBV viremia, since we also identified lytic EBV-mRNA in B cell plasmablasts at high frequencies (1 UMI per 300 cells) in 4 out of 6 MIS-C patients analysed, and we consider even this an underestimation, since cells in the lytic EBV replication cycle are very fragile and may have escaped our transcriptome analyses, and a range of EBV transcripts are not expressed at levels high enough to likely be captured by single cell approaches. At the level of this sensitivity, we observe a highly significant viremia in MIS-C and severe COVID-19 adult patients, as compared to healthy donors, donors with mild COVID-19 and vaccinated donors (**new Fig. 5E** and **new Extended Data. Fig. 13J**).

Do the titers reflect low levels of reactivation in the setting of immunosuppression or high levels of seen with primary EBV infection?

Response: The EBV titres observed by PCR in whole blood of MIS-C patients more likely resemble the lower titres also observed in immunosuppressed patients, which show virus reactivation with consecutive disease manifestation or worsening including organ dysfunctions¹⁰⁻¹³. Additionally, whenever anti-EBV-antibody titres were quantified for MIS-C patients and controls, titres of all positive samples were compared between the groups. In these tests, we observed higher anti-EBV-antibody-titres in the patients with MIS-C, indicating an activated anti-EBV-immune response (**new Fig. 5E** and **new Extended Data Fig. 13I**). Furthermore, the presence of EBV-specific IgA2 antibodies in all MIS-C patients and only in 1 out of 57 controls, substantiates the role of TGF- β in regulating the immune response to EBV in MIS-C, since TGF- β is the only cytokine able to target class switching to IgA2^{14,15} (**new Fig. 5D**).

What about MIS-C patients that are negative for EBV? Do they have similar disease manifestations and expansion of TCRVB21.3+ T cells?

Response: Due to the rare prevalence of MIS-C and the high prevalence of EBV seropositivity among them, we refrained from further speculation. It just could be mentioned that MIS-C patients negative for EBV are frequently positive for CMV or other herpesviridae (Extended Data Fig. 13 E-H and Extended Data Table 9-11).

I do not dispute that these datapoints could be connected in a way that is compatible with the authors' hypothesis, which remains interesting. However, they do not prove it, or even make it especially plausible, in my view. A more likely story is that sick patients upregulate many cytokines, including immunosuppressive ones, in an inadequate attempt to quell inflammation. The resulting immunosuppression causes incidental reactivation of EBV or other herpesviruses in a subset of patients (as the authors note happens commonly in severe adult COVID); this reactivation passes uneventfully, as it does many times during a normal lifetime.

Response: We sincerely hope that the additional evidence we are providing, as described in detail above, in particular the investigation of additional MIS-C patients from the US, Turkey, Chile, and Italy have made our hypothesis more plausible to the referee. We now confirm the EBV-specificity of the expanded T cells, and show that the condition of MIS-C and severe COVID-19 in adults is distinct from other inflammatory conditions, like severe influenza infection. We report evidence that the pathomechanism behind these severe clinical conditions may be an induction of TGF- β by SARS-CoV-2 in infected cells, which has been documented for severe COVID-19 in adults already¹. TGF- β then reactivates latent EBV (and probably also other herpesviruses) and at the same time blocks the cytotoxic activity of anti-viral T cells. The resulting EBV viremia is significantly associated with a severe inflammatory syndrome, MIS-C and severe COVID-19 in adults. The situation resembles the presumed involvement of EBV in the pathogenesis of Multiple Sclerosis. It is different from other inflammatory conditions, which show a limited reactivation of EBV, and effective immunosurveillance. As a new concept for the pathomechanism of severe COVID-19 in adults and MIS-C, it is of high relevance for the development of new therapies blocking the driving forces, i.e. TGF- β and EBV.

When it comes to the pathogenicity of herpesviruses, here are just a few references: Evidence for EBV as a key factor in development of HLH^{6,7,16,17} and Multiple sclerosis¹⁸⁻²⁰ or other autoimmune diseases^{21,22}, and long COVID²³. EBV is associated with elevated risks for multiple autoimmune diseases, hyperinflammatory diseases and accounts for 1.5% of human malignancies as reviewed elsewhere²⁴⁻²⁷. EBV reactivation in sepsis and other severe infections including COVID-19 has recently been shown to increase morbidity and mortality and prolonged length of intensive care unit stay^{13,28-30}.

I am grateful to the authors for pointing me toward the fascinating HLA-B27 studies – these had eluded me and I look forward to reading them.

Refere #2 (Remarks to the Author):

I do not have additional comments and am satisfied with the responses

Shiv Pillai

We would like to thank Shiv Pillai for this positive evaluation of our manuscript.

Referee #3 (Remarks to the Author):

The authors have satisfactorily addressed my comments, and the paper is much improved.

We would like to thank Referee #3 once more for their helpful comments that guided us to improve the manuscript and are happy to see that they have evaluated our manuscript well.

References:

- 1 Witkowski, M. *et al.* Untimely TGF β responses in COVID-19 limit antiviral functions of NK cells. *Nature* **600**, 295-301 (2021). <https://doi.org/10.1038/s41586-021-04142-6>
- 2 Foster, A. E. *et al.* Antitumor activity of EBV-specific T lymphocytes transduced with a dominant negative TGF-beta receptor. *J Immunother* **31**, 500-505 (2008). <https://doi.org/10.1097/CJI.0b013e318177092b>
- 3 Zhao, B. Epstein-Barr Virus B Cell Growth Transformation: The Nuclear Events. *Viruses* **15** (2023). <https://doi.org/10.3390/v15040832>
- 4 Yuan, J., Cahir-McFarland, E., Zhao, B. & Kieff, E. Virus and cell RNAs expressed during Epstein-Barr virus replication. *J Virol* **80**, 2548-2565 (2006). <https://doi.org/10.1128/jvi.80.5.2548-2565.2006>
- 5 Terrell, C. E. & Jordan, M. B. Perforin deficiency impairs a critical immunoregulatory loop involving murine CD8(+) T cells and dendritic cells. *Blood* **121**, 5184-5191 (2013). <https://doi.org/10.1182/blood-2013-04-495309>
- 6 Gather, R. *et al.* Trigger-dependent differences determine therapeutic outcome in murine primary hemophagocytic lymphohistiocytosis. *Eur J Immunol* **50**, 1770-1782 (2020). <https://doi.org/10.1002/eji.201948123>
- 7 Marsh, R. A. Epstein-Barr Virus and Hemophagocytic Lymphohistiocytosis. *Front Immunol* **8**, 1902 (2017). <https://doi.org/10.3389/fimmu.2017.01902>
- 8 Chaturvedi, V. *et al.* T-cell activation profiles distinguish hemophagocytic lymphohistiocytosis and early sepsis. *Blood* **137**, 2337-2346 (2021). <https://doi.org/10.1182/blood.2020009499>
- 9 Beer, A. D. *Die Prävalenz des Epstein-Barr-Virus im Stadt-Land-Gefälle bei Kindern und Jugendlichen bis zum vollendeten 16. Lebensjahr* MD thesis, Charité Universitätsmedizin Berlin, (2017).
- 10 Al Hamed, R., Bazarbachi, A. H. & Mohty, M. Epstein-Barr virus-related post-transplant lymphoproliferative disease (EBV-PTLD) in the setting of allogeneic stem cell transplantation: a comprehensive review from pathogenesis to forthcoming treatment modalities. *Bone Marrow Transplantation* **55**, 25-39 (2020). <https://doi.org/10.1038/s41409-019-0548-7>
- 11 Glaser, R. *et al.* Stress-related immune suppression: health implications. *Brain Behav Immun* **1**, 7-20 (1987). [https://doi.org/10.1016/0889-1591\(87\)90002-x](https://doi.org/10.1016/0889-1591(87)90002-x)
- 12 Tse, E. & Kwong, Y. L. Epstein Barr virus-associated lymphoproliferative diseases: the virus as a therapeutic target. *Exp Mol Med* **47**, e136 (2015). <https://doi.org/10.1038/emm.2014.102>

- 13 Goh, C. *et al.* Epstein-Barr virus reactivation in sepsis due to community-acquired pneumonia is associated with increased morbidity and an immunosuppressed host transcriptomic endotype. *Scientific Reports* **10**, 9838 (2020). <https://doi.org/10.1038/s41598-020-66713-3>
- 14 Ferreira-Gomes, M. *et al.* SARS-CoV-2 in severe COVID-19 induces a TGF- β -dominated chronic immune response that does not target itself. *Nat Commun* **12**, 1961 (2021). <https://doi.org/10.1038/s41467-021-22210-3>
- 15 van Vlasselaer, P., Punnonen, J. & de Vries, J. E. Transforming growth factor-beta directs IgA switching in human B cells. *J Immunol* **148**, 2062-2067 (1992).
- 16 Kelesidis, T. *et al.* Epstein-Barr virus-associated hemophagocytic lymphohistiocytosis in Los Angeles County. *J Med Virol* **84**, 777-785 (2012). <https://doi.org/10.1002/jmv.23267>
- 17 Griffin, G., Shenoi, S. & Hughes, G. C. Hemophagocytic lymphohistiocytosis: An update on pathogenesis, diagnosis, and therapy. *Best Pract Res Clin Rheumatol* **34**, 101515 (2020). <https://doi.org/10.1016/j.berh.2020.101515>
- 18 Bjornevik, K., Münz, C., Cohen, J. I. & Ascherio, A. Epstein-Barr virus as a leading cause of multiple sclerosis: mechanisms and implications. *Nature Reviews Neurology* **19**, 160-171 (2023). <https://doi.org/10.1038/s41582-023-00775-5>
- 19 Bjornevik, K. *et al.* Longitudinal analysis reveals high prevalence of Epstein-Barr virus associated with multiple sclerosis. *Science* **375**, 296-301 (2022). <https://doi.org/doi:10.1126/science.abj8222>
- 20 Vietzen, H. *et al.* Ineffective control of Epstein-Barr-virus-induced autoimmunity increases the risk for multiple sclerosis. *Cell* **186**, 5705-5718.e5713 (2023). <https://doi.org/10.1016/j.cell.2023.11.015>
- 21 Houen, G. & Trier, N. H. Epstein-Barr Virus and Systemic Autoimmune Diseases. *Front Immunol* **11**, 587380 (2020). <https://doi.org/10.3389/fimmu.2020.587380>
- 22 Nagata, K. *et al.* Reactivation of persistent Epstein-Barr virus (EBV) causes secretion of thyrotropin receptor antibodies (TRABs) in EBV-infected B lymphocytes with TRABs on their surface. *Autoimmunity* **48**, 328-335 (2015). <https://doi.org/10.3109/08916934.2015.1022163>
- 23 Klein, J. *et al.* Distinguishing features of Long COVID identified through immune profiling. *Nature* (2023). <https://doi.org/10.1038/s41586-023-06651-y>
- 24 Cao, Y. *et al.* Targeting the signaling in Epstein-Barr virus-associated diseases: mechanism, regulation, and clinical study. *Signal Transduction and Targeted Therapy* **6**, 15 (2021). <https://doi.org/10.1038/s41392-020-00376-4>
- 25 Damania, B., Kenney, S. C. & Raab-Traub, N. Epstein-Barr virus: Biology and clinical disease. *Cell* **185**, 3652-3670 (2022). <https://doi.org/10.1016/j.cell.2022.08.026>
- 26 van Esser, J. W. *et al.* Epstein-Barr virus (EBV) reactivation is a frequent event after allogeneic stem cell transplantation (SCT) and quantitatively predicts EBV-lymphoproliferative disease following T-cell-depleted SCT. *Blood* **98**, 972-978 (2001). <https://doi.org/10.1182/blood.v98.4.972>
- 27 Al Hamed, R., Bazarbachi, A. H. & Mohty, M. Epstein-Barr virus-related post-transplant lymphoproliferative disease (EBV-PTLD) in the setting of allogeneic stem cell transplantation: a comprehensive review from pathogenesis to forthcoming treatment modalities. *Bone Marrow Transplant* **55**, 25-39 (2020). <https://doi.org/10.1038/s41409-019-0548-7>
- 28 Manoharan, S. & Ying, L. Y. Epstein Barr Virus Reactivation during COVID-19 Hospitalization Significantly Increased Mortality/Death in SARS-CoV-2(+)/EBV(+) than SARS-CoV-2(+)/EBV(-) Patients: A Comparative Meta-Analysis. *Int J Clin Pract* **2023**, 1068000 (2023). <https://doi.org/10.1155/2023/1068000>
- 29 Ong, D. S. Y. *et al.* Epidemiology of Multiple Herpes Viremia in Previously Immunocompetent Patients With Septic Shock. *Clin Infect Dis* **64**, 1204-1210 (2017). <https://doi.org/10.1093/cid/cix120>
- 30 Kim, S. Y., Ryu, I. S., Baek, S. H., Chung, K. S. & Koh, H. Concurrent reactivation of latent EBV with hepatitis A can affect clinical feature of childhood hepatitis. *Acta Paediatr* **99**, 1258-1262 (2010). <https://doi.org/10.1111/j.1651-2227.2010.01752.x>

We once more wish to express our sincere gratitude to the referees for their valuable arguments. Their input has helped us to enhance the quality of our manuscript again and thereby has strengthened the conclusions made.

My understanding of the authors' claim is as follows: MIS-C reflects TGF- β -mediated viral reactivation, usually but potentially not always from EBV. The face validity of this hypothesis is uncertain, because: (1) MIS-C is a pediatric disease, whereas (as the authors show) TGF β levels are as high in COVID-infected adults, adults are almost all EBV-positive, and late-phase MIS-C in adults is extremely rare; and (2) MIS-C has essentially vanished, whereas COVID and background EBV seropositivity have not.

Ad (1): The manuscript provides conclusive evidence that in MIS-C and severe COVID-19 TGF- β induced by SARS-CoV-2 in infected cells reactivated latent Herpesviruses and suppresses anti-viral immunosurveillance by cytotoxic T cells. The result is "hyperinflammation", as manifested in MIS-C and severe COVID-19. The reason for age-specific differences in hyperinflammatory syndromes in children versus adults remains unknown and is beyond the scope of the present manuscript.

Ad (2): while COVID and EBV are persisting, severe COVID-19 and MIS-C have become less abundant, probably due to vaccinations¹ and/or evolution of SARS-CoV-2 (<https://covid.cdc.gov/covid-data-tracker/#datatracker-home>). Nonetheless, new generations of unvaccinated children and new variants of SARS-CoV-2 might change that picture again, and it should be noted that there has been pre-pandemic MIS-C, obviously independent of SARS-CoV-2 infections², suggesting that other viruses have the potential to trigger MIS-C, and they may use the same mechanism.

Thus it is important to look at each step of the process to see whether the data overcome a reasonable burden of proof. These, and my comments, are:

1. Patients are infected with COVID.

This is clear from the data provided, but at the time of MIS-C the acute SARS-CoV-2 infection had been fully resolved.

2. Patients then develop high levels of TGF β , which leads to impairment of T cell responses.

This is definitely demonstrated in the manuscript.

a. Whether this elevation is common or rare is not defined or tested (this is not a criticism and nothing the authors need to address, in my view)

We tested TGF- β serum levels in healthy children, children acutely infected with SARS-CoV-2 and children 6 weeks after an infection with SARS-CoV-2, and children infected with Influenza virus. In all these groups, elevated TGF- β levels were rare. See Figure 1b of the manuscript.

b. Data presented in support of Step 2 are TGF β levels during the acute phase of COVID. Since the proposed mechanism requires high levels of TGF β before MIS-C begins, and TGF β elevation is perhaps not surprising as a manifestation of severe illness, the relevance of these data is uncertain (though arguably suggestive in favor of the authors' thesis). It is understandable that pre-MIS-C samples are not available, yet the lack of information here renders this step of the hypothesis difficult to test. High levels of IgA2 antibodies (Figure 5E) may provide further support, but these are compared against healthy controls rather than children with COVID but no MIS-C, so the finding is of uncertain weight.

We have now introduced the new figure 5f demonstrating that children with COVID but no MIS-C have significantly lower anti-EBV IgA1 levels than children with COVID and MIS-C, i.e. exactly the control asked for. We excuse for having mislabelled earlier the figure as showing IgA2, and have now corrected it, showing IgA1 levels. Antibody class switching to both IgA1 and IgA2 is dependent on TGF- β ³.

3. In patients previously infected with EBV, TGFb elevation results in EBV reactivation.
 - a. The authors show expansion of a T cell V-beta population that is enriched for responsiveness to peptides from EBV and ability to be activated / and to kill LCL cells (Figures 3, 4, 5). This is proposed as evidence for EBV reactivation. This is a reasonable thought, since once would have expected EBV reactivation to trigger a temporary activation of these cells. I can't help but wonder whether there could be other explanations (e.g. expansion for other reasons of a very large and diverse TCR-bearing population that also happens to have some EBV reactivity, or incidental expansion of this population during EBV reactivation that occurred with acute illness but was unrelated to that illness).

The claim that latent EBV is reactivated in EBV+ MIS-C and severe COVID-19 patients is central to the manuscript and based on two observations: (1) the detection of EBV transcripts in newly generated plasmablasts (see below), and (2) the expansion of EBV-reactive T cells. With respect to the expansion of T cells, we show in Figure 3 the selective expansion of TCRV β 21.3⁺ T cells, in Figure 4 their reaction to the EBV-nuclear-antigen-2 (EBNA2), and in Figure 5 their ability to kill EBV-infected B cells (in the absence of TGF- β).

- b. 5F shows evidence for EBV genome in a small fraction of B cells and plasmablasts have EBV. This is helpful information; it should be specified in the legend that the control samples were similarly gated/sorted, if this is the case; if not, the comparison is problematic. Of note, such evidence had previously been sought in MIS-C and not identified: <https://pubmed.ncbi.nlm.nih.gov/33891889/>.

Thank you for pointing this out. We have sorted and physically isolated CD19⁺, CD27⁺ and CD38⁺ cells (memory B cells and plasmablasts) in all groups of donors analysed in the same way (gating strategy shown in Extended data Figure 9a), capturing in total 122.964 of activated B cells/plasmablasts for single-cell analysis (Extended data Figure 9b). The same cells were analysed from the datasets of Ferreira-Gomes et al. (severe COVID-19 in adults cohort)⁴ and Ferreira-Gomes et al. (vaccination cohorts)⁵. The same individual performed the gating and cytometric sorting, using identical equipment across all three studies. Thus, we provide consistent and comparable data across all cohorts, as shown in Figure 5g. We could include this clarification in the figure legend.

Regarding the argument that EBV transcripts had not really been detected in a previous study by Ramaswamy et al.⁶, we argue that their analysis underestimates EBV reactivation and T cell activation due to three technical limitations:

- (1) EBV transcripts are expressed mostly in plasmablasts and activated B cells, which in their analysis account for approx. 2 % of the PBMC they analysed, a total of 860 activated B cells/plasma blasts, compared to the 122.964 activated B cells/plasmablasts in the present analysis. Of those, less than 1% express EBV transcripts. If analysed on the level of PBMC, these cells become rare events, and the 4 EBV expressing cells in the analysis of Ramaswamy and colleagues did not reach significance.
- (2) Given that up to 60% of transcripts in plasmablasts and plasma cells encode immunoglobulins⁵, we sequenced each cell to a depth of at least 50,000 reads, which is twice the level recommended by the manufacturer. This approach enhances sensitivity for detecting transcripts other than those encoding

antibodies, such as EBV transcripts. Ramaswamy and colleagues achieved a sequencing depth varying from 12000 to 63000 reads per cell for the different patients they analysed.

(3) The antiviral gene expression signatures used by Ramaswamy and colleagues are mainly based on type 1 interferon induced genes of myeloid cells and neutrophils (their Fig. 2D). In contrast, we focus on activation of EBV-specific T cells, overall, a rare population among PBMC. Our data consistently demonstrate their activation and proliferation, but also their lack of cytotoxicity in MIS-C.

In summary, we believe that the low detection rate of EBV-transformed cells in the study by Ramaswamy and colleagues, compared to our findings, reflects technical differences, as our analysis specifically focuses on activated B and T cells.

c. Figure 13J shows EBV PCR in MIS-C, and is helpful data. These could perhaps be brought into the main figures, for expositional clarity.

We included this figure into the main figure 5h and 5i.

d. The most compelling evidence for a connection with EBV is the higher seroprevalence for EBV in MIS-C patients (Fig 5D). Despite the large amount of information assembled, the dataset from which this claim is drawn is relatively small. Every age group of MIS-C is <10 individuals except for the 6-14 age group, which has 38, of which 82% are positive, compared to 62% in two control populations listed in Table 13. However, 6-14 is a large range, during which EBV seropositivity will transition from negative to positive in the majority of people. If the MIS-C patients were skewed older and the controls groups skewed younger, then the observation could reflect epidemiology not biology. This concern could be addressed by providing histograms of the distributions, and/or by performing more precise age matching rather by these very large categories. (It could be that the authors have address this in their response, but frankly this was difficult for me to understand and must be clear and evident in any published manuscript.)

Initially, we had adjusted the age matching according to Beer et al. (2017), to compare our data with their cohort of 2,573 children. The age groups used in that publication were 0–3, 3–5, 6–14, 15–16, and 17+. Unfortunately, the raw data from this cohort are not accessible for further detailed age-matching analysis.

To address the referee's concern, we have now generated a histogram illustrating the age distribution in our MIS-C cohort and the corresponding no-MIS-C control group, i.e. children who recovered from COVID-19 but did not develop MIS-C. The figure shows that the age distributions are comparable (Figure 5e). Interestingly, the median age of MIS-C children is a year lower than for the control "no MIS-C" group. This observation is further highlighted in the violin plot (Extended data 8m) focussing on the 6–14 years age group. The plot shows a median age of 9.05 years in the MIS-C group compared to 10.30 years in the no-MIS-C group. This difference is statistically not significant. We thus do not observe a skewing towards older age in the MIS-C group. We included these graphs into the manuscript (new Figure 5e and new Extended Data Figure 8m).

Overall, we have serological data of 64 MIS-C patients, 99 no MIS-C controls and 2573 prepandemic subjects. These are binned to age groups as described above. As the referee correctly noted, the n-values for children aged 0–3 and 3–5 are indeed in both cases 9 subjects each and children aged 15 and older include 8 subjects. However, this reflects the typical age distribution of MIS-C, which predominantly affects children aged 7–10 years. As a result, we have 38 cases within the 6–14 years age group.

4. EBV reactivation results in a hyperinflammatory response

a. As far as I can tell, no evidence is provided to support this claim, and conceptually there may be a little difficulty reconciling the hyperinflammatory reaction with the idea that TGF β levels were so high that T cells were too impaired to contain the EBV reactivation.

The concept that impaired immunosurveillance leads to hyperinflammation has already been demonstrated for several other pediatric hyperinflammatory syndromes, based on genetic immunodeficiencies, including ZAP70 deficiency⁷, 4-1BB deficiency⁸, and Hemophagocytic Lymphohistiocytosis (HLH)⁹.

In EBV associated hyperinflammatory conditions, hyperinflammatory cytokine environments coexist with profound immune dysregulation and impaired T-cell functionality. Whilst in these patients T cell dysfunction is due to inborn errors of immunity, in MIS-C we show that this is a temporary T cell dysfunction driven by a similar autoinflammatory mechanism.

We can further clarify this point in the revised manuscript by adding a discussion of these examples to support our argument if necessary.

My conclusion from review of the data is that the hypothesis is consistent with the data but not confirmed – it remains possible that EBV reactivation, even where present, is a bystander effect, a possibility that is plausible since it is also seen in severe adult COVID. (Of course it could be that EBV reactivation participates in disease severity in both, an idea suggested in the rebuttal letter, but this is yet another claim that would need to be shown, especially since the phenotypes of MIS-C and severe adult COVID diverge substantially.) If the work is published in this prestigious venue, my hope is that the epidemiological concern with respect to EBV seroprevalence (which make that critical piece of data uninterpretable) are addressed and that the limitations and gaps in the story should be stated clearly.

We strongly believe that hypotheses cannot be proven but only be disproven, we hope that the hypothesis underpinned by our data has now become clear, namely that MIS-C and severe COVID-19 are due to induction of TGF- β , which (1) reactivates latent herpesviridae, and (2) suppresses anti-viral immunosurveillance, leading to hyperinflammation. We have formulated the limitations of the study in the revised manuscript, in particular the concern with respect to EBV seroprevalence.

References:

- 1 Levy, M. *et al.* Multisystem Inflammatory Syndrome in Children by COVID-19 Vaccination Status of Adolescents in France. *JAMA* **327**, 281-283 (2022). <https://doi.org/10.1001/jama.2021.23262>
- 2 Benezech, S. *et al.* Pre-Covid-19, SARS-CoV-2–Negative Multisystem Inflammatory Syndrome in Children. *New England Journal of Medicine* **389**, 2105-2107 (2023). <https://doi.org/10.1056/NEJMc2307574>
- 3 van Vlasselaer, P., Punnonen, J. & de Vries, J. E. Transforming growth factor-beta directs IgA switching in human B cells. *J Immunol* **148**, 2062-2067 (1992).
- 4 Ferreira-Gomes, M. *et al.* SARS-CoV-2 in severe COVID-19 induces a TGF- β -dominated chronic immune response that does not target itself. *Nat Commun* **12**, 1961 (2021). <https://doi.org/10.1038/s41467-021-22210-3>
- 5 Ferreira-Gomes, M. *et al.* Recruitment of plasma cells from IL-21-dependent and IL-21-independent immune reactions to the bone marrow. *Nature Communications* **15**, 4182 (2024). <https://doi.org/10.1038/s41467-024-48570-0>

- 6 Ramaswamy, A. *et al.* Immune dysregulation and autoreactivity correlate with disease severity in SARS-CoV-2-associated multisystem inflammatory syndrome in children. *Immunity* **54**, 1083-1095.e1087 (2021). <https://doi.org/10.1016/j.immuni.2021.04.003>
- 7 Hoshino, A. *et al.* Dysregulation of Epstein-Barr Virus Infection in Hypomorphic ZAP70 Mutation. *The Journal of Infectious Diseases* **218**, 825-834 (2018). <https://doi.org/10.1093/infdis/jiy231>
- 8 Alosaimi, M. F. *et al.* Immunodeficiency and EBV-induced lymphoproliferation caused by 4-1BB deficiency. *J Allergy Clin Immunol* **144**, 574-583.e575 (2019). <https://doi.org/10.1016/j.jaci.2019.03.002>
- 9 Marsh, R. A. Epstein-Barr Virus and Hemophagocytic Lymphohistiocytosis. *Front Immunol* **8**, 1902 (2017). <https://doi.org/10.3389/fimmu.2017.01902>